# Rebounding Bandits for Modeling Satiation Effects

**Liu Leqi**
Machine Learning Department
Carnegie Mellon University
Pittsburgh, PA 15213
leqi@cs.cmu.edu

**Fatma Kılınç-Karzan**
Tepper School of Business
Carnegie Mellon University
Pittsburgh, PA 15213
fkilinc@andrew.cmu.edu

**Zachary C. Lipton**
Machine Learning Department
Carnegie Mellon University
Pittsburgh, PA 15213
zlipton@cmu.edu

**Alan L. Montgomery**
Tepper School of Business
Carnegie Mellon University
Pittsburgh, PA 15213
alanmontgomery@cmu.edu

## Abstract

Psychological research shows that enjoyment of many goods is subject to satiation, with short-term satisfaction declining after repeated exposures to the same item. Nevertheless, proposed algorithms for powering recommender systems seldom model these dynamics, instead proceeding as though user preferences were fixed in time. In this work, we introduce *rebounding bandits*, a multi-armed bandit setup, where satiation dynamics are modeled as time-invariant linear dynamical systems. Expected rewards for each arm decline monotonically with consecutive exposures and rebound towards the initial reward whenever that arm is not pulled. Unlike classical bandit algorithms, methods for tackling rebounding bandits must plan ahead and model-based methods rely on estimating the parameters of the satiation dynamics. We characterize the planning problem, showing that the greedy policy is optimal when the arms exhibit identical deterministic dynamics. To address stochastic satiation dynamics with unknown parameters, we propose Explore-Estimate-Plan, an algorithm that pulls arms methodically, estimates the system dynamics, and then plans accordingly.

## 1 Introduction

Recommender systems suggest such diverse items as music, news, restaurants, and even job candidates. Practitioners hope that by leveraging historical interactions, they might provide services better aligned with their users' preferences. However, despite their ubiquity in application, the dominant learning framework suffers several conceptual gaps that can result in misalignment between machine behavior and human preferences. For example, because human preferences are seldom directly observed, these systems are typically trained on the available observational data (e.g., purchases, ratings, or clicks) with the objective of predicting customer behavior [4, 27]. Problematically, such observations tend to be confounded (reflecting exposure bias due to the current recommender system) and subject to censoring (e.g., users with strong opinions are more likely to write reviews) [41, 16].

Even if we could directly observe the utility experienced by each user, we might expect it to depend, in part, on the history of past items consumed. For example, consider the task of automated (music) playlisting. As a user is made to listen to the same song over and over again, we might expect that the utility derived from each consecutive listen would decline [35]. However, after listening to other music for some time, we might expect the utility associated with that song to bounce back towards its baseline level. Similarly, a diner served pizza for lunch might feel diminished pleasure upon eating pizza again for dinner.

The psychology literature on *satiation* formalizes the idea that enjoyment depends not only on one's intrinsic preference for a given product but also on the sequence of previous exposures and the time between them [3, 6]. Research on satiation dates to the 1960s (if not earlier) with early studies addressing brand loyalty [42, 28]. Interestingly, even after controlling for marketing variables like price, product design, promotion, etc., researchers still observe brand-switching behavior in consumers. Such behavior, referred as *variety seeking*, has often been explained as a consequence of utility associated with the change itself [25, 17]. For a comprehensive review on hedonic decline caused by repeated exposure to a stimulus, we refer the readers to [11].

In this paper, we introduce *rebounding bandits*, a multi-armed bandits (MABs) [37] framework that models satiation via linear dynamical systems. While traditional MABs draw rewards from *fixed* but unknown distributions, rebounding bandits allow each arm's rewards to evolve as a function of both the per-arm characteristics (susceptibility to satiation and speed of rebounding) and the historical pulls (e.g., past recommendations). In rebounding bandits, even if the dynamics are known and deterministic, selecting the optimal sequence of $T$ arms to play requires planning in a Markov decision process (MDP) whose state space scales exponentially in the horizon $T$. When the satiation dynamics are known and stochastic, the states are only partially observable, since the satiation of each arm evolves with (unobserved) stochastic noises between pulls. And when the satiation dynamics are unknown, learning requires that we identify a stochastic dynamical system.

We propose Explore-Estimate-Plan (EEP) an algorithm that (i) collects data by pulling each arm repeatedly, (ii) estimates the dynamics using this dataset; and (iii) plans using the estimated parameters. We provide guarantees for our estimators in § 6.2 and bound EEP's regret in § 6.3.

Our main contributions are: (i) the rebounding bandits problem (§3), (ii) analysis showing that when arms share rewards and (deterministic) dynamics, the optimal policy pulls arms cyclically, exhibiting variety-seeking behavior (§4.1); (iii) an estimator (for learning the satiation dynamics) along with a sample complexity bound for identifying an affine dynamical system using a single trajectory of data (§6.2); (iv) EEP, an algorithm for learning with unknown stochastic dynamics that achieves sublinear $w$-step lookahead regret [34] (§6); and (v) experiments demonstrating EEP's efficacy (§7).

## 2 Related Work

Satiation effects have been addressed by such diverse disciplines as psychology, marketing, operations research, and recommendation systems. In the psychology and marketing literatures, satiation has been proposed as an explanation for variety-seeking consumer behavior [11, 25, 26]. In operations research, addressing continuous consumption decisions, [3] propose a deterministic linear dynamical system to model satiation effects. In the recommendation systems community, researchers have used semi-Markov models to explicitly model two states: (i) *sensitization*—where the user is highly interested in the product; and (ii) *boredom*—where the user is not engaged [18].

The bandits literature has proposed a variety of extensions where rewards depend on past exposures, both to address satiation and other phenomena. [14, 21, 39] tackle settings where each arm's expected reward grows (or shrinks) monotonically in the number of pulls. By contrast, [19, 2, 7] propose models where rewards increase as a function of the time elapsed since the last pull. [34] model the expected reward as a function of the time since the last pull drawn from a Gaussian Process with known kernel. [43] propose a model where rewards are linear functions of the recent history of actions and [29] model the reward as a function of a context that evolves according to known deterministic dynamics. In rested bandits [12], an arm's expected rewards change only when it is played, and in restless bandits [44] rewards evolve independently from the play of each arm.

**Key Differences**  This may be the first bandits paper to model evolving rewards through continuous-state linear stochastic dynamical systems with unknown parameters. Our framework captures several important aspects of satiation: rewards decline by diminishing amounts with consecutive pulls and rebound towards the baseline with disuse. Unlike models that depend only on fixed windows or the time since the last pull, our model expresses satiation more organically as a quantity that evolves according to stochastic dynamics. To estimate the reward dynamics, we leverage recent advances in the identification of linear dynamical systems [40, 38] that rely on the theory of self-normalized processes [33, 1] and block martingale conditions [40].

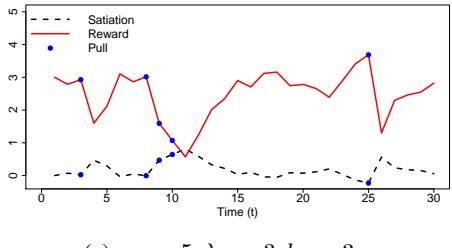 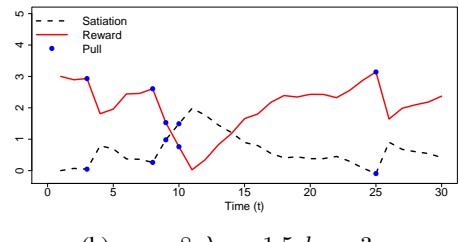

(a) $\gamma_k = .5, \lambda_k = 3, b_k = 3$          (b) $\gamma_k = .8, \lambda_k = 1.5, b_k = 3$

Figure 1: These plots illustrate the satiation level and reward of an arm from time 1 to 30. The two plots are generated with the same pull sequence, base rewards $b_k = 3$ and realized noises with variance $\sigma_z = .1$. In Figure 1a, $\gamma_k = .5$ and $\lambda_k = 3$. In Figure 1b, $\gamma_k = .8$ and $\lambda_k = 1.5$. In both cases, the arm has started with 0 as its base satiation level. **Black dashed line:** the satiation level. **Red solid line:** the reward. **Blue dots:** time steps where the arm is pulled.

## 3 Rebounding Bandits Problem Setup

Consider the set of $K$ arms $[K] := \{1, \dots, K\}$ with bounded base rewards $b_1, \dots, b_K$. Given a horizon $T$, a policy $\pi_{1:T} := (\pi_1, \dots, \pi_T)$ is a sequence of actions, where $\pi_t \in [K]$ depends on past actions and observed rewards. For any arm $k \in [K]$, we denote its pull history from time 0 to $T$ as the binary sequence $u_{k,0:T} := (u_{k,0}, \dots, u_{k,T})$, where $u_{k,0} = 0$ and for $t \in [T]$, $u_{k,t} = 1$ if $\pi_t = k$ and $u_{k,t} = 0$ otherwise. The subsequence of $u_{k,0:T}$ from $t_1$ to $t_2$ (including both endpoints) is denoted by $u_{k,t_1:t_2}$.

At time $t$, each arm $k$ has a satiation level $s_{k,t}$ that depends on a *satiation retention* factor $\gamma_k \in [0, 1)$, as follows

$$s_{k,t} := \gamma_k(s_{k,t-1} + u_{k,t-1}) + z_{k,t-1}, \ \forall t > t_0^k, \tag{1}$$

where $t_0^k := \min_t\{t : \ u_{k,t} = 1\}$ is the first time arm $k$ is pulled and $z_{k,t-1}$ is independent and identically distributed noise drawn from $\mathcal{N}(0, \sigma_z^2)$, accounting for incidental (uncorrelated) factors in the satiation dynamics. Because satiation requires exposure, arms only begin to have nonzero satiation levels after their first pull, i.e., $s_{k,0} = \dots = s_{k,t_0^k} = 0$.

At time $t \in [T]$, if arm $k$ is played with a current satiation level $s_{k,t}$, the agent receives reward $\mu_{k,t} := b_k - \lambda_k s_{k,t}$, where $b_k$ is the base reward for arm $k$ and $\lambda_k \geq 0$ is a bounded *exposure influence* factor. We use *satiation influence* to denote the product of the exposure influence factor $\lambda_k$ and the satiation level $s_{k,t}$. In Figure 1, we show how rewards evolve in response to both pulls and the stochastic dynamics under two sets of parameters. The expected reward of arm $k$ (where the expectation is taken over all noises associated with the arm) monotonically decreases by diminishing amounts with consecutive pulls and increases with disuse by diminishing amounts.

*Remark* 1. We note that there exist choices of $b_k, \gamma_k, \lambda_k$ for which the expected reward of arm $k$ can be negative. In the traditional bandits setup, one must pull an arm at every time step. Thus, what matters are the relative rewards and the problem is mathematically identical, regardless of whether the expected rewards range from $-10$ to $0$ or $0$ to $10$. In addition, one might construct settings where negative expected rewards are reasonable. For example, when one of the arms corresponds to no recommendation with 0 being its expected reward (e.g., $b_k = 0$, $\lambda_k = 0$), then the interpretation of negative expected reward would be that pulling (recommending) the corresponding arm (item) is less preferred relative to not pulling (no recommendation).

Given horizon $T \geq 1$, we seek a pull sequence $\pi_{1:T}$, where $\pi_t$ depends on past rewards and actions $(\pi_1, \mu_{\pi_1,1}, \dots, \pi_{t-1}, \mu_{\pi_{t-1},t-1})$, that maximizes the expected cumulative reward:

$$G_T(\pi_{1:T}) := \mathbb{E}\left[\sum_{t=1}^T \mu_{\pi_t,t}\right]. \tag{2}$$

**Additional Notation**    Let $\overline{\gamma} := \max_{k \in [K]} \gamma_k$ and $\overline{\lambda} := \max_{k \in [K]} \lambda_k$. We use $a \lesssim b$ when $a \leq Cb$ for some positive constant $C$.

# 4 Planning with Known Dynamics

Before we can hope to learn an optimal policy with unknown stochastic dynamics, we need to establish a procedure for planning when the satiation retention factors, exposure influence factors, and base rewards are known. We begin by presenting several planning strategies and analyzing them under deterministic dynamics, where the past pulls exactly determine each arm's satiation level, i.e., $s_{k,t} = \gamma_k(s_{k,t-1} + u_{k,t-1})$, $\forall t > t_0^k$. With some abuse of notation, at time $t \geq 2$, given a pull sequence $u_{k,0:t-1}$, we can express the satiation and the expected[1] reward of each arm as

$$s_{k,t}(u_{k,0:t-1}) = \gamma_k\left(s_{k,t-1} + u_{k,t-1}\right) = \gamma_k\left(\gamma_k\left(s_{k,t-2} + u_{k,t-2}\right)\right) + \gamma_k u_{k,t-1} = \sum_{i=1}^{t-1} \gamma_k^{t-i} u_{k,i},$$

$$\mu_{k,t}(u_{k,0:t-1}) = b_k - \lambda_k\left(\sum_{i=1}^{t-1} \gamma_k^{t-i} u_{k,i}\right). \tag{3}$$

At time $t = 1$, we have that $s_{k,1}(u_{k,0:0}) = 0$ and $\mu_{k,1}(u_{k,0:0}) = b_k$ for all $k \in [K]$. Since the arm parameters $\{\lambda_k, \gamma_k, b_k\}_{k=1}^K$ are known, our goal (2) simplifies to finding a pull sequence that solves the following bilinear integer program:

$$\max_{u_{k,t}} \left\{ \sum_{k=1}^K \sum_{t=1}^T u_{k,t}\left(b_k - \lambda_k \sum_{i=0}^{t-1} \gamma_k^{t-i} u_{k,i}\right) : \begin{array}{l} \sum_{k=1}^K u_{k,t} = 1, \quad \forall t \in [T], \\ u_{k,t} \in \{0,1\}, \quad u_{k,0} = 0, \forall k \in [K], \forall t \in [T] \end{array} \right\} \tag{4}$$

where the objective maximizes the expected cumulative reward associated with the pull sequence and the constraints ensure that at each time period we pull exactly one arm. Note that (4) includes products of decision variables $u_{k,t}$ leading to bilinear terms in the objective. In Appendix A, we provide an equivalent integer linear program.

## 4.1 The Greedy Policy

At each step, the greedy policy $\pi^g$ picks the arm with the highest instantaneous expected reward. Formally, at time $t$, given the pull history $\{u_{k,0:t-1}\}_{k=1}^K$, the greedy policy picks

$$\pi_t^g \in \arg\max_{k \in [K]} \mu_{k,t}(u_{k,0:t-1}).$$

In order to break ties, when all arms have the same expected reward, the greedy policy chooses the arm with the lowest index.

Note that the greedy policy is not, in general, optimal. Sometimes, we are better off allowing the current best arm to rebound even further, before pulling it again.

**Example 1.** *Consider the case with two arms. Suppose that arm 1 has base reward $b_1$, satiation retention factor $\gamma_1 \in (0,1)$, and exposure influence factor $\lambda_1 = 1$. For any fixed time horizon $T > 2$, suppose that arm 2 has $b_2 = b_1 + \frac{\gamma_2 - \gamma_2^T}{1 - \gamma_2}$ where $\gamma_2 \in (0,1)$ and $\lambda_2 = 1$. The greedy policy $\pi_{1:T}^g$ will keep pulling arm 2 until time $T - 1$ and then play arm 1 (or arm 2) at time $T$. This is true because if we keep pulling arm 2 until $T - 1$, at time $T$, we have $\mu_{2,T}(u_{2,0:T-1}) = b_1 = \mu_{1,T}(u_{1,0:T-1})$. However, the policy $\pi_{1:T}^n$, where $\pi_t^n = 2$ if $t \leq T - 2$, $\pi_{T-1}^n = 1$, and $\pi_T^n = 2$, obtains a higher expected cumulative reward. In particular, the difference $G_T(\pi_{1:T}^n) - G_T(\pi_{1:T}^g)$ will be $\gamma_2 - \gamma_2^{T-1}$.*

## 4.2 When is Greedy Optimal?

When the satiation effect is always 0, e.g., when the satiation retention factors $\gamma_k = 0$ for all $k \in [K]$, we know that the greedy policy (which always plays the arm with the highest instantaneous expected reward) is optimal. However, when satiation can be nonzero, it is less clear under what conditions the greedy policy performs optimally. This question is of special interest when we consider human decision-making, since we cannot expect people to solve large-scale bilinear integer programs every time they pick music to listen to.

In this section, we show that when all arms share the same properties ($\gamma_k, \lambda_k, b_k$ are identical for $k \in [K]$), the greedy policy is optimal. In this case, the greedy policy exhibits variety-seeking behavior as it plays the arms cyclically. Interestingly, this condition aligns with early research that

---

[1]We use "expected reward" to emphasize that all results in this section apply to settings where the satiation dynamics are deterministic but the rewards are stochastic, i.e., $\mu_{k,t} = b_k - \lambda_k s_{k,t} + e_{k,t}$ for independent mean-zero noises $e_{k,t}$.

has motivated studies on satiation [42, 28]: when controlling for marketing variables (e.g., the arm parameters $\gamma_k, \lambda_k, b_k$), researchers still observe variety-seeking behaviors of consumers (e.g., playing arms in a cyclic order).

**Assumption 1.** $\gamma_1 = \ldots = \gamma_K = \gamma$, $\lambda_1 = \ldots = \lambda_K = \lambda$, and $b_1 = \ldots = b_K = b$.

We start with characterizing the greedy policy when Assumption 1 holds.

**Lemma 1** (Greedy Policy Characterization). *Under Assumption 1 and the tie-breaking rule that when all arms have the same expected reward, the greedy policy chooses the one with the lowest arm index, the sequence of arms pulled by the greedy policy forms a periodic sequence: $\pi_1 = 1, \pi_2 = 2, \cdots, \pi_K = K$, and $\pi_{t+K} = \pi_t$, $\forall t \in \mathbb{N}_+$.*

In this case, the greedy policy is equivalent to playing the arms in a cyclic order. All proofs for the paper are deferred to the Appendices.

**Theorem 1.** *Under Assumption 1, given any horizon $T$, the greedy policy $\pi_{1:T}^g$ is optimal.*

*Remark* 2. Theorem 1 suggests that when the (deterministic) satiation dynamics and base rewards are identical across arms, planning does not require knowledge of those parameters, since playing the arms in a cyclic order is optimal.

Lemma 1 and Theorem 1 lead us to conclude the following result: when recommending items that share the same properties, the best strategy is to show the users a variety of recommendations by following the greedy policy.

On a related note, Theorem 1 also gives an exact Max K-Cut of a complete graph $\mathcal{K}_T$ on $T$ vertices, where the edge weight connecting vertices $i$ and $j$ is given by $e(i,j) = \lambda \gamma^{|j-i|}$ for $i \neq j$. The Max K-Cut problem partitions the vertices of a graph into $K$ subsets $P_1, \ldots P_K$, such that the sum of the edge weights connecting the subsets are maximized [10]. Mapping the Max K-Cut problem back to our original setup, each vertex $i$ represents a time step. If vertex $i$ is assigned to subset $P_k$, it suggests that arm $k$ should be played at time $i$. The edge weights $e(i,j) = \lambda \gamma^{|j-i|}$ for $i \neq j$ can be seen as the reduction in satiation influence achieved by not playing the same arm at both time $i$ and time $j$. The goal (4) is to maximize the total satiation influence reduction.

**Proposition 2** (Connection to Max K-Cut). *Under Assumption 1, an optimal solution to* (4) *is given by a Max K-Cut on $\mathcal{K}_T$, where $\mathcal{K}_T$ is a complete graph on $T$ vertices with edge weights $e(i,j) = \lambda \gamma^{|j-i|}$ for all $i \neq j$.*

Using Lemma 1 and Theorem 1, we obtain an exact Max K-Cut of $\mathcal{K}_T$: $\forall k \in [K], P_k = \{t \in [T] : t \equiv k \ (\text{mod } K)\}$.

### 4.3 The $w$-lookahead Policy

To model settings where the arms correspond to items with different characteristics (e.g., we can enjoy tacos on consecutive days but require time to recover from a trip to the steakhouse) we must allow the satiation parameters to vary across arms. Here, the greedy policy may not be optimal. Thus, we consider more general lookahead policies (the greedy policy is a special case). Given a window of size $w$ and the current satiation levels, the $w$-lookahead policy picks actions to maximize the total reward over the next $w$ time steps. Let $l$ denote $\lceil T/w \rceil$. Define $t_i = \min\{iw, T\}$ for $i \in [l]$ and $t_0 = 0$. More formally, the $w$-lookahead policy $\pi_{1:T}^w$ is defined as follows: for any $i \in [l]$, given the previously chosen arms' corresponding pull histories $\{u_{k,0:t_{i-1}}^w\}_{k=1}^K$ where $u_{k,0}^w = 0$ and $u_{k,t}^w = 1$ if (and only if) $\pi_t^w = k$, the next $w$ (or $T$ mod $w$) actions $\pi_{t_{i-1}+1:t_i}^w$ are given by

$$
\max_{\pi_{t_{i-1}+1:t_i}^w} \left\{ \sum_{t=t_{i-1}+1}^{t_i} \mu_{\pi_t^w, t}(u_{\pi_t, 0:t-1}) : \begin{array}{l} u_{k,0:t_{i-1}} = u_{k,0:t_{i-1}}^w, \quad \forall k \in [K], \\ \sum_{k=1}^K u_{k,t} = 1, \quad \forall t \in [t_i], \\ u_{k,t} \in \{0,1\}, \quad \forall k \in [K], t \in [t_i] \end{array} \right\} \tag{5}
$$

In the case of a tie, one can pick any of the sequences that maximize (5). We recover the greedy policy when the window size $w = 1$, and finding the $w$-lookahead policy for the window size $w = T$ is equivalent to solving (4).

*Remark* 3. Another reasonable lookahead policy, which requires planning ahead at every time step, would be the following: at every time $t$, plan for the next $w$ actions and follow them for a single time step. To lighten the computational load, we adopt the current $w$-lookahead policy which only requires planning every $w$ time steps.

For the rest of the paper, we use $\texttt{Lookahead}(\{\lambda_k, \gamma_k, b_k\}_{k=1}^K, \{u_{k,0:t_{i-1}}^w\}_{k=1}^K, t_{i-1}, t_i)$ to refer to the solution of (5), when the arm parameters are $\{\lambda_k, \gamma_k, b_k\}_{k=1}^K$, the historical pull sequences of all arms till time $t_{i-1}$ are given by $\{u_{k,0:t_{i-1}}^w\}_{k=1}^K$. The solution corresponds to the actions that should be taken by the $w$-lookahead policy for the next $t_i - t_{i-1}$ time steps.

**Theorem 2.** *Given any horizon $T$, let $\pi_{1:T}^*$ be a solution to (4). For a fixed window size $w \leq T$, we have that*

$$G_T(\pi_{1:T}^*) - G_T(\pi_{1:T}^w) \leq \frac{\overline{\lambda}\overline{\gamma}(1 - \overline{\gamma}^{T-w})}{(1 - \overline{\gamma})^2} \lceil T/w \rceil.$$

*Remark* 4. Note that when $w = T$, the $w$-lookahead policy by definition is the optimal policy and in such a case, the upper bound for the optimality gap of $w$-lookahead established in Theorem 2 is also 0. In contrast to the optimal policy, the computational benefit of the $w$-lookahead policy becomes apparent when the horizon $T$ is large since it requires solving for a much smaller program (5). In general, the $w$-lookahead policy is expected to perform much better than the greedy policy (which corresponds to the case of $w = 1$) at the expense of a higher computational cost. Finally, we note that for the window size of $w = \sqrt{T}$, we obtain $G_T(\pi_{1:T}^*) - G_T(\pi_{1:T}^w) \leq O(\sqrt{T})$.

## 5 Learning with Unknown Dynamics: Preliminaries

When the satiation dynamics are unknown and stochastic ($\sigma_z > 0$), the learner faces a continuous-state partially observable MDP because the satiation levels are not observable. To set the stage, we first introduce our state representation (§ 5.1) and a regret-based performance measure (§ 5.2). In the next section, we will introduce EEP, our algorithm for rebounding bandits.

### 5.1 State Representation

Following [32], at any time $t \in [T]$, we define a state vector $x_t$ in the state space $\mathcal{X}$ to be $x_t = (x_{1,t}, n_{1,t}, x_{2,t}, n_{2,t}, \ldots, x_{K,t}, n_{K,t})$, where $n_{k,t} \in \mathbb{N}$ is the number of steps at time $t$ since arm $k$ was last selected and $x_{k,t}$ is the satiation influence (product of $\lambda_k$ and the satiation level) as of the most recent pull of arm $k$. Since the most recent pull happens at $t - n_{k,t}$, we have $x_{k,t} = b_k - \mu_{k,t-n_{k,t}} = \lambda_k s_{k,t-n_{k,t}}$. Recall that $\mu_{k,t-n_{k,t}}$ is the reward collected by pulling arm $k$ at time $t - n_{k,t}$. Note that $b_k$ is directly observed when arm $k$ is pulled for the first time because there is no satiation effect. The state at the first time step is $x_1 = (0, \ldots, 0)$. At time $t$, if arm $k$ is chosen at state $x_t$, and reward $\mu_{k,t}$ is obtained, then the next state $x_{t+1}$ will satisfy (i) for the pulled arm $k$, $n_{k,t+1} = 1$ and $x_{k,t+1} = b_k - \mu_{k,t}$; (ii) for other arms $k' \neq k$, $n_{k',t+1} = n_{k',t} + 1$ if $n_{k',t} \neq 0$, $n_{k',t+1} = 0$ if $n_{k',t} = 0$, and the satiation influence remains the same $x_{k',t+1} = x_{k',t}$.

Given $\{\gamma_k, \lambda_k, b_k\}_{k=1}^K$, the reward function $r : \mathcal{X} \times [K] \to \mathbb{R}$ represents the *expected* reward of pulling arm $k$ under state $x_t$:

If $n_{k,t} = 0$, then $r(x_t, k) = b_k$. If $n_{k,t} \geq 1$, $r(x_t, k) = \mathbb{E}[\mu_{k,t}|x_t] = b_k - \gamma_k^{n_{k,t}} x_{k,t} - \lambda_k \gamma_k^{n_{k,t}}$, where the expectation is taken over the noises in between the current pull and the last pull of arm $k$. See Appendix C.1 for the full description of the MDP setup (including the transition kernel and value function definition) of rebounding bandits.

### 5.2 Evaluation Criteria: $w$-step Lookahead Regret

In reinforcement learning (RL), the performance of a learner is often measured through a regret that compares the expected cumulative reward obtained by the learner against that of an optimal policy in a competitor class [20]. In most episodic (e.g., finite horizon) RL literature [31, 15], regrets are defined in terms of episodes. In such cases, the initial state is reset (e.g., to a fixed state) after each episode ends, independent of previous actions taken by the leaner. Unlike these episodic RL setups, in rebounding bandits, we cannot restart from the initial state because the satiation level cannot be reset and user's memory depends on past received recommendations. Instead, [34] proposed a version of $w$-*step lookahead regret* that divides the $T$ time steps into $\lceil T/w \rceil$ episodes where each episode (besides the last) consists of $w$ time steps. At the beginning of each episode, the initial state is reset but depends on how the learner has interacted with the user previously. In particular, at the beginning of episode $i + 1$ (at time $t = iw + 1$), given that the learner has played $\pi_{1:iw}$ with corresponding pull sequence $u_{k,0:iw}$ for $k \in [K]$, we reset the initial state to be $x^i = (\mu_{1,iw+1}(u_{1,0:iw}), n_{1,iw+1}, \ldots, \mu_{K,iw+1}(u_{K,0:iw}), n_{K,iw+1})$ where $\mu_{k,t}(\cdot)$ is defined in (3) and $n_{k,iw+1}$ is the number of steps since arm $k$ is last pulled by the learner as of time $iw + 1$.

Then, given the learner's policy $\pi_{1:T}$, where $\pi_t : \mathcal{X} \to [K]$, the $w$-step lookahead regret, against a competitor class $\mathcal{C}^w$ (which we define later), is defined as follows:

$$\text{Reg}^w(T) = \sum_{i=0}^{\lceil T/w \rceil - 1} \max_{\tilde{\pi}_{1:w} \in \mathcal{C}^w} \mathbb{E}\left[ \sum_{j=1}^{\min\{w, T-iw\}} r(x_{iw+j}, \tilde{\pi}_j(x_{iw+j})) \Big| x_{iw+1} = x^i \right]$$
$$- \mathbb{E}\left[ \sum_{j=1}^{\min\{w, T-iw\}} r(x_{iw+j}, \pi_{iw+j}(x_{iw+j})) \Big| x_{iw+1} = x^i \right], \tag{6}$$

where the expectation is taken over $x_{iw+2}, \ldots, x_{\min\{(i+1)w, T\}}$.

The competitor class $\mathcal{C}^w$ that we have chosen consists of policies that depend on time steps, i.e., $\mathcal{C}^w = \{\tilde{\pi}_{1:w} : \tilde{\pi}_t = \tilde{\pi}_t(x_t) = \tilde{\pi}_t(x'_t), \tilde{\pi}_t \in [K], \forall t \in [w], x_t, x'_t \in \mathcal{X}\}$. We note that $\mathcal{C}^w$ subsumes many traditional competitor classes in bandits literature, including the class of fixed-action policies considered in adversarial bandits [20] and the class of periodic ranking policies [7]. In our paper, the $w$-lookahead policy (including the $T$-lookahead policy given by (4)) is a time-dependent policy that belongs to $\mathcal{C}^w$, since at time $t$, it will play a fixed action by solving (5) using the true reward parameters $\{\lambda_k, \gamma_k, b_k\}_{k=1}^K$. The time-dependent competitor class $\mathcal{C}^w$ differs from a state-dependent competitor class which includes all measurable functions $\tilde{\pi}_t$ that map from $\mathcal{X}$ to $[K]$. The state-dependent competitor class contains the optimal policy $\pi^*$ where $\pi_t^*(x_t)$ depends on not just the time step but also the exact state $x_t$. Finding the optimal state-dependent policy requires optimal planning for a continuous-state MDP, which relies on state space discretizion [31] or function approximation (e.g., approximate dynamic programming algorithms [30, 9, 36]). In Appendix C, we provide discussion and analysis on an algorithm compared against the optimal state-dependent policy. We proceed the rest of the main paper with $\mathcal{C}^w$ defined above.

When $w = 1$, the 1-step lookahead regret is also known as the instantaneous regret, which is commonly used in restless bandits literature and some nonstationary bandits papers including [29]. Note that low instantaneous regret does not imply high expected cumulative reward in the long-term, i.e., one may benefit more by waiting for certain arms to rebound. When $w = T$, we recover the full horizon regret. As we have noted earlier, finding the optimal competitor policy in this case is computationally intractable because the number of states, even when the satiation dynamics are deterministic, grows exponentially with the horizon $T$. Finally, we note that the $w$-step lookahead regret can be obtained for not just policies designed to look $w$ steps ahead but any given policy. For a more comprehensive discussion on these notions of regret, see [34, Section 4].

## 6  Explore-Estimate-Plan

We now present *Explore-Estimate-Plan (EEP)*, an algorithm for learning in rebounding bandits with stochastic dynamics and unknown parameters, that (i) collects data by pulling each arm a fixed number of times; (ii) estimates the model's parameters based on the logged data; and then (iii) plans according to the estimated model. Finally, we analyze EEP's regret.

Because each arm's base reward is known from the first pull, whenever arm $k$ is pulled at time $t$ and $n_{k,t} \neq 0$, we measure the satiation influence $\lambda_k s_{k,t}$, which becomes the next state $x_{k,t+1}$:

$$x_{k,t+1} = \lambda_k s_{k,t} = \lambda_k \gamma_k^{n_{k,t}} s_{k,t-n_{k,t}} + \lambda_k \gamma_k^{n_{k,t}} + \lambda_k \sum_{i=0}^{n_{k,t}-1} \gamma_k^i z_{k,t-1-i}$$
$$= \gamma_k^{n_{k,t}} x_{k,t+1-n_{k,t}} + \lambda_k \gamma_k^{n_{k,t}} + \lambda_k \sum_{i=0}^{n_{k,t}-1} \gamma_k^i z_{k,t-1-i}. \tag{7}$$

We note that the current state $x_{k,t}$ equals $x_{k,t+1-n_{k,t}}$, since $x_{k,t+1-n_{k,t}}$ is the last observed satiation influence for arm $k$ and $n_{k,t}$ is the number of steps since arm $k$ was last pulled.

### 6.1  The Exploration Phase: Repeated Pulls

We collect a dataset $\mathcal{P}_k^n$ by consecutively pulling each arm $n+1$ times, in turn, where $n \geq \lfloor T^{2/3}/K \rfloor$ (Line 4-7 of Algorithm 1). Specifically, for each arm $k \in [K]$, the dataset $\mathcal{P}_k^n$ contains a single trajectory of $n+1$ observed satiation influences $\tilde{x}_{k,1}, \ldots, \tilde{x}_{k,n+1}$, where $\tilde{x}_{k,1} = 0$ and $\tilde{x}_{k,j}$ ($j > 1$) is the difference between the first reward and the $j$-th reward from arm $k$. Thus, for $\tilde{x}_{k,j}, \tilde{x}_{k,j+1} \in \mathcal{P}_k^n$, using (7) with $n_{k,t} = 1$ (because pulls are consecutive), it follows that

$$\tilde{x}_{k,j+1} = \gamma_k \tilde{x}_{k,j} + d_k + \tilde{z}_{k,j}, \tag{8}$$

where $d_k = \lambda_k \gamma_k$ and $\tilde{z}_{k,j}$ are independent samples from $\mathcal{N}(0, \sigma_{z,k}^2)$ with $\sigma_{z,k}^2 = \lambda_k^2 \sigma_z^2$. In Appendix E.2, we discuss other exploration strategies (e.g., playing the arms in a cyclic order) for EEP and their regret guarantees.

---
**Algorithm 1:** $w$-lookahead Explore-Estimate-Plan
---
**Input:** Lookahead window size $w$, Number of arms $K$, Horizon $T$

1  **Initialize** $t = 1$, $\pi_{1:T}$ to be an empty array of length $T$ and $\widetilde{T} = T^{2/3} + w - (T^{2/3} \bmod w)$.

2  **for** $k = 1, \ldots, K$ **do**

3     Set $t' = t$ and initialize an empty array $\mathcal{P}_k^n$.

4     **for** $c = 0, \ldots, \lfloor \widetilde{T}/K \rfloor$ **do**

5        Play arm $k$ to obtain reward $\mu_{k,t'+c}$ and add $\mu_{k,t'} - \mu_{k,t'+c}$ to $\mathcal{P}_k^n$.

6        Set $\pi_t = k$ and increase $t$ by 1.

7     **end**

8     Obtain $\widehat{\gamma}_k, \widehat{d}_k$ using the estimator (9),set $\widehat{\lambda}_k = |\widehat{d}_k/\widehat{\gamma}_k|$ and $\widehat{b}_k = \mu_{k,t'}$.

9  **end**

10  Let $t_0 = \widetilde{T}$, set $\pi_{t:t_0} = (1, \ldots, \widetilde{T} - t + 1)$, and play $\pi_{t:t_0}$.

11  **for** $i = 1, \ldots, \lceil \frac{T - t_0}{w} \rceil$ **do**

12     Set $t_i = \min\{t_{i-1} + w, T\}$.

13     Obtain $\pi_{t_{i-1}+1:t_i} = \texttt{Lookahead}(\{\widehat{\lambda}_k, \widehat{\gamma}_k, \widehat{b}_k\}_{k=1}^K, \{u_{k,0:t_{i-1}}\}_{k=1}^K, t_{i-1}, t_i)$ where
       $\{u_{k,0:t_{i-1}}\}_{k=1}^K$ are the arm pull histories correspond to $\pi_{1:t_{i-1}}$.

14     Play $\pi_{t_{i-1}+1,t_i}$.

15  **end**
---

## 6.2 Estimating the Reward Model and Satiation Dynamics

For all $k \in [K]$, given the dataset $\mathcal{P}_k^n$, we estimate $A_k = (\gamma_k, d_k)^\top$ using the *ordinary least squares estimator*:

$$\widehat{A}_k \in \underset{A \in \mathbb{R}^2}{\arg\min} \|\mathbf{Y_k} - \overline{\mathbf{X}}_\mathbf{k} A\|_2^2, \tag{9}$$

where $\mathbf{Y_k} \in \mathbb{R}^n$ is an $n$-dimensional vector whose $j$-th entry is $\tilde{x}_{k,j+1}$ and $\overline{\mathbf{X}}_\mathbf{k} \in \mathbb{R}^{n \times 2}$ takes as its $j$-th row the vector $\overline{x}_{k,j} = (\tilde{x}_{k,j}, 1)^\top$, i.e., $\tilde{x}_{k,j+1}$ is treated to be the response to the covariates $\overline{x}_{k,j}$. For $n \geq 2$, we have that

$$\widehat{A}_k = \begin{pmatrix} \widehat{\gamma}_k \\ \widehat{d}_k \end{pmatrix} = \left( \overline{\mathbf{X}}_\mathbf{k}^\top \overline{\mathbf{X}}_\mathbf{k} \right)^{-1} \overline{\mathbf{X}}_\mathbf{k}^\top \mathbf{Y_k}, \tag{10}$$

and we take $\widehat{\lambda}_k = |\widehat{d}_k/\widehat{\gamma}_k|$.

The difficulty in analyzing the ordinary least squares estimator (10) for identifying an affine dynamical system (8) using a single trajectory of data comes from the fact that the samples are not independent. Asymptotic guarantees of the ordinary least squares estimators in this case have been studied previously in the control theory and time series communities [13, 22]. Recent work on system identifications for linear dynamical systems focuses on the sample complexity [40, 38]. Adapting the proof of [40, Theorem 2.4], we derive the following theorem for identifying our affine dynamical system (8).

**Theorem 3.** *Fix* $\delta \in (0, 1)$. *For all* $k \in [K]$, *there exists a constant* $n_0(\delta, k)$ *such that if the dataset* $\mathcal{P}_k^n$ *satisfies* $n \geq n_0(\delta, k)$, *then*

$$\mathbb{P}\left( \|\widehat{A}_k - A_k\|_2 \gtrsim \sqrt{1/(\psi n)} \right) \leq \delta,$$

*where* $\psi = \sqrt{\min\left\{ \frac{\sigma_{z,k}^2(1-\gamma_k)^2}{16d_k^2(1-\gamma_k^2) + (1-\gamma_k)^2\sigma_{z,k}^2}, \frac{\sigma_{z,k}^2}{4(1-\gamma_k^2)} \right\}}$.

As shown in Theorem 3, when $d_k = \lambda_k \gamma_k$ gets larger, the convergence rate for $\widehat{A}_k$ gets slower. Given a single trajectory of sufficient length, we obtain $|\widehat{\gamma}_k - \gamma_k| \leq O(1/\sqrt{n})$ and $|\widehat{d}_k - d_k| \leq O(1/\sqrt{n})$. In Corollary 4, we show that the estimator of $\lambda_k$ also achieves $O(1/\sqrt{n})$ estimation error.

**Corollary 4.** *Fix* $\delta \in (0, 1)$. *Suppose that for all* $k \in [K]$, *we have* $\mathbb{P}(\|\widehat{A}_k - A_k\|_2 \gtrsim 1/\sqrt{n}) \leq \delta$ *and* $\widehat{\gamma}_k > 0$. *Then, with probability* $1 - \delta$, *we have that for all* $k \in [K]$,

$$|\widehat{\gamma}_k - \gamma_k| \leq O\left( \frac{1}{\sqrt{n}} \right), \quad |\widehat{\lambda}_k - \lambda_k| \leq O\left( \frac{1}{\sqrt{n}} \right).$$

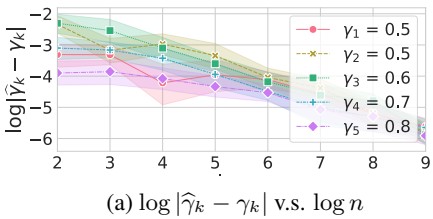 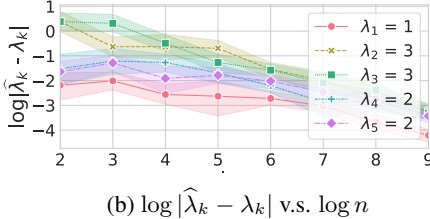

(a) $\log |\widehat{\gamma}_k - \gamma_k|$ v.s. $\log n$     (b) $\log |\widehat{\lambda}_k - \lambda_k|$ v.s. $\log n$

Figure 2: Figure 2a and 2b are the $\log$-$\log$ plots of absolute errors of $\widehat{\gamma}_k$ and $\widehat{\lambda}_k$ with respect to the number of samples $n$ in a single trajectory. The results are averaged over 30 random runs, where the shaded area represents one standard deviation.

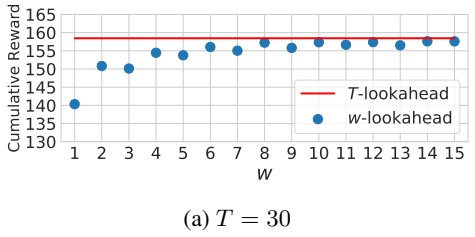 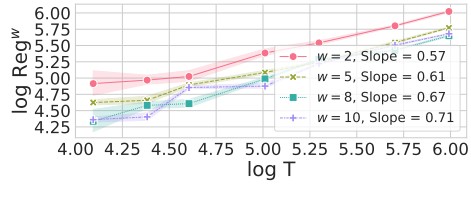

(a) $T = 30$     (b) $\log \operatorname{Reg}^w$ v.s. $\log T$

Figure 3: Figure 3a shows the expected cumulative reward collected by the $T$-lookahead policy (red line) and $w$-lookahead policy (blue dots) when $T = 30$. Figure 3b shows the log-log plot of the $w$-step lookahead regret of $w$-lookahead EEP under different $T$ averaged over 20 random runs.

### 6.3 Planning and Regret Bound

In the planning stage of Algorithm 1 (Line 11-15), at time $t_{i-1} + 1$, the next $w$ arms to play are obtained through the Lookahead function defined in (5) based on the estimated parameters from the estimation stage (Line 8). Using the results in Corollary 4, we obtain the following sublinear regret bound for $w$-lookahead EEP.

**Theorem 5.** *There exists a constant $T_0$ such that for all $T > T_0$ and $w \leq T^{2/3}$, the $w$-step lookahead regret of $w$-lookahead Explore-Estimate-Plan satisfies*

$$\operatorname{Reg}^w(T) \leq O(K^{1/2}T^{2/3}\log T).$$

*Remark* 5. The fact that EEP incurs a regret of order $O(T^{2/3})$ is expected for two reasons: First, EEP can be viewed as an explore-then-commit (ETC) algorithm that first explores then exploits. The regret of EEP resembles the $O(T^{2/3})$ regret of the ETC algorithm in the classical $K$-armed bandits setting [20]. In rebounding bandits, the fundamental obstacle to mixing the exploration and exploitation stages is the need to estimate the satiation dynamics. When the rewards of each arm are not observed periodically, the obtained satiation influences can no longer be viewed as samples from the same time-invariant affine dynamical system, since the parameters of the system depend on the duration between pulls. In practice, one may utilize the maximum likelihood estimator to obtain estimates of the reward parameters but obtaining the sample complexity of such an estimator with dependent data is difficult. Second, it has been shown in [5] that when the rewards of the arms have temporal variation that depends on the horizon $T$, the worst case instantaneous regret has a lower bound $\Omega(T^{2/3})$. On the other hand, in the traditional $K$-armed bandits setup, the regret (following the classical definition [20]) is lower bounded by $\Omega(T^{1/2})$, and can be attained by methods like the upper confidence bound algorithm [20]. Precisely characterizing the regret lower bound for rebounding bandits is of future interest.

## 7 Experiments

We now evaluate the performance of EEP experimentally, separately investigating the sample efficiency of our proposed estimators (10) for learning the satiation and reward models (Figure 2) and the computational performance of the $w$-lookahead policies (5) (Figure 3a). For the experimental setup, we have 5 arms with satiation retention factors $\gamma_1 = \gamma_2 = .5, \gamma_3 = .6, \gamma_4 = .7, \gamma_5 = .8$, exposure influence factors $\lambda_1 = 1, \lambda_2 = \lambda_3 = 3, \lambda_4 = \lambda_5 = 2$, base rewards $b_1 = 2, b_2 = 3, b_3 = 4, b_4 = 2, b_5 = 10$, and noise with variance $\sigma_z = 0.1$.

**Parameter Estimation**    We first evaluate our proposed estimator for using a single trajectory per arm to estimate the arm parameters $\gamma_k, \lambda_k$. In Figure 2, we show the absolute error (averaged over 30 random runs) between the estimated parameters and the true parameters for each arm. Aligning with our theoretical guarantees (Corollary 4), the log-log plots show that the convergence rate of the absolute error is on the scale of $O(n^{-1/2})$.

$w$-**lookahead Performance**    To evaluate $w$-lookahead policies, we solve (5) using the true reward parameters and report expected cumulative rewards of the obtained $w$-lookahead policies (Figure 3a). Recall that the greedy policy is precisely the 1-lookahead policy. In order to solve the resulting integer programs, we use Gurobi 9.1 [23] and set the number of threads for solving the problem to 10. When $T = 30$, the $T$-lookahead policy (expected cumulative rewards given by the red line in Figure 3a) solved through (4) is obtained in 1610s. On the other hand, all $w$-lookahead policies (expected cumulative rewards given by the blue dots in Figure 3a) for $w$ in between 1 and 15 are solved within 2s. We provide the results when $T = 100$ in Appendix G. Despite using significantly lower computational time, $w$-lookahead policies achieve a similar expected cumulative reward to the $T$-lookahead policy.

**EEP Performance**    We evaluate the performance of EEP when $T$ ranges from 60 to 400. For each horizon $T$, we examine the $w$-step lookahead regret of $w$-lookahead EEP where $w = 2, 5, 8, 10$. All results are averaged over 20 random runs. As $T$ increases, the exploration stage of EEP becomes longer, which results in collecting more data for estimating the reward parameters and lower variance of the parameter estimators. We fit a line for the regrets with the same lookahead size $w$ to examine the order of the regret with respect to the horizon $T$. The slopes of the lines (see Figure 3b's legend) are close to $2/3$, which aligns with our theoretical guarantees (Theorem 5), i.e., the regrets are on the order of $O(T^{2/3})$. In Appendix G, we present the $T$-step lookahead regret of $w$-lookahead EEP under the same settings, and additional experimental setups and results.

## 8    Conclusions

While our work has taken strides towards modeling the exposure-dependent evolution of preferences through dynamical systems, there are many avenues for future work. First, while our satiation dynamics are independent across arms, a natural extension might allow interactions among the arms. For example, a diner sick of pizza after too many trips to Di Fara's, likely would also avoid Grimaldi's until the satiation effect wore off. On the system identification side, we might overcome our reliance on evenly spaced pulls, producing more adaptive algorithms (e.g., optimism-based algorithms) that can refine their estimates, improving the agent's policy even past the pure exploration period. Finally, our satiation model captures just one plausible dynamic according to which preferences might evolve in response to past recommendations. Characterizing other such dynamics (e.g., the formation of brand loyalty where the rewards of an arm increase with more pulls) in bandits setups is of future interest.

## Acknowledgement

LL is generously supported by an Open Philanthropy AI Fellowship. This research has been partly funded by the Center for Marketing Information and Technology at the Tepper School of Business. The authors would like to thank David Childers, Biswajit Paria, Eyan P. Noronha, Sai Sandeep and Max Simchowitz for very helpful discussions, and Stephen Tu for his insightful suggestions on system identification of affine dynamical systems.

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
