## Contents (Appendix)

# A  Integer Linear Programming Formulation

The bilinear integer program of (4) admits the following equivalent linear integer programming formulation:

$$\max_{u_{k,t}, z_{k,t,i}} \sum_{k \in [K]} \sum_{t \in [T]} b_k u_{k,t} - \lambda_k \sum_{i=0}^{t-1} \gamma_k^{t-i} z_{k,t,i}$$

$$\text{s.t.} \sum_{k \in [K]} u_{k,t} = 1, \qquad \forall t \in [T],$$

$$z_{k,t,i} \leq u_{k,i}, \ z_{k,t,i} \leq u_{k,t}, \ u_{k,i} + u_{k,t} - 1 \leq z_{k,t,i}, \qquad \forall k \in [K], t \in [T], i \in \{0, \ldots, t-1\},$$

$$u_{k,t} \in \{0, 1\}, \ u_{k,0} = 0, \qquad \forall k \in [K], t \in [T],$$

$$z_{k,t,i} \in \{0, 1\}, \qquad \forall k \in [K], t \in [T], i \in \{0, \ldots, t-1\}.$$

# B Proofs and Discussion of Section 4

## B.1 Proof of Lemma 1

*Proof.* When the expected rewards of all arms are the same, we know that the arm with the lowest index will be chosen and thus the first $K$ pulls will be $\pi_1 = 1, \ldots, \pi_K = K$. We will complete the proof through induction. Suppose that the greedy pull sequence is periodic with $\pi_1 = 1, \ldots, \pi_K = K$ and $\pi_{t+K} = \pi_t$ until time $h > K$. We define $k'$ to be $h \bmod K$ and $n$ to be $(h - k')/K$. We will show that $\pi_{h+1} = 1$ if $\pi_h = K$ and $\pi_{h+1} = \pi_h + 1$ otherwise. When $k' = 0$ (i.e., $\pi_h = K$), all arms have been pulled exactly $n$ times as of time $h$. By the induction assumption, we know that $u_{1,1:h-K} = u_{2,2:h-K+1} = \ldots = u_{K,K:h}$, which implies that last time when each arm is pulled, all of them have the same expected rewards, i.e.,

$$\mu_{1,h-K+1}(u_{1,0:h-K}) = \mu_{2,h-K+2}(u_{2,0:h-K+1}) = \cdots = \mu_{K,h}(u_{K,0:h-1}).$$

Moreover, $u_{1,h-K+1:h} = (1, \underbrace{0, \cdots 0}_{K \text{ times}})$, $u_{2,h-K+1:h} = (1, \underbrace{0, \cdots 0}_{K\text{-1 times}})$, $\cdots$, $u_{K,h:h} = (1)$.

Therefore, by (3), at time $h + 1$, arm 1 has the highest expected reward and will be chosen. In the case where $k' > 0$ (i.e., $\pi_h = k'$), we let $h' := h - k'$. We have that $\mu_{1,h'-K+1}(u_{1,0:h'-K}) = \ldots = \mu_{K,h}(u_{K,0:h'-1})$ and $s = s_{1,h'-K+1}(u_{1,0:h'-K}) = \ldots = s_{K,h'}(u_{K,0:h'-1}) \leq \frac{\gamma^K}{1-\gamma^K}$. Then, at time $h + 1$, the satiation level for the arms will be $s_{k,h+1}(u_{k,0:h}) = \gamma^{k'-k+1}\left(1 + \gamma^K s\right)$ for all $k \leq k'$ and $s_{k,h+1}(u_{k,0:h}) = \gamma^{K-k+k'+1}s$ for all $k > k'$. Thus, the arm with the lowest satiation level will be $\pi_{h+1} = k' + 1 = \pi_h + 1$, since $s_{k'+1,h+1}(u_{k'+1,0:h}) < s_{1,h+1}(u_{1,0:h})$. Consequently, the greedy policy will select arm $\pi_h + 1$ at time $h + 1$. $\qquad\square$

## B.2 Proof of Theorem 1

*Proof.* First, when $T \leq K$, greedy policy is optimal since its cumulative expected reward is $Tb$. So, we consider the case of $T > K$. Assume for contradiction that there exists another policy $\pi_{1:T}^o$ that is optimal and is not greedy, i.e., $\exists t \in [T], \pi_t^o \notin \arg\max_{k \in [K]} b - \lambda s_{k,t}^o$ where $s_{k,t}^o$ denotes the satiation level of arm $k$ at time $t$ under the policy $\pi_{1:T}^o$. We will construct a new policy $\pi_{1:T}^n$ that obtains a higher cumulative expected reward than $\pi_{1:T}^o$. Throughout the proof, we use $s_{k,t}^n$ to denote the satiation levels for the new policy.

We first note two illustrative facts to give the intuition of the proof.

*Fact 1:* Any policy $\pi_{1:T}^o$ that does not pick the arm with the lowest satiation level (i.e., highest expected reward) at the last time step $T$ is not optimal.
*Proof of Fact 1:* In this case, the policy $\pi_{1:T}^n = (\pi_1^o, \ldots, \pi_{T-1}^o, \pi_T)$ where $\pi_T \in \arg\max_{k \in [K]} b - \lambda s_{k,T}^o$ will obtain a higher cumulative expected reward.

*Fact 2:* If a policy $\pi_{1:T}^o$ picks the lowest satiation level for the final pull $\pi_T^o$ but does not pick the arm with the lowest satiation level at time $T - 1$, we claim that $\pi_{1:T}^n = (\pi_1^o, \ldots, \pi_{T-2}^o, \pi_T^o, \pi_{T-1}^o) \neq \pi_{1:T}^o$ obtains a higher cumulative expected reward.
*Proof of Fact 2:* First, note that $\pi_{T-1}^o \neq \pi_T^o$ because otherwise $\pi_{T-1}^o$ is the arm with the lowest satiation level at $T - 1$. Moreover, at time $T - 1$, $\pi_T^o \in \arg\min_k s_{k,T-1}^o$ has the smallest satiation, since if not, then there exists another arm $k \neq \pi_T^o$ and $k \neq \pi_{T-1}^o$ that has a smaller satiation level than $\pi_T^o$ at time $T - 1$. In that case, $\pi_T^o$ will not be the arm with the lowest satiation at time $T$, which is a contradiction. Then, we deduce $s_{\pi_{T-1}^o,T-1}^o > s_{\pi_T^o,T-1}^o$. Combining this with $\pi_{T-1}^o \neq \pi_T^o$, we arrive at

$$G_T(\pi_{1:T}^n) - G_T(\pi_{1:T}^o) = \lambda(1-\gamma)\left(s_{\pi_{T-1}^o,T-1}^o - s_{\pi_T^o,T-1}^o\right) > 0.$$

For the general case, given any policy $\pi_{1:T}^o$ that is not a greedy policy, we construct the new policy $\pi_{1:T}^n$ that has a higher cumulative expected reward through the following procedure:

1. Find $t^* \in [T]$ such that for all $t > t^*$, $\pi_t^o \in \arg\max_{k \in [K]} b - \lambda s_{k,t}^o$ and $\pi_{t^*}^o \notin \arg\max_{k \in [K]} b - \lambda s_{k,t^*}^o$. Further, we know that $\pi_{t^*+1}^o \in \arg\max_{k \in [K]} b - \lambda s_{k,t^*}^o$, using the same reasoning as the above example, i.e., otherwise $\pi_{t^*+1}^o \notin \arg\max_{k \in [K]} b - \lambda s_{k,t^*+1}^o$. To ease the notation, we use $k_1$ to denote $\pi_{t^*}^o$ and $k_2$ to denote $\pi_{t^*+1}^o$.

2. For the new policy, we choose $\pi^n_{1:t^*+1} = (\pi^o_1, \ldots, \pi^o_{t^*-1}, k_2, k_1)$. Let $A^o_{t_1,t_2}$ denote the set $\{t' : t^* + 2 \leq t' \leq t_2, \pi^o_{t'} = \pi^o_{t_1}\}$. $A^o_{t_1,t_2}$ contains a set of time indices in between $t^* + 2$ and $t_2$ when arm $\pi^o_{t_1}$ is played under policy $\pi^o_{1:T}$. We construct the following three sets $T_A := \{t : t^* + 2 \leq t \leq T, |A^o_{t^*,t}| < |A^o_{t^*+1,t}|\}$, $T_B := \{t : t^* + 2 \leq t \leq T, |A^o_{t^*,t}| > |A^o_{t^*+1,t}|\}$ and $T_C := \{t : t^* + 2 \leq t \leq T, |A^o_{t^*,t}| = |A^o_{t^*+1,t}|\}$. For time $t \geq t^* + 2$, we consider the following three cases:

Case I. $T_B = \varnothing$, which means that at any time $t$ in between $t^*+2$ and $T$, arm $k_1$ is played more than arm $k_2$ from $t^* + 2$ to $t$. In this case, the new policy follows $\pi^n_{t^*+2:T} = \pi^o_{t^*+2:T}$.

Case II. $T_A = \varnothing$, which means that at any time $t$ in between $t^*+2$ and $T$, arm $k_2$ is played more than arm $k_1$ from $t^* + 2$ to $t$. In this case, the new policy satisfies: for all $t \geq t^* + 2$, 1) $\pi^n_t = \pi^o_t$ if $\pi^o_t \neq k_1$ and $\pi^o_t \neq k_2$; 2) $\pi^n_t = k_2$ if $\pi^o_t = k_1$; and 3) $\pi^n_t = k_1$ if $\pi^o_t = k_2$.

Case III. $T_A \neq \varnothing$ and $T_B \neq \varnothing$. Then, starting from $t^* + 2$, if $t \in T_A$, $\pi^n_t$ follows the new policy construction in Case I, i.e., $\pi^n_t = \pi^o_t$. If $t \in T_B$, $\pi^n_t$ follows the new policy construction in Case II. Finally, for all $t \in T_C$, define $t'_{A,t} = \max_{\substack{t' \in T_A \\ t' < t}} t'$ and $t'_{B,t} = \max_{\substack{t' \in T_B \\ t' < t}} t'$.

If $t'_{A,t} > t'_{B,t}$, then $\pi^n_t$ follows the new policy construction as Case I. If $t'_{A,t} < t'_{B,t}$, $\pi^n_t$ follows the new policy construction as Case II. We note that $t'_{A,t} \neq t'_{B,t}$ since $T_A \cap T_B = \varnothing$.

When $T_A = \varnothing$ and $T_B = \varnothing$, we know that $k_1$ and $k_2$ are not played in $\pi^o_{t^*+2:T}$. In this case, the new policy construction can follow either Case I or Case II. To complete the proof, we state some facts first:

- From $t^*$, the expected rewards collected by the policies $\pi^o_{1:T}$ and $\pi^n_{1:T}$ only differ at times when arm $k_1$ or arm $k_2$ is played.

- $\pi^n_{1:t^*+1}$ obtains a higher cumulative expected reward than $\pi^o_{1:t^*+1}$.

- At time $t^* + 2$, the new policy follows that $s^n_{k_1,t^*+2} = \gamma + \gamma^2 s^o_{k_1,t^*}$ and $s^n_{k_2,t^*+2} = \gamma^2 + \gamma^2 s^o_{k_2,t^*}$. On the other hand, the old policy has $s^o_{k_1,t^*+2} = \gamma^2 + \gamma^2 s^o_{k_1,t^*}$ and $s^o_{k_2,t^*+2} = \gamma + \gamma^2 s^o_{k_2,t^*}$.

Let $N_{k_1} := \{t : t^* + 2 \leq t \leq T, \pi^o_t = k_1\}$ and $N_{k_2} := \{t : t^* + 2 \leq t \leq T, \pi^o_t = k_2\}$ denote the sets of time steps when $k_1$ and $k_2$ are played in $\pi^o_{1:T}$. For a given satiation level $x$ at time $t'$ together with the time steps the arm is pulled $N_k$, we have that at time $t \geq t'$, the arm has satiation level $g_{N_k}(x, t, t') = \gamma^{t-t'} x + \sum_{N_{k,i} < t} \gamma^{t-N_{k,i}}$ where $N_{k,i}$ is the $i$-th smallest element in $N_k$.

In Case I, the difference of the cumulative expected rewards between the two policies satisfies:

$$G_T(\pi^n_{1:T}) - G_T(\pi^o_{1:T}) > \sum_{i=1}^{|N_{k_2}|} -\lambda g_{N_{k_2}}(s^n_{k_2,t^*+2}, N_{k_2,i}, t^* + 2) + \lambda g_{N_{k_2}}(s^o_{k_2,t^*+2}, N_{k_2,i}, t^* + 2)$$

$$+ \sum_{j=1}^{|N_{k_1}|} -\lambda g_{N_{k_1}}(s^n_{k_1,t^*+2}, N_{k_1,j}, t^* + 2) + \lambda g_{N_{k_1}}(s^o_{k_1,t^*+2}, N_{k_1,j}, t^* + 2)$$

$$= \lambda\left(s^o_{k_2,t^*+2} - s^n_{k_2,t^*+2}\right) \sum_{i=1}^{|N_{k_2}|} \gamma^{N_{k_2,i}-(t^*+2)} + \lambda\left(s^o_{k_1,t^*+2} - s^n_{k_1,t^*+2}\right) \sum_{j=1}^{|N_{k_1}|} \gamma^{N_{k_1,j}-(t^*+2)} > 0,$$

where we have used the fact that $s^o_{k_2,t^*+2} - s^n_{k_2,t^*+2} = -\left(s^o_{k_1,t^*+2} - s^n_{k_1,t^*+2}\right) > 0$, $|N_{k_2}| \geq |N_{k_1}|$ and for all $j \in [|N_{k_1}|]$, $N_{k_2,j} < N_{k_1,j}$. In Case II, similarly, we have that

$$G_T(\pi^n_{1:T}) - G_T(\pi^o_{1:T}) > \sum_{j=1}^{|N_{k_1}|} -\lambda g_{N_{k_1}}(s^n_{k_2,t^*+2}, N_{k_1,j}, t^* + 2) + \lambda g_{N_{k_1}}(s^o_{k_1,t^*+2}, N_{k_1,j}, t^* + 2)$$

$$+ \sum_{i=1}^{|N_{k_2}|} -\lambda g_{N_{k_2}}(s^n_{k_1,t^*+2}, N_{k_2,i}, t^* + 2) + \lambda g_{N_{k_2}}(s^o_{k_2,t^*+2}, N_{k_2,i}, t^* + 2)$$

$$= \lambda \left(s^o_{k_1,t^*+2} - s^n_{k_2,t^*+2}\right) \sum_{j=1}^{|N_{k_1}|} \gamma^{N_{k_1,j}-(t^*+2)} + \lambda \left(s^o_{k_2,t^*+2} - s^n_{k_1,t^*+2}\right) \sum_{i=1}^{|N_{k_2}|} \gamma^{N_{k_2,i}-(t^*+2)} > 0,$$

since $s^o_{k_1,t^*+2} - s^n_{k_2,t^*+2} = -\left(s^o_{k_2,t^*+2} - s^n_{k_1,t^*+2}\right) > 0$, $|N_{k_2}| \le |N_{k_1}|$ and for all $i \in [|N_{k_2}|]$, $N_{k_1,i} < N_{k_2,i}$.

Finally, for Case III, the new policy construction is a mix of Case I and Case II. We represent the time interval $[t^*+2, T]$ to be $[t^*+2, T] = [t_{i_1,s_1}, t_{i_1,e_1}] \cup [t_{i_2,s_2}, t_{i_2,e_2}] \cup \cdots \cup [t_{i_M,s_M}, t_{i_M,e_M}]$ where $t^*+2 = t_{i_1,s_1} \le \ldots \le t_{i_M,s_M} = T$, $\cap_{m=1}^{M}[t_{i_m,s_m}, t_{i_m,e_m}] = \varnothing$ and $M-1$ is the number of new policy construction switches happen in between $t^*+2$ and $T$. We say that a new policy construction switch happens at time $t$ if the policy construction follows Case I at time $t-1$ but follows Case II at time $t$ or vice versa. Each $i_m \ne i_{m-1}$ can take values I or II, representing which policy construction rule is used between the time period $t_{i_m,s_m}$ and $t_{i_m,e_m}$. For any time index set $V$, we use the notation $V[t_{i_m,s_m}, t_{i_m,e_m}] := \{t \in V : t_{i_m,s_m} \le t \le t_{i_m,e_m}\}$.

We notice that at any switching time $t_{i_m,s_m}$, the number of previous pulls of arm $k_1$ and $k_2$ from time $t_{i_{m-1},s_{m-1}}$ to $t_{i_{m-1},e_{m-1}}$ are equivalent, which is denoted by $l_m = |N_{k_1}[t_{i_m,s_m}, t_{i_m,e_m}]| = |N_{k_2}[t_{i_m,s_m}, t_{i_m,e_m}]|$ for all $m < M$. From our analysis of Case I and Case II, we know that to show that $\pi^n_{1:T}$ obtains a higher cumulative expected reward, it suffices to prove: for all $m < M$ such that

$$s^o_{k_2,t_{i_m,s_m}} - s^n_{k_2,t_{i_m,s_m}} = -\left(s^o_{k_1,t_{i_m,s_m}} - s^n_{k_1,t_{i_m,s_m}}\right) > 0,$$

$$s^o_{k_1,t_{i_m,s_m}} - s^n_{k_2,t_{i_m,s_m}} = -\left(s^o_{k_2,t_{i_m,s_m}} - s^n_{k_1,t_{i_m,s_m}}\right) > 0,$$

we have

$$s^o_{k_2,t_{i_{m+1},s_{m+1}}} - s^n_{k_2,t_{i_{m+1},s_{m+1}}} = -\left(s^o_{k_1,t_{i_{m+1},s_{m+1}}} - s^n_{k_1,t_{i_{m+1},s_{m+1}}}\right) > 0,$$

$$s^o_{k_1,t_{i_{m+1},s_{m+1}}} - s^n_{k_2,t_{i_{m+1},s_{m+1}}} = -\left(s^o_{k_2,t_{i_{m+1},s_{m+1}}} - s^n_{k_1,t_{i_{m+1},s_{m+1}}}\right) > 0.$$

We will establish these facts in Lemma 3. Finally, we note that the above required conditions are held at time $t_{i_1,s_1} = t^*+2$. □

**Lemma 3.** *Let $N_k[t_s, t_e]$ denote the set of time steps when arm $k$ is pulled in between (and including) time $t_s$ and $t_e$ under policy $\pi^o_{1:T}$. Let $s^o_{k,t}$ and $s^n_{k,t}$ represent the satiation level of arm $k$ at time $t$ when following the policy $\pi^o_{1:T}$ and $\pi^n_{1:T}$, respectively. For two different arms $k_1$ and $k_2$, suppose that at time $t_s$ we have*

$$s^o_{k_2,t_s} - s^n_{k_2,t_s} = -\left(s^o_{k_1,t_s} - s^n_{k_1,t_s}\right) > 0,$$
$$s^o_{k_1,t_s} - s^n_{k_2,t_s} = -\left(s^o_{k_2,t_s} - s^n_{k_1,t_s}\right) > 0.$$

*Further, suppose that from time $t_s$ to $t_e$, $\pi^n_{1:T}$ follows either Case I (or Case II) of new policy construction (see proof of Theorem 1 for their definitions); and at time $t'_s = t_e + 1$, the new policy construction for $\pi^n_{1:T}$ has switched to Case II (or Case I if Case II is used from $t_s$ to $t_e$). Then at time $t'_s$, we have that*

$$s^o_{k_2,t'_s} - s^n_{k_2,t'_s} = -\left(s^o_{k_1,t'_s} - s^n_{k_1,t'_s}\right) > 0,$$

$$s^o_{k_1,t'_s} - s^n_{k_2,t'_s} = -\left(s^o_{k_2,t'_s} - s^n_{k_1,t'_s}\right) > 0.$$

*Proof of Lemma 3.* Following the definition in the proof of Theorem 1, given that at time $t_s$, arm $k$ has satiation $s$, let $g_{N_k[t_s, t_e]}(s, t'_s, t_s)$ denote the satiation level of arm $k$ at time $t'_s$ after being pulled at the time steps in the set $N_k[t_s, t_e]$. Let $N_{k,i}[t_s, t_e]$ be the $i$-th smallest element in the set $N_k[t_s, t_e]$. From the definition of the new policy construction given in the proof of Theorem 1, we also know that (1) $N := |N_{k_1}[t_s, t_e]| = |N_{k_2}[t_s, t_e]|$; (2) if Case I is applied in between $t_s$ and $t_e$, we have that for all $i \in [N]$, $N_{k_2,i}[t_s, t_e] < N_{k_1,i}[t_s, t_e]$; and (3) if Case II is applied in between $t_s$ and $t_e$, we have that for all $i \in [N]$, $N_{k_2,i}[t_s, t_e] > N_{k_1,i}[t_s, t_e]$.

We first consider the setting when Case I new policy construction is applied, then at time $t'_s$, we can show that

$$s^o_{k_1,t'_s} - s^n_{k_2,t'_s} = g_{N_{k_1}[t_s,t_e]}\left(s^o_{k_1,t_s},t'_s,t_s\right) - g_{N_{k_2}[t_s,t_e]}\left(s^n_{k_2,t_s},t'_s,t_s\right)$$

$$= \gamma^{t'_s-t_s}\left(s^o_{k_1,t_s} - s^n_{k_2,t_s}\right) + \sum_{i=1}^{l}\gamma^{t'_s-N_{k_1,i}[t_s,t_e]} - \gamma^{t'_s-N_{k_2,i}[t_s,t_e]}$$

$$= \gamma^{t'_s-t_s}\left(s^n_{k_1,t_s} - s^o_{k_2,t_s}\right) + \sum_{i=1}^{l}\gamma^{t'_s-N_{k_1,i}[t_s,t_e]} - \gamma^{t'_s-N_{k_2,i}[t_s,t_e]}$$

$$= s^n_{k_1,t'_s} - s^o_{k_2,t'_s} > 0,$$

where the last inequality has used the fact that when we use Case I construction, we have $N_{k_2,i}[t_s,t_e] < N_{k_1,i}[t_s,t_e]$. Meanwhile, we also have that

$$s^o_{k_2,t'_s} - s^n_{k_2,t'_s} = g_{N_{k_2}[t_s,t_e]}\left(s^o_{k_2,t_s},t'_s,t_s\right) - g_{N_{k_2}[t_s,t_e]}\left(s^n_{k_2,t_s},t'_s,t_s\right)$$

$$= \gamma^{t'_s-t_s}\left(s^o_{k_2,t_s} - s^n_{k_2,t_s}\right) = -\gamma^{t'_s-t_s}\left(s^o_{k_1,t_s} - s^n_{k_1,t_s}\right)$$

$$= -\left(s^o_{k_1,t'_s} - s^n_{k_1,t'_s}\right) > 0.$$

When Case II new policy construction is applied, then at time $t'_s$, we get

$$s^o_{k_1,t'_s} - s^n_{k_2,t'_s} = g_{N_{k_1}[t_s,t_e]}\left(s^o_{k_1,t_s},t'_s,t_s\right) - g_{N_{k_1}[t_s,t_e]}\left(s^n_{k_2,t_s},t'_s,t_s\right)$$

$$= \gamma^{t'_s-t_s}\left(s^o_{k_1,t_s} - s^n_{k_2,t_s}\right) = -\gamma^{t'_s-t_s}\left(s^o_{k_2,t_s} - s^n_{k_1,t_s}\right)$$

$$= -\left(s^o_{k_2,t'_s} - s^n_{k_1,t'_s}\right) > 0,$$

since $s^o_{k_1,t_s} - s^n_{k_2,t_s} > 0$. On the other hand, we have that

$$s^o_{k_2,t'_s} - s^n_{k_2,t'_s} = g_{N_{k_2}[t_s,t_e]}\left(s^o_{k_2,t_s},t'_s,t_s\right) - g_{N_{k_1}[t_s,t_e]}\left(s^n_{k_2,t_s},t'_s,t_s\right)$$

$$= \gamma^{t'_s-t_s}\left(s^o_{k_2,t_s} - s^n_{k_2,t_s}\right) + \sum_{i=1}^{l}\gamma^{t'_s-N_{k_2,i}[t_s,t_e]} - \gamma^{t'_s-N_{k_1,i}[t_s,t_e]}$$

$$= \gamma^{t'_s-t_s}\left(s^n_{k_1,t_s} - s^o_{k_1,t_s}\right) + \sum_{i=1}^{l}\gamma^{t'_s-N_{k_2,i}[t_s,t_e]} - \gamma^{t'_s-N_{k_1,i}[t_s,t_e]}$$

$$= s^n_{k_1,t'_s} - s^o_{k_1,t'_s} > 0,$$

where the last inequality is true because when Case II new policy construction is applied, we have $N_{k_1,i}[t_s,t_e] < N_{k_2,i}[t_s,t_e]$. $\square$

### B.3 Proof of Proposition 2

*Proof.* If $T \leq K$, a Max K-Cut of $\mathcal{K}_T$ is $\forall k \in [T], P_k = \{k\}$, which is the same as an optimal solution to (4). Let $\mathbf{1}\{\cdot\}$ denote the indicator function. When $T > K$, the integer program in (4) is equivalent to

$$\max_{\substack{u_{k,t}\in\{0,1\}:\\ \forall t\in[T],\sum_k u_{k,t}=1}} \sum_{k=1}^{K}bu_{k,1} + \sum_{k=1}^{K}\sum_{t=2}^{T}\left(bu_{k,t} - \lambda\sum_{i=1}^{t-1}\gamma^{t-i}u_{k,i}u_{k,t}\right)$$

$$= \max_{\substack{P_1,\ldots,P_K\subseteq[T]:\\ \cup_k P_k=[T],\\ \forall k\neq k',P_k\cap P_{k'}=\varnothing}} \sum_{k=1}^{K}b\mathbf{1}\{1\in P_k\} + \sum_{k=1}^{K}\sum_{t=2}^{T}\left(b\mathbf{1}\{t\in P_k\} - \lambda\sum_{i=1}^{t-1}\gamma^{t-i}\mathbf{1}\{i\in P_k\}\mathbf{1}\{t\in P_k\}\right)$$

$$= \max_{\substack{P_1,\ldots,P_K\subseteq[T]:\\ \cup_k P_k=[T],\\ \forall k\neq k',P_k\cap P_{k'}=\varnothing}} Tb - \sum_{k=1}^{K}\sum_{\substack{t,i\in P_k:\\ i<t}}\lambda\gamma^{t-i}$$

$$=Tb - \sum_{t=2}^{T}\sum_{i=1}^{t-1}\lambda\gamma^{t-i} + \max_{\substack{P_1,\dots,P_K\subseteq[T]:\\ \cup_k P_k=[T],\\ \forall k\neq k',P_k\cap P_{k'}=\varnothing}} \sum_{k=1}^{K-1}\sum_{k'=k+1}^{K}\sum_{\substack{t\in P_k,\\ i\in P_{k'}:\\ i<t}}\lambda\gamma^{t-i},$$

where the second equality uses the fact $\sum_{k=1}^{K}\mathbf{1}\{t\in P_k\}=1$ for all $t\in[T]$ and the third equality is true because for any $P_1,\dots P_K$ such that $\forall k\neq k'$, $P_k\cap P_{k'}=\varnothing$ and $\cup_k P_k=[T]$, we have

$$\text{Total Edge Weights of }\mathcal{K}_T = \sum_{t=2}^{T}\sum_{i=1}^{t-1}e(t,i) = \sum_{\substack{t,i\in[T]:i<t,\\ \exists k\in[K],i,t\in P_k}}e(t,i) \quad + \sum_{\substack{t,i\in[T]:i<t,\\ \forall k\in[K],i,t\notin P_k}}e(t,i).$$

$\square$

## B.4 Proof of Theorem 2

*Proof.* Given $\pi_{1:T}^*$ and $\pi_{1:T}^w$, define a set of new policies $\{\tilde{\pi}_{1:T}^i\}_{i=1}^{l-1}$ such that for all $i$, $\tilde{\pi}_{1:T}^i = (\pi_{1:iw}^w,\pi_{iw+1:T}^*)$. Based on this, we have the following decomposition

$$G_T(\pi_{1:T}^*)-G_T(\pi_{1:T}^w) = \underbrace{G_T(\pi_{1:T}^*) - G_T(\tilde{\pi}_{1:T}^1)}_{A_0} + \left(\sum_{i=1}^{l-2}\underbrace{G_T(\tilde{\pi}_{1:T}^i) - G_T(\tilde{\pi}_{1:T}^{i+1})}_{A_i}\right) + \underbrace{G_T(\tilde{\pi}_{1:T}^{l-1}) - G_T(\pi_{1:T}^w)}_{A_{l-1}}.$$

To distinguish the past pull sequences of each arm under different policies, we use the following notations: $\mu_{k,t}(u_{k,0:t-1};\pi')$ gives the expected reward of arm $k$ at time $t$ by following pull sequence $\pi'_{1:t-1}$. By the definition of $\pi_{1:T}^w$, we have that

$$A_0 = \sum_{t=1}^{w}\mu_{\pi_t^*,t}(u_{\pi_t^*,0:t-1};\pi^*) - \mu_{\pi_t^w,t}(u_{\pi_t^w,0:t-1};\pi^w) + \sum_{t=w+1}^{T}\mu_{\pi_t^*,t}(u_{\pi_t^*,0:t-1};\pi^*) - \mu_{\pi_t^*,t}(u_{\pi_t^*,0:t-1};\tilde{\pi}^1)$$

$$\leq \sum_{t=w+1}^{T}\mu_{\pi_t^*,t}(u_{\pi_t^*,0:t-1};\pi^*) - \mu_{\pi_t^*,t}(u_{\pi_t^*,0:t-1};\tilde{\pi}^1),$$

where the inequality follows from the fact that $\pi_{1:w}^w$ is optimal for (4) when $T=w$. Similarly, we obtain that for all $i\in[l-2]$,

$$A_i = \sum_{t=1}^{iw}\underbrace{\mu_{\pi_t^w,t}(u_{\pi_t^w,0:t-1};\pi^w) - \mu_{\pi_t^w,t}(u_{\pi_t^w,0:t-1};\pi^w)}_{=0} + \sum_{t=iw+1}^{(i+1)w}\underbrace{\mu_{\pi_t^*,t}(u_{\pi_t^w,0:t-1};\tilde{\pi}^i) - \mu_{\pi_t^w,t}(u_{\pi_t^w,0:t-1};\pi^w)}_{\leq 0}$$

$$+ \sum_{t=(i+1)w+1}^{T}\mu_{\pi_t^*,t}(u_{\pi_t^*,0:t-1};\tilde{\pi}^i) - \mu_{\pi_t^*,t}(u_{\pi_t^*,0:t-1};\tilde{\pi}^{i+1})$$

$$\leq \sum_{t=(i+1)w+1}^{T}\mu_{\pi_t^*,t}(u_{\pi_t^*,0:t-1};\tilde{\pi}^i) - \mu_{\pi_t^*,t}(u_{\pi_t^*,0:t-1};\tilde{\pi}^{i+1}).$$

Finally, we have $A_{l-1} = \sum_{t=(l-1)w+1}^{T}\mu_{\pi_t^*,t}(u_{\pi_t^*,0:t-1};\tilde{\pi}^{l-1}) - \mu_{\pi_t^w,t}(u_{\pi_t^w,0:t-1};\pi^w) \leq 0$. To complete the proof, it suffices to use the fact that for all $i\in\{1,\dots,l-1\}$,

$$\max_{\substack{\pi'_{1:T},\pi_{1:T}:\\ \pi'_{iw+1:T}=\pi_{iw+1:T}}}\sum_{t=iw+1}^{T}\mu_{\pi_t,t}(u_{\pi_t,0:t-1};\pi) - \mu_{\pi_t,t}(u_{\pi_t,0:t-1};\pi') \leq \sum_{t=0}^{T-iw-1}\overline{\lambda}\overline{\gamma}^t\frac{\overline{\gamma}}{1-\overline{\gamma}} \leq \frac{\overline{\lambda}\overline{\gamma}(1-\overline{\gamma}^{T-iw})}{(1-\overline{\gamma})^2}$$

$$\leq \frac{\overline{\lambda}\overline{\gamma}(1-\overline{\gamma}^{T-w})}{(1-\overline{\gamma})^2},$$

where the first inequality holds because for any arm, the maximum satiation level discrepancy under two pull sequences (after $iw$ time steps) is $\overline{\gamma}/(1-\overline{\gamma})$ and from time $iw+1$ till time $T$, the objective will be maximized when the arm with the maximum satiation discrepancy is played all the time. $\square$

# C  More Discussion on Learning with Unknown Dynamics

As we have noted in Section 5, when the learner makes a decision on which arm to pull, the learner does not observe the hidden satiation level the user has for the arms. The POMDP the learner faces can be cast as a fully observable MDP (Appendix C.1) where the estimated reward model (Appendix C.2) can be used for planning (Appendix C.3). In addition to policies that are time-dependent (actions taken by time-dependent policies only depend on the time steps at which they are taken) considered in Section 6, we also consider state-dependent policies where the states are continuous.

## C.1  MDP Setup

We begin with describing the full MDP setup of rebounding bandits, including the state representation and reward function defined in Section 5.1. Following [32], at any time $t \in [T]$, we define our state vector to be $x_t = (x_{1,t}, n_{1,t}, x_{2,t}, n_{2,t}, \ldots, x_{K,t}, n_{K,t})$, where $n_{k,t} \in \mathbb{N}$ is the number of steps since arm $k$ is last selected and $x_{k,t}$ is the satiation influenceas of the most recent pull of arm $k$. Since the most recent pull happens at $t - n_{k,t}$, we have $x_{k,t} = b_k - \mu_{k,t-n_{k,t}} = \lambda_k s_{k,t-n_{k,t}}$. We note that $b_k$ can be obtained when arm $k$ is pulled for the first time since the satiation effect is $0$ if an arm has not been pulled before. The initial state is $x_{\text{init}} = (0, \ldots, 0)$. Transitions between two states $x_t$ and $x_{t+1}$ are defined as follows: If arm $k$ is chosen at time $t$, i.e., $\pi_t = k$, and reward $\mu_{k,t}$ is obtained, then the next state $x_{t+1}$ will be:

A.1  For the pulled arm $k$, $n_{k,t+1} = 1$ and $x_{k,t+1} = b_k - \mu_{k,t}$.

A.2  For other arms $k' \neq k$, $n_{k',t+1} = n_{k',t} + 1$ if $n_{k',t} \neq 0$ and $n_{k',t+1} = 0$ if $n_{k',t} = 0$. The satiation influence remains the same, i.e., $x_{k',t+1} = x_{k',t}$.

For all $x_t \in \mathcal{X}$ and $k \in [K]$, we have that $\mathbb{E}[x_{k,t}] \leq \overline{\lambda}\overline{\gamma}/(1 - \overline{\gamma})$ and $\text{Var}[x_{k,t}] \leq \overline{\lambda}^2 \sigma_z^2/(1 - \overline{\gamma}^2)$. Hence, for any $\delta \in (0,1)$, $\mathbb{P}\left(\max_{k,t} |x_{k,t}| \geq B(\delta)\right) \leq \delta$, where

$$B(\delta) := \frac{\overline{\lambda}\overline{\gamma}}{1 - \overline{\gamma}} + \overline{\lambda}\sigma_z \sqrt{\frac{2\log(2KT/\delta)}{1 - \overline{\gamma}^2}}. \tag{10}$$

The MDP the learner faces can be described as a tuple $\mathcal{M} := \langle x_{\text{init}}, [K], \{\gamma_k, \lambda_k, b_k\}_{k=1}^K, T \rangle$ of the initial state $x_{\text{init}}$, actions (arms) $[K]$, the horizon $T$ and parameters $\{\gamma_k, \lambda_k, b_k\}_{k=1}^K$. Let $\Delta(\cdot)$ denote the probability simplex. Given $\{\gamma_k, \lambda_k, b_k\}_{k=1}^K$, the expected reward $r : \mathcal{X} \times [K] \to \mathbb{R}$ and transition functions $p : \mathcal{X} \times [K] \times [T] \to \Delta(\mathcal{X})$ are defined as follows:

1. $r : \mathcal{X} \times [K] \to \mathbb{R}$ gives the *expected* reward of pulling arm $k$ conditioned on $x_t$, i.e., $r(x_t, k) = \mathbb{E}[\mu_{k,t}|x_t]$.[2] If $n_{k,t} = 0$, then $r(x_t, k) = b_k$. If $n_{k,t} \geq 1$, $r(x_t, k) = b_k - \gamma_k^{n_{k,t}} x_{k,t} - \lambda_k \gamma_k^{n_{k,t}}$.

2. When pulling arm $k$ at time $t$ and state $x_t$, $p\left(x_{t+1}|x_t, k, t\right) = 0$ if $x_{t+1}$ does not satisfy A.1 or A.2. When $x_{t+1}$ fulfills both A.1 and A.2, we consider two cases of $x_t$. If $n_{k,t} \neq 0$, then the transition function $p\left(x_{t+1}|x_t, k, t\right)$ is given by the Gaussian density with mean $\gamma_k^{n_{k,t}}(x_{k,t} + \lambda_k)$ and variance $\lambda_k^2 \sigma_z^2 \sum_{i=0}^{n_{k,t}-1} \gamma_k^{2i}$, as illustrated in (11). If $n_{k,t} = 0$, then $p(x_{t+1}|x_t, k, t) = 1$ since for the first pull of arm $k$, the obtained reward $\mu_{k,t} = b_k$.

At time $t$, the learner follows an action $\pi_t : \mathcal{X} \to [K]$ that depends on the state. We use $V_{t,\mathcal{M}}^\pi : \mathcal{X} \to \mathbb{R}$ to denote the value function of policy $\pi_{1:T}$ at time $t$ under MDP $\mathcal{M}$: $V_{t,\mathcal{M}}^\pi(x_t) = r(x_t, \pi_t(x_t)) + \mathbb{E}_{x_{t+1} \sim p(\cdot|x_t, \pi_t(x_t), t)}[V_{t+1,\mathcal{M}}^\pi(x_{t+1})]$ and $V_{T+1,\mathcal{M}}^\pi(x) = 0$ for all $x \in \mathcal{X}$. To restate our goal (2) in terms of the value function: for an MDP $\mathcal{M}$, we would like to find a policy $\pi_{1:T}$ that maximizes

$$V_{1,\mathcal{M}}^\pi(x_{\text{init}}) = \mathbb{E}\left[\sum_{t=1}^T r(x_t, \pi_t(x_t)) \middle| x_1 = x_{\text{init}}\right].$$

To simplify the notation, we use $\pi$ to refer to a policy $\pi_{1:T}$. Given an MDP $\mathcal{M}$, we denote its optimal policy by $\pi_{\mathcal{M}}^*$ and the value function for the optimal policy by $V_{t,\mathcal{M}}^*$, i.e., $V_{t,\mathcal{M}}^*(x) := V_{t,\mathcal{M}}^{\pi_{\mathcal{M}}^*}(x)$.

---

[2]By conditioning on $x_t$, we mean conditioning on the $\sigma$-algebra generated by past actions and observed rewards.

## C.2 Exploration and Estimation of the Reward Model

As we have discussed in § 6.1, based on our satiation and reward models, the satiation influence $x_{k,t}$ of arm $k$ forms a dynamical system where we only observe the value of the system when arm $k$ is pulled. When arm $k$ is pulled at time $t$ and $n_{k,t} \neq 0$, we observe the satiation influence $\lambda_k s_{k,t}$ which becomes the next state $x_{k,t+1}$, i.e.,

$$
\begin{aligned}
x_{k,t+1} = \lambda_k s_{k,t} &= \lambda_k \gamma_k^{n_{k,t}} s_{k,t-n_{k,t}} + \lambda_k \gamma_k^{n_{k,t}} + \lambda_k \sum_{i=0}^{n_{k,t}-1} \gamma_k^i z_{k,t-1-i} \\
&= \gamma_k^{n_{k,t}} x_{k,t+1-n_{k,t}} + \lambda_k \gamma_k^{n_{k,t}} + \lambda_k \sum_{i=0}^{n_{k,t}-1} \gamma_k^i z_{k,t-1-i}.
\end{aligned} \tag{11}
$$

We note that the current state $x_{k,t}$ equals to $x_{k,t+1-n_{k,t}}$ since $x_{k,t+1-n_{k,t}}$ is the last observed satiation influence for arm $k$ and $n_{k,t}$ is the number of steps since arm $k$ is last pulled.

**Exploration Settings**   Depending on the nature of the recommendation domain, we consider two types of exploration settings: one where the users only interact with the recommendation systems for a short time after they log in to the service (Appendix C.2.1) and the other where the users tend to interact with the system for a much longer time, e.g., automated music playlisting (Appendix C.2.2). In the first case, the learner collects multiple ($n$) short trajectories of user utilities, while in the second case, similar to § 6.2, the learner obtains a single trajectory of user utilities that has length $n$. In both settings, we obtain that under some mild conditions, the estimation errors of our estimators for $\gamma_k$ and $\lambda_k$ are $O(1/\sqrt{n})$.

**Exploration Strategies**   Generalizing from the case where arms are pulled repeatedly, we explore by pulling the same arm at a fixed interval $m$. In particular, when $m = 1$, the exploration strategy is the same as repeatedly pulling the same arm for multiple times, which is the exploration strategy used in § 6.1. When $m = K$, the exploration strategy is to pull the arms in a cyclic order. We present the estimator for $\gamma_k, \lambda_k$ using the dataset collected by this exploration strategy in both the multiple trajectory and single trajectory settings.

### C.2.1   Estimation using Multiple Trajectories

For each arm $k \in [K]$, we use $\mathcal{D}_k^{n,m}$ to denote a dataset containing $n$ trajectories of evenly spaced observed satiation influences that are collected by our exploration phase. The time interval between two pulls of an arm is denoted by $m$. Each trajectory is of length at least $T_{\min} + 1$ for $T_{\min} > 1$. For trajectory $i \in [n]$, the observed satiation influences are denoted by $\tilde{x}_{k,1}^{(i)}, \ldots, \tilde{x}_{k,T_{\min}+1}^{(i)}, \ldots$, where $\tilde{x}_{k,1}^{(i)} = 0$ is the initial satiation influence and the rest of the satiation influences $\tilde{x}_{k,j}^{(i)}$ ($j > 1$) is the difference between the first received reward, i.e., the base reward $b_k$, and the reward from the $j$-th pull of arm $k$. In other words, for $\tilde{x}_{k,j}^{(i)}, \tilde{x}_{k,j+1}^{(i)} \in \mathcal{D}_k^{n,m}$, it follows that

$$
\tilde{x}_{k,j+1}^{(i)} = a_k \tilde{x}_{k,j}^{(i)} + d_k + \tilde{z}_{k,j}^{(i)}, \tag{12}
$$

where $a_k = \gamma_k^m$, $d_k = \lambda_k \gamma_k^m$ and $\tilde{z}_{k,j}^{(i)}$ are the independent samples from $\mathcal{N}\left(0, \sigma_{z,k}^2\right)$ with $\sigma_{z,k}^2 = \lambda_k^2 \sigma_z^2 (1 - \gamma_k^{2m})/(1 - \gamma_k^2)$.

To estimate $d_k$, we use the estimator $\widehat{d}_k = \frac{1}{n} \sum_{i=1}^n \tilde{x}_{k,2}^{(i)} = d_k + \frac{1}{n} \sum_{i=1}^n \tilde{z}_{k,1}^{(i)}$. By the standard Gaussian tail bound, we obtain that for $\delta \in (0, 1)$, with probability $1 - \delta$,

$$
|\widehat{d}_k - d_k| \leq \sqrt{\frac{2\sigma_{z,k}^2 \log(2/\delta)}{n}} =: \epsilon_d(n, \delta, k). \tag{13}
$$

When estimating $a_k$, we first take the difference between the first $T_{\min} + 1$ entries of two trajectories $i$ and $2i$ for $i \in \lfloor n/2 \rfloor$ and obtain a new trajectory $\tilde{y}_{k,1}^{(i)}, \ldots, \tilde{y}_{k,T_{\min}+1}^{(i)}$ where $\tilde{y}_{k,j}^{(i)} = \tilde{x}_{k,j}^{(i)} - \tilde{x}_{k,j}^{(2i)}$ for $j \in [T_{\min} + 1]$. We note that the new trajectory forms a linear dynamical system without the bias term $d_k$, i.e.,

$$
\tilde{y}_{k,j+1}^{(i)} = a_k \tilde{y}_{k,j}^{(i)} + \tilde{w}_{k,j}^{(i)},
$$

where $\tilde{w}_{k,j}^{(i)}$ are samples from $\mathcal{N}(0, 2\sigma_{z,k}^2)$. We use the ordinary least squares estimator to estimate $a_k$:

$$\widehat{a}_k = \arg\min_a \sum_{i=1}^{\lfloor n/2 \rfloor} \left( \tilde{y}_{k,T_{\min}+1}^{(i)} - a\tilde{y}_{k,T_{\min}}^{(i)} \right)^2$$

$$= \frac{\sum_{i=1}^{\lfloor n/2 \rfloor} \tilde{y}_{k,T_{\min}}^{(i)} \tilde{y}_{k,T_{\min}+1}^{(i)}}{\sum_{i=1}^{\lfloor n/2 \rfloor} \left( \tilde{y}_{k,T_{\min}}^{(i)} \right)^2}. \tag{14}$$

**Theorem 6.** *[24, Theorem II.4] Fix $\delta \in (0,1)$. Given $n \geq 64 \log(2/\delta)$, with probability $1 - \delta$, we have that*

$$|\widehat{a}_k - a_k| \leq 4\sqrt{\frac{2\log(4/\delta)}{n \sum_{t=0}^{T_{\min}} a_k^{2t}}} =: \epsilon_a(n, \delta, k). \tag{15}$$

We notice that as the minimum length of the trajectory gets greater, the upper bound of the estimation error of $a_k$ gets smaller. Using our estimators for $a_k$ and $d_k$, we estimate $\gamma_k$ and $\lambda_k$ through $\widehat{\gamma}_k = |\widehat{a}_k|^{1/m}$ and $\widehat{\lambda}_k = |\widehat{d}_k/\widehat{a}_k|$.

**Corollary 7.** *Fix $\delta \in (0,1)$. Suppose that for all $k \in [K]$, we are given $\mathcal{D}_k^{n,m}$ where $n \geq 64\log(2/\delta)$ and $\widehat{a}_k > 0$ where $\widehat{a}_k$ is defined in (14). Then, with probability $1 - \delta$, we have that for all $k \in [K]$,*

$$|\widehat{\gamma}_k - \gamma_k| \leq \frac{\epsilon_a(n, \delta/K, k)}{\gamma_k^{m-1}} = O\left(\frac{1}{\sqrt{n}}\right) \quad and \quad |\widehat{\lambda}_k - \lambda_k| \leq O\left(\frac{1}{\sqrt{n}}\right).$$

The proof of Corollary 7 can be found in Appendix F.1. In the case where we are have collected $n$ trajectories of evenly spaced user utilities for each arm, when the sample size $n$ is sufficient large, the estimation errors of $\widehat{\gamma}_k$ and $\widehat{\lambda}_k$ are $O(1/\sqrt{n})$.

### C.2.2 Estimation using a Single Trajectory

In the case where the learner gets to interact with the user for a long period of time (which is the setting considered in § 5 and § 6), we collect a single trajectory of evenly spaced arm pulls for each arm: for each arm $k \in [K]$, we use $\mathcal{P}_k^{n,m}$ to denote a dataset containing a single trajectory of $n+1$ observed satiation influences $\tilde{x}_{k,1}, \ldots, \tilde{x}_{k,n+1}$, where similar to the multiple trajectories case, $\tilde{x}_{k,1} = 0$, $\tilde{x}_{k,j}$ $(j > 1)$ is the difference between the first received reward and the $j$-th received reward and the time interval between two consecutive pulls is $m$. Thus, for $\tilde{x}_{k,j}, \tilde{x}_{k,j+1} \in \mathcal{P}_k^{n,m}$, it follows that

$$\tilde{x}_{k,j+1} = a_k \tilde{x}_{k,j} + d_k + \tilde{z}_{k,j}, \tag{16}$$

where $a_k$, $d_k$ and $\tilde{z}_{k,j}$ are defined the same as the ones in (12). For all $k \in [K]$, given $\mathcal{P}_k^{n,m}$, we use the following estimators to estimate $A_k = (a_k, d_k)^\top$,

$$\widehat{A}_k = \begin{pmatrix} \widehat{a}_k \\ \widehat{d}_k \end{pmatrix} = (\overline{\mathbf{X}}_{\mathbf{k}}^\top \overline{\mathbf{X}}_{\mathbf{k}})^{-1} \overline{\mathbf{X}}_{\mathbf{k}}^\top \mathbf{Y}_{\mathbf{k}}, \tag{17}$$

where $\mathbf{Y}_{\mathbf{k}} \in \mathbb{R}^n$ is an $n$-dimensional vector whose $j$-th entry is $\tilde{x}_{k,j+1}$ and $\overline{\mathbf{X}}_{\mathbf{k}} \in \mathbb{R}^{n \times 2}$ has its $j$-th row to be the vector $\overline{x}_{k,j} = (\tilde{x}_{k,j}, 1)^\top$. Finally, we take $\widehat{\gamma}_k = |\widehat{a}_k|^{1/m}$ and $\widehat{\lambda}_k = |\widehat{d}_k/\widehat{a}_k|$. We note that $\widehat{A}_k = \arg\min_{A_k \in \mathbb{R}^2} \|\mathbf{Y}_{\mathbf{k}} - \overline{\mathbf{X}}_{\mathbf{k}} A_k\|_2^2$, i.e., it is the ordinary least squares estimator for $A_k$ given the dataset that treats $\tilde{x}_{k,j+1}$ to be the response of the covariates $\overline{x}_{k,j}$.

As we have noted earlier (§ 6.2), unlike the multiple trajectories setting, in the single trajectory case, the difficulty in analyzing the ordinary least squares estimator (17) comes from the fact that the samples are not independent. Asymptotic guarantees of the ordinary least squares estimators in this case have been studied previously in control theory and time series community [13, 22]. The recent work on system identifications for linear dynamical systems focuses on studying the sample complexity of the problem [40, 38]. Adapting the proof of [40, Theorem 2.4], we derive the following theorem for identifying our affine dynamical system (16).

**Theorem 8.** *Fix $\delta \in (0, 1)$. For all $k \in [K]$, there exists a constant $n_0(\delta, k)$ such that if the dataset $\mathcal{P}_k^{n,m}$ satisfies $n \geq n_0(\delta, k)$, then*

$$\mathbb{P}\left(\|\widehat{A}_k - A_k\|_2 \gtrsim \sqrt{1/(\psi n)}\right) \leq \delta,$$

*where $\psi = \sqrt{\min\left\{\frac{\sigma_{z,k}^2(1-a_k)^2}{16d_k^2(1-a_k^2)+(1-a_k)^2\sigma_{z,k}^2}, \frac{\sigma_{z,k}^2}{4(1-a_k^2)}\right\}}$.*

As shown in Theorem 8, when $d_k = \lambda_k \gamma_k^m$ gets larger, the rates of convergence for $\widehat{A}_k$ gets slower. Given that we have a single trajectory of sufficient length, $|\widehat{a}_k - a_k| \leq O(1/\sqrt{n})$ and $|\widehat{d}_k - d_k| \leq O(1/\sqrt{n})$. Similar to the multiple trajectories case, as shown in Corollary 9, the estimators of $\gamma_k$ and $\lambda_k$ also achieve $O(1/\sqrt{n})$ estimation error.

**Corollary 9.** *Fix $\delta \in (0, 1)$. Suppose that for all $k \in [K]$, we have $\mathbb{P}(\|\widehat{A}_k - A_k\|_2 \gtrsim 1/\sqrt{n}) \leq \delta$ and $\widehat{a}_k > 0$ where $\widehat{A}_k$ and $\widehat{a}_k$ are defined in (17). Then, with probability $1 - \delta$, we have that for all $k \in [K]$,*

$$|\widehat{\gamma}_k - \gamma_k| \leq O\left(\frac{1}{\sqrt{n}}\right) \quad and \quad |\widehat{\lambda}_k - \lambda_k| \leq O\left(\frac{1}{\sqrt{n}}\right).$$

In the next section, we assume that the satiation and reward models are estimated using the dataset collected by the proposed exploration strategies and estimators for multiple trajectories or a single trajectory of user utilities. We will show that performing planning based on these estimated models will give us policies that perform well for the true MDP.

### C.3 Planning

For a continuous-state MDP, planning can be done through either dynamic programming with a discretized state space or approximate dynamic programming that uses function approximations. In Appendix C.3.2, we consider the case where we are given a continuous-state MDP planning oracle and provide guarantees of the optimal state-dependent policy planned under the estimated satiation dynamics and reward model. Within the state-dependent policies, we also consider a set of policies that only depend on time (Appendix C.3.1), i.e., the time-dependent competitor class defined in § 5.2. In addition to not requiring discretization of the state space to solve the planning problem, such policies can be deployed to settings where user utilities are hard to attain after the exploration stage. We will show that using the dataset (collected by our exploration strategy in Appendix C.2) with sufficient trajectories (or a sufficient long trajectory) to estimate $\{\gamma_k, \lambda_k\}_{k=1}^K$, the optimal policy $\pi_{\widehat{\mathcal{M}}}^*$ for $\widehat{\mathcal{M}} = \langle x_1, [K], \{\widehat{\gamma}_k, \widehat{\lambda}_k, b_k\}_{k=1}^K, T\rangle$ also performs well in the original MDP $\mathcal{M}$. We note that $b_k$ is known exactly since it is the same as the first observed reward for arm $k$, as discussed in Appendix C.2.

#### C.3.1 Time-dependent Policy

We first show that finding the optimal time-dependent policy is equivalent to solving the bilinear program (4).

**Lemma 4.** *Consider a policy $\pi$ that depends only on the time step $t$ but not the state $x_t$, i.e., $\pi$ satisfies $\pi_t = \pi_t(x_t) = \pi_t(x_t')$ for all $t \in [T]$ and $x_t, x_t' \in \mathcal{X}$. Then, we have*

$$V_{1,\mathcal{M}}^\pi(x_{init}) = \sum_{t=1}^T \mu_{\pi_t,t}(u_{\pi_t,0:t-1}),$$

*where $u_{\pi_t,0:t-1}$ is the corresponding pull sequence of arm $\pi_t$ under policy $\pi$ and $\mu_{k,t}$ is defined in (3).*

*Remark* 6. We denote the policy obtained by solving (4) using model parameters in $\mathcal{M}$ by $\pi_{\mathcal{M}}^T$. Because solving (4) is equivalent to maximizing $\sum_{t=1}^T \mu_{\pi_t,t}(u_{\pi_t,0:t-1})$, Lemma 4 suggests that, for MDP $\mathcal{M}$, the best policy $\pi$ that depends only on the time step $t$ but not the exact state $x_t$ (which we refer as time-dependent policies), is $\pi_{\mathcal{M}}^T$.

**Proposition 5.** *Fix $\delta \in (0, 1)$. Suppose that for all $k \in [K]$, we are given $\mathcal{D}_k^{n,m}$ such that $n \geq 64\log(2/\delta)$ and $\widehat{a}_k \in (\underline{a}, \overline{a})$ for some $0 < \underline{a} < \overline{a} < 1$ almost surely where $\widehat{a}_k$ is defined in (14).*

*Consider a policy $\pi$ that depends on only the time step $t$ but not the state $x_t$. Then, with probability $1 - \delta$, we have that*

$$|V_{1,\mathcal{M}}^{\pi}(x_{init}) - V_{1,\widehat{\mathcal{M}}}^{\pi}(x_{init})| \leq O\left(\frac{T}{\sqrt{n}}\right).$$

*Remark* 7. Proposition 5 applies to time-dependent policies. Such policies can be constructed from an optimal solution to (4) or the $w$-lookahead policy (5). From these results, we deduce that when the historical trajectory is of size $n = O(T)$, the $\sqrt{T}$-lookahead policy $\pi_{\widehat{\mathcal{M}}}^w$ obtained from solving (5) with the parameters from the estimated MDP $\widehat{\mathcal{M}}$ will be $O(\sqrt{T})$-separated from the optimal time-dependent policy $\pi_{\mathcal{M}}^T$ obtained by solving (4) with the true parameters of $\mathcal{M}$. That is,

$$0 \leq V_{1,\mathcal{M}}^{\pi_{\mathcal{M}}^T}(x_{\text{init}}) - V_{1,\mathcal{M}}^{\pi_{\widehat{\mathcal{M}}}^w}(x_{\text{init}}) = V_{1,\mathcal{M}}^{\pi_{\mathcal{M}}^T}(x_{\text{init}}) - V_{1,\widehat{\mathcal{M}}}^{\pi_{\mathcal{M}}^T}(x_{\text{init}}) + V_{1,\widehat{\mathcal{M}}}^{\pi_{\mathcal{M}}^T}(x_{\text{init}}) - V_{1,\widehat{\mathcal{M}}}^{\pi_{\widehat{\mathcal{M}}}^T}(x_{\text{init}})$$

$$+ V_{1,\widehat{\mathcal{M}}}^{\pi_{\widehat{\mathcal{M}}}^T}(x_{\text{init}}) - V_{1,\widehat{\mathcal{M}}}^{\pi_{\widehat{\mathcal{M}}}^w}(x_{\text{init}}) + V_{1,\widehat{\mathcal{M}}}^{\pi_{\widehat{\mathcal{M}}}^w}(x_{\text{init}}) - V_{1,\mathcal{M}}^{\pi_{\widehat{\mathcal{M}}}^w}(x_{\text{init}})$$

$$\leq |V_{1,\mathcal{M}}^{\pi_{\mathcal{M}}^T}(x_{\text{init}}) - V_{1,\widehat{\mathcal{M}}}^{\pi_{\mathcal{M}}^T}(x_{\text{init}})| + |V_{1,\widehat{\mathcal{M}}}^{\pi_{\widehat{\mathcal{M}}}^T}(x_{\text{init}}) - V_{1,\widehat{\mathcal{M}}}^{\pi_{\widehat{\mathcal{M}}}^w}(x_{\text{init}})| + |V_{1,\widehat{\mathcal{M}}}^{\pi_{\widehat{\mathcal{M}}}^w}(x_{\text{init}}) - V_{1,\mathcal{M}}^{\pi_{\widehat{\mathcal{M}}}^w}(x_{\text{init}})|$$

$$\leq O(\sqrt{T}),$$

where the second inequality follows from the fact that $V_{1,\widehat{\mathcal{M}}}^{\pi_{\mathcal{M}}^T}(x_{\text{init}}) - V_{1,\widehat{\mathcal{M}}}^{\pi_{\widehat{\mathcal{M}}}^T}(x_{\text{init}}) \leq 0$ (since for the MDP $\widehat{\mathcal{M}}$, $\pi_{\widehat{\mathcal{M}}}^T$ is the optimal time-dependent policy), and the third (last) inequality is derived by applying Proposition 5 twice and using Remark 4.

### C.3.2 State-dependent Policy

In Proposition 6, we show that the difference between the value of the optimal state-dependent policy $\pi_{\mathcal{M}}^*$, and the value of the optimal state-dependent policy $\pi_{\widehat{\mathcal{M}}}^*$ planned under the estimated $\widehat{\mathcal{M}}$ is of order $O(T^2/\sqrt{n})$ where $n$ is the number of historical trajectories if we use multiple trajectories to estimate $\gamma_k$ and $\lambda_k$.

**Proposition 6.** *Fix $\delta \in (0,1)$. Suppose that for all $k \in [K]$, we are given $\mathcal{D}_k^{n,m}$ such that $n \geq 64 \log(2/\delta)$ and $\widehat{a}_k \in (\underline{a}, \overline{a})$ for some $0 < \underline{a} < \overline{a} < 1$ almost surely where $\widehat{a}_k$ is defined in (14). Then, with probability $1 - \delta$,*

$$|V_{1,\mathcal{M}}^{*}(x_{init}) - V_{1,\widehat{\mathcal{M}}}^{\pi_{\widehat{\mathcal{M}}}^*}(x_{init})| \leq O\left(\frac{T^2}{\sqrt{n}}\right).$$

*Remark* 8. The assumptions in Proposition 5 and 6 correspond to the case where we use multiple trajectories to estimate the satiation dynamics and reward model. They can be replaced by conditions on single trajectory datasets when one uses a single trajectory to estimate the parameters.

In summary, as Proposition 6 suggests, when given a continuous-state MDP planning oracle, our algorithm obtain a policy $\pi_{\widehat{\mathcal{M}}}^*$ that is $O(T^2/\sqrt{n})$ away from the optimal policy $\pi_{\mathcal{M}}^*$ under the true MDP $\mathcal{M}$ where the size of the exploration stage for our algorithm (EEP) is $O(Kn)$ and the horizon of the exploitation/planning stage is $T$. We also note that the optimal state-dependent policy $\pi_{\mathcal{M}}^*$ is the optimal competitor policy when the competitor class (§ 5.2) contains all measurable functions from $\mathcal{X}$ to $[K]$.

# D Proofs of Section 6.2 and Appendix C.2.2

## D.1 Proof of Theorem 3 and Theorem 8

We notice that Theorem 3 is a consequence of Theorem 8 when $m = 1$. More specifically, the dataset $\mathcal{P}_k^n$ and the parameter $A_k = (\gamma_k, \lambda_k \gamma_k)^\top$ in Theorem 3 is a special case of the dataset $\mathcal{P}_k^{n,m}$ and parameter $A_k = (\gamma_k^m, \lambda_k \gamma_k^m)^\top$ considered in Theorem 8 by taking $m = 1$. Thus, below we directly present the proof of Theorem 8 where we use the notation from Theorem 8 (and Appendix C.2.2), i.e., $a_k = \gamma_k^m$ and $d_k = \lambda_k \gamma_k^m$.

We begin with presenting some key results from [40]; we utilize these results in establishing the sample complexity of our estimator for identifying an affine dynamical system in Appendix C.2.2.

**Definition 1.** *[40, Definition 2.1] Let $\{\phi_t\}_{t\geq 1}$ be an $\{\mathcal{F}_t\}_{t\geq 1}$-adapted random process taking values in $\mathbb{R}$. We say $(\phi_t)_{t\geq 1}$ satisfies the $(k, \nu, p)$-block martingale small-ball (BMSB) condition if, for any $j \geq 0$, one has $\frac{1}{k}\sum_{i=1}^k \mathbb{P}(|\phi_{j+i}| \geq \nu | \mathcal{F}_j) \geq p$ almost surely. Given a process $(X_t)_{t\geq 1}$ taking values in $\mathbb{R}^d$, we say that it satisfies the $(k, \Gamma_{sb}, p)$-BMSB condition for $\Gamma_{sb} \succ 0$ if for any fixed $w$ in the unit sphere of $\mathbb{R}^d$, the process $\phi_t := \langle w, X_t \rangle$ satisfies $(k, \sqrt{w^\top \Gamma_{sb} w}, p)$-BMSB.*

**Proposition 7.** *[40, Proposition 2.5] Fix a unit vector $w \in \mathbb{R}^d$, define $\phi_t = w^\top X_t$. If the scalar process $\{\phi_t\}_{t\geq 1}$ satisfies the $(l, \sqrt{w^\top \Gamma_{sb} w}, p)$-BMSB condition for some $\Gamma_{sb} \in \mathbb{R}^{d\times d}$, then*

$$\mathbb{P}\left(\sum_{t=1}^n \phi_t^2 \leq \frac{w^\top \Gamma_{sb} w p^2}{8} l \lfloor T/l \rfloor \right) \leq \exp\left(-\frac{\lfloor T/l \rfloor p^2}{8}\right).$$

**Theorem 10.** *[40, Theorem 2.4] Fix $\delta \in (0,1)$, $T \in \mathbb{N}$ and $0 \prec \Gamma_{sb} \preceq \overline{\Gamma}$. Then if $(X_t, Y_t)_{t\geq 1} \in (\mathbb{R}^d \times \mathbb{R}^n)^n$ is a random sequence such that (a) $Y_t = AX_t + \eta_t$, where $\mathcal{F}_t = \sigma(\eta_1, \ldots, \eta_t)$ and $\eta_t | \mathcal{F}_{t-1}$ is $\sigma^2$-sub-Gaussian and mean zero, (b) $X_1, \ldots, X_T$ satisfies the $(l, \Gamma_{sb}, p)$-BMSB condition, and (c) $\mathbb{P}(\sum_{t=1}^n X_t X_t^\top \not\succeq T\overline{\Gamma}) \geq \delta$. Then if*

$$T \geq \frac{10l}{p^2}\left(\log(1/\delta) + 2d\log(10/p) + \log\det(\overline{\Gamma}\Gamma_{sb}^{-1})\right),$$

*we have that for $\widehat{A} = \arg\min_{A\in\mathbb{R}^{n\times d}} \sum_{t=1}^T \|Y_t - AX_t\|_2^2$,*

$$\mathbb{P}\left(\|\widehat{A} - A\|_{op} > \frac{90\sigma}{p}\sqrt{\frac{n + d\log(10/p) + \log\det\left(\overline{\Gamma}\Gamma_{sb}^{-1}\right) + \log(1/\delta)}{T\lambda_{\min}(\Gamma_{sb})}}\right) \leq 3\delta.$$

We note that in the proof of Theorem 10 in [40], condition (b) is used through applying Proposition 7 to ensure that for any unit vector $w \in \mathbb{R}^d$,

$$\mathbb{P}\left(\sum_{t=1}^T \langle w, X_t \rangle^2 \leq \frac{(w^\top \Gamma_{sb} w)p^2}{8} l \lfloor T/l \rfloor\right) \leq \exp\left(-\frac{\lfloor T/l \rfloor p^2}{8}\right). \tag{18}$$

To apply Theorem 10 in our setting to obtain Theorem 8, we verify condition $(a)$ and $(c)$. For condition $(b)$, we show a result similar to (18). The below technical lemmas are used in our proof of Theorem 8.

**Lemma 8.** *Let $a, b$ be scalars with $b > 0$. Suppose that $X \sim N(a, b)$. Then for any $\theta \in [0, 1]$,*

$$\mathbb{P}(|X| \geq \sqrt{\theta(a^2 + b)}) \geq \frac{(1-\theta)^2}{9}.$$

*Proof.* By the Paley-Zygmund inequality,

$$\mathbb{P}(|X| \geq \sqrt{\theta\mathbb{E}[X^2]}) = \Pr(X^2 \geq \theta\mathbb{E}[X^2]) \geq (1-\theta)^2\frac{\mathbb{E}[X^2]^2}{\mathbb{E}[X^4]}.$$

Using the mean and variance of non-central chi-squared distributions, we obtain that

$$\mathbb{E}[X^2] = a^2 + b,$$
$$\mathbb{E}[X^4] = a^4 + 6a^2b + 3b^2 = (a^2 + 3b)^2 - 6b^2.$$

Plugging them back to the Paley-Zygmund inequality, we have that

$$\mathbb{P}(|X| \geq \sqrt{\theta(a^2 + b)}) \geq \frac{(1-\theta)^2}{9},$$

where the last inequality uses the fact that $\mathbb{E}[X^4] \leq (a^2 + 3b)^2 \leq 9(a^2 + b)^2 = 9\mathbb{E}[X^2]^2$.  □

**Lemma 9.** *Let* $\{\phi_t\}_{t \geq 1}$ *be a scalar process satisfying that*

$$\frac{1}{l} \sum_{i=1}^{l} \mathbb{P}(|\phi_{t+i}| \geq \nu_t | \mathcal{F}_t) \geq p,$$

*for* $\nu_t$ *depending on* $\mathcal{F}_t$. *If* $\mathbb{P}(\min_t \nu_t \geq \nu) \geq 1 - \delta$ *for* $\nu > 0$ *that depends on* $\delta$, *then*

$$\mathbb{P}\left(\sum_{t=1}^{T} \phi_t^2 \leq \frac{\nu^2 p^2}{8} l \lfloor T/l \rfloor\right) \leq \exp\left(-\frac{3\lfloor T/l \rfloor p}{4}\right) + \delta.$$

*Proof.* We begin with partitioning $Z_1, \ldots, Z_T$ into $S := \lfloor T/l \rfloor$ blocks of size $l$. Consider the random variables

$$B_j = \mathbf{1}\left(\sum_{i=1}^{l} \phi_{jl+i}^2 \geq \frac{\nu_{jl}^2 pk}{2}\right), \quad \text{for } 0 \leq j \leq S - 1.$$

We observe that

$$\mathbb{P}\left(\sum_{t=1}^{T} \phi_t^2 \leq \frac{\nu^2 p^2}{8} l \lfloor T/l \rfloor\right) = \mathbb{P}\left(\left\{\sum_{t=1}^{T} \phi_t^2 \leq \frac{\nu^2 p^2}{8} l \lfloor T/l \rfloor\right\} \cap \{\min_t \nu_t \geq \nu\}\right)$$

$$+ \mathbb{P}\left(\left\{\sum_{t=1}^{T} \phi_t^2 \leq \frac{\nu^2 p^2}{8} l \lfloor T/l \rfloor\right\} \cap \{\min_t \nu_t < \nu\}\right)$$

$$\leq \mathbb{P}\left(\left\{\sum_{t=1}^{T} \phi_t^2 \leq \frac{\nu_{\lfloor t/l \rfloor l}^2 p^2}{8} lS\right\} \cap \{\min_t \nu_t \geq \nu\}\right) + \mathbb{P}(\min_t \nu_t < \nu)$$

$$\leq \mathbb{P}\left(\sum_{t=1}^{T} \phi_t^2 \leq \frac{\nu_{\lfloor t/l \rfloor l}^2 p^2}{8} kS\right) + \delta.$$

Using Chernoff bound, we obtain that

$$\mathbb{P}\left(\sum_{t=1}^{T} \phi_t^2 \leq \frac{\nu_{\lfloor t/l \rfloor l}^2 p^2}{8} kS\right) \leq \mathbb{P}\left(\sum_{j=0}^{S-1} \sum_{i=1}^{l} \phi_{jl+i}^2 \leq \frac{\nu_{jl}^2 p^2}{8} lS\right) = \mathbb{P}\left(\sum_{j=0}^{S-1} \sum_{i=1}^{l} \phi_{jl+i}^2 \leq \frac{\nu_{jl}^2 p^2}{8} lS\right)$$

$$\leq \mathbb{P}\left(\sum_{j=0}^{S-1} B_j \leq \frac{p}{4} S\right) \leq \inf_{\lambda \leq 0} e^{-\frac{pS}{4}} \mathbb{E}[e^{\lambda \sum_{j=0}^{S-1} B_j}],$$

where the second to the last inequality uses the fact that $\frac{\nu_{jl}^2 pl}{2} B_j \leq \sum_{i=1}^{l} \phi_{jl+i}^2$ Further, we have that

$$\mathbb{E}[B_j | \mathcal{F}_{jl}] = \mathbb{P}\left(\sum_{i=1}^{l} \phi_{jl+i}^2 \geq \frac{\nu_{jl}^2 pl}{2} \Big| \mathcal{F}_{jl}\right) \geq \mathbb{P}\left(\frac{1}{l} \sum_{i=1}^{l} \mathbf{1}\{|\phi_{jl+i}| \geq \nu_{jl}\} \geq \frac{p}{2} \Big| \mathcal{F}_{jl}\right)$$

$$\geq \frac{p}{2},$$

where the first inequality uses the fact that $\frac{1}{\nu_{jl}^2}\phi_{jl+i}^2 \geq \mathbf{1}\{\phi_{jl+i}| \geq \nu_{jl}\}$ and the last inequality uses the fact that for a random variable $X$ supported on $[0,1]$ almost surely such that $\mathbb{E}[X] \geq p$ for some $p \in (0,1)$, then for all $t \in [0,p]$, $\mathbb{P}(X \geq t) \geq \frac{p-t}{1-t}$. This is true because

$$\mathbb{P}(X \geq t) = \int_t^1 d\mathbb{P}(x) \geq \int_t^1 x d\mathbb{P}(x) = \int_0^1 x d\mathbb{P}(x) - \int_0^t x d\mathbb{P}(x) = p - t(1 - \mathbb{P}(X \geq t)).$$

In our case, $\mathbb{E}\left[\frac{1}{l}\sum_{i=1}^l \mathbf{1}\{|\phi_{jl+i}| \geq \nu_{jl}\}\Big|\mathcal{F}_{jl}\right] = \frac{1}{l}\sum_{i=1}^l \mathbb{P}\left(|\phi_{jl+i}| \geq \nu_{jl}\Big|\mathcal{F}_{jl}\right) \geq p$. Thus, we obtain that for $\lambda \leq 0$, i.e., $e^\lambda \leq 1$,

$$\mathbb{E}[e^{\lambda B_j}|\mathcal{F}_{jl}] = e^\lambda \mathbb{P}\left(B_j = 1\Big|\mathcal{F}_{jl}\right) + \mathbb{P}(B_j = 0) = (e^\lambda - 1)\mathbb{E}[B_j|\mathcal{F}_{jl}] + 1 \leq (e^\lambda - 1)\frac{p}{2} + 1.$$

By law of iterated expectation, we obtain that

$$\mathbb{E}[e^{\lambda \sum_{j=0}^{S-1} B_j}] = \mathbb{E}\left[e^{\lambda \sum_{j=0}^{S-2} B_j}\mathbb{E}[e^{\lambda B_j}|\mathcal{F}_{(S-1)k}]\right] \leq \left((e^\lambda - 1)\frac{p}{2} + 1\right)\mathbb{E}\left[e^{\lambda \sum_{j=0}^{S-2} B_j}\right] \leq \left((e^\lambda - 1)\frac{p}{2} + 1\right)^S.$$

Finally, we need to find

$$\inf_{\lambda \leq 0} e^{-pS/4}\left((e^\lambda - 1)\frac{p}{2} + 1\right)^S.$$

We can see that $\lambda^* = -\infty$, which gives that

$$\inf_{\lambda \leq 0} e^{-pS/4}\left((e^\lambda - 1)\frac{p}{2} + 1\right)^S = e^{-pS/4}\left(1 - \frac{p}{2}\right)^S \leq e^{-pS/4}e^{-pS/2} = e^{-3pS/4},$$

where we have used the fact that $1 + x \leq e^x$ for all real-valued $x$. $\qquad\square$

To apply Theorem 10, we first recall that the affine dynamical system we aim to identify is as follows:

$$\tilde{x}_{k,j+1} = a_k\tilde{x}_{k,j} + d_k + \tilde{z}_{k,j},$$

where $\tilde{x}_{k,1} = 0$, $a_k \in (0,1)$ and $\tilde{z}_{k,j} \sim \mathcal{N}(0, \sigma_{z,k}^2)$. We define the following quantities

$$\Gamma_{k,j} := \sigma_{z,k}^2 \sum_{i=0}^{j-1} a_k^{2i}, \qquad d_{k,j} := \sum_{i=0}^{j-1} a_k^j d_k,$$

and $\Gamma_{k,\infty} = \sigma_{z,k}^2 \sum_{i=0}^\infty a_k^{2i} = \frac{\sigma_{z,k}^2}{1-a_k^2}$. We notice that for all $t \in [T]$, $j \geq 1$,

$$\tilde{x}_{k,t+j}|\tilde{x}_{k,t} \sim \mathcal{N}\left(a_k^j\tilde{x}_{k,t} + d_{k,j}, \Gamma_{k,j}\right).$$

**Lemma 10.** *Fix $t \geq 0$ and $j \geq 1$. Recall that $\overline{x}_{k,t} := (\tilde{x}_{k,t}, 1) \in \mathbb{R}^2$. Fix a unit vector $w \in \mathbb{R}^2$. For any $\epsilon \in (0,1)$, we have*

$$\mathbb{P}\left(|\langle w, \overline{x}_{k,t+j}\rangle| \geq \frac{1}{\sqrt{2}}\sqrt{\min\left\{1 - \epsilon, \Gamma_{k,j} - \left(\frac{1}{\epsilon} - 1\right)(a_k^j\tilde{x}_{k,t} + d_{k,j})^2\right\}}\right) \geq \frac{1}{36}$$

*Proof.* By Lemma 8, we have that for any unit vector $w \in \mathbb{R}^2$,

$$\mathbb{P}\left\{|\langle w, \overline{x}_{k,t+j}\rangle| \geq \frac{1}{\sqrt{2}}\sqrt{\left(w_1\left(a_k^j\tilde{x}_{k,t} + d_{k,j}\right) + w_2\right)^2 + w_1^2\Gamma_{k,j}}\,\Big|\,\overline{x}_{k,t}\right\} \geq \frac{1}{36}.$$

For all $\epsilon \in (0,1)$, we have

$$((w_1(a_k^j\tilde{x}_{k,t} + d_{k,j}) + w_2)^2 + w_1^2\Gamma_{k,j} = \left(w_1\left(a_k^j\tilde{x}_{k,t} + d_{k,j}\right)\right)^2 + w_2^2 + 2w_2w_1\left(a_k^j\tilde{x}_{k,t} + d_{k,j}\right) + w_1^2\Gamma_{k,j}$$

$$\geq (1 - \epsilon)w_2^2 - \left(\frac{1}{\epsilon} - 1\right)\left(w_1\left(a_k^j\tilde{x}_{k,t} + d_{k,j}\right)\right)^2 + w_1^2\Gamma_{k,j}$$

$$\geq \min\left\{1 - \epsilon, \Gamma_{k,j} - \left(\frac{1}{\epsilon} - 1\right)(a_k^j\tilde{x}_{k,t} + d_{k,j})^2\right\}.$$

$\qquad\square$

**Lemma 11.** *Fix $\delta \in (0,1)$. $\{\overline{x}_{k,t}\}_{t=1}^n$ satisfy that for any unit vector $w \in \mathbb{R}^2$,*

$$\mathbb{P}\left(\sum_{t=1}^n \langle w, \overline{x}_{k,t}\rangle^2 \leq \frac{\psi^2 p^2}{16} j_\star \lfloor n/j_\star \rfloor\right) \leq \exp\left(-\frac{3\lfloor n/j_\star \rfloor p}{4}\right) + \delta$$

*with $p = 1/72$,*

$$j_\star := \left\lceil \max\left\{-\log_{a_k}\left(1 + (1-a_k)\frac{\sqrt{2\Gamma_{k,\infty}\log(n/\delta)}}{d_k}\right), -\log_{a_k}\sqrt{2}\right\}\right\rceil,$$

$$\psi := \sqrt{\min\left\{\frac{\Gamma_{k,\infty}}{\frac{16d_k^2}{(1-a_k)^2} + \Gamma_{k,\infty}}, \frac{\Gamma_{k,\infty}}{4}\right\}}.$$

*Proof.* Fix $\delta \in (0,1)$. Recall that from Lemma 9, we have shown that for all $t \geq 0$ and $k \geq 1$, given a unit vector $w \in \mathbb{R}^2$, for any $\epsilon \in (0,1)$, we have

$$\mathbb{P}\left\{|\langle w, \overline{x}_{k,t+j}\rangle| \geq \frac{1}{\sqrt{2}}\sqrt{\min\left\{1-\epsilon, \Gamma_{k,j} - \left(\frac{1}{\epsilon}-1\right)(a_k^j \tilde{x}_{k,t} + d_{k,j})^2\right\}}\right\} \geq \frac{1}{36}.$$

Denote $q_{t,j} = a_k^j \tilde{x}_{k,t} + d_{k,j}$ where $\tilde{x}_{k,t} \sim \mathcal{N}(d_{k,t}, \Gamma_{k,t})$. Fix $\delta \in (0,1)$. Using the standard Gaussian tail bound and the union bound, we have that with probability $1-\delta$,

$$\max_{t \in [T]} q_{t,j} \leq a_k^j \left(\frac{d_k}{1-a_k} + \sqrt{2\Gamma_\infty \log(n/\delta)}\right) + \frac{d_k}{1-a_k}.$$

When $j \geq j_\star$, $\Gamma_{k,j} \geq \Gamma_{k,\infty}/2$, and with probability $1-\delta$, $\max_{t \in [T]} q_{t,j} \leq \frac{2d_k}{1-a_k}$. Thus, for $j \geq j_\star$, and

$$\epsilon = \frac{\frac{4d_k^2}{(1-a_k)^2}}{\frac{4d_k^2}{(1-a_k)^2} + \Gamma_\infty/4},$$

we have

$$\nu_{t,j}^2 := \min\left\{1-\varepsilon, \Gamma_{k,j} - \left(\frac{1}{\varepsilon}-1\right)q_{t,j}^2\right\}$$

$$\geq \min\left\{1-\varepsilon, \Gamma_{k,\infty}/2 - \left(\frac{1}{\varepsilon}-1\right)\frac{4d_k^2}{(1-a_k)^2}\right\}$$

$$\geq \min\left\{\frac{\Gamma_{k,\infty}}{\frac{16d_k^2}{(1-a_k)^2} + \Gamma_{k,\infty}}, \frac{\Gamma_{k,\infty}}{4}\right\} = \psi^2.$$

Putting it altogether, we have

$$\frac{1}{2j_\star}\sum_{j=1}^{2j_\star}\mathbb{P}\left(|\langle w, \overline{x}_{k,t+j}\rangle| \geq \nu_{t,j}/\sqrt{2}|\mathcal{F}_t\right) \geq \frac{1}{2j_\star}\sum_{j=j_\star}^{2j_\star}\Pr(|\langle w, \overline{x}_{k,t+j}\rangle| \geq \nu_{t,j_\star}/\sqrt{2}|\mathcal{F}_t) \geq \frac{1}{72}.$$

Further, we have

$$\mathbb{P}\left(\min_{t \in [T]}\nu_{t,j_\star}^2 \geq \psi^2\right) \geq 1-\delta.$$

Applying Lemma 9, we have that for $p = \frac{1}{72}$,

$$\mathbb{P}\left(\sum_{t=1}^n \langle w, \overline{x}_{k,t}\rangle^2 \leq \frac{\psi^2 p^2}{16}j_\star \lfloor n/j_\star \rfloor\right) \leq \exp\left(-\frac{3\lfloor n/j_\star \rfloor p}{4}\right) + \delta.$$

$\square$

*Proof of Theorem 8.* Based on our setup, condition $(a)$ of Theorem 10 is satisfied. For any $n$, using Lemma 11 with $\delta = \exp(-n)$, we have that

$$\forall w \in \mathbb{R}^2, \quad \mathbb{P}\left(\sum_{t=1}^n \langle w, \overline{x}_{k,t}\rangle^2 \leq \frac{\psi^2 p^2}{16} j_\star \lfloor n/j_\star \rfloor\right) \leq \exp\left(-\frac{3\lfloor n/j_\star \rfloor p}{4}\right) + \delta \leq 2\exp\left(-\frac{3\lfloor n/j_\star \rfloor p}{4}\right),$$

with $p = 1/72$,

$$j_\star := \left\lceil \max\left\{-\log_{a_k}\left(1 + (1 - a_k)\frac{\sqrt{2\Gamma_{k,\infty}(\log(n) + n)}}{d_k}\right), -\log_{a_k}\sqrt{2}\right\}\right\rceil,$$

$$\psi := \sqrt{\min\left\{\frac{\Gamma_{k,\infty}}{\frac{16d_k^2}{(1-a_k)^2} + \Gamma_{k,\infty}}, \frac{\Gamma_{k,\infty}}{4}\right\}}.$$

Thus, we have provided a similar result to (18), which is what condition (b) of Theorem 10 is used for. In this case, we have $\Gamma_{\text{sb}} = \psi I$ where $I$ is a $2 \times 2$ identity matrix. Finally, to verify condition (c), we notice that we have

$$\overline{\Gamma}_{k,j} := \mathbb{E}[\overline{x}_{k,j}\overline{x}_{k,j}^\top] = \begin{pmatrix} \frac{b_k^2(1-a_k^{j-1})^2}{(1-a_k)^2} + \frac{\sigma_{z,k}^2(1-a_k^{2j-2})}{1-a_k^2} & \frac{(1-a_k^{j-1})b_k}{1-a_k} \\ \frac{(1-a_k^{j-1})b_k}{1-a_k} & 1 \end{pmatrix}.$$

and we denote

$$\overline{\Gamma} := \overline{\Gamma}_{k,n} + \begin{pmatrix} 0 & 0 \\ 0 & 1 \end{pmatrix} + \Gamma_{\text{sb}},$$

which gives that $0 \prec \Gamma_{\text{sb}} \prec \overline{\Gamma}$ and for all $j \geq 1$, $0 \preceq \overline{\Gamma}_{k,j} \prec \overline{\Gamma}$. Then, we have that

$$\mathbb{P}\left(\overline{\mathbf{X}}_k^\top \overline{\mathbf{X}}_k \not\preceq \frac{2n}{\delta}\overline{\Gamma}\right) = \mathbb{P}\left(\lambda_{\max}\left((n\overline{\Gamma})^{-1/2}\overline{\mathbf{X}}_k^\top \overline{\mathbf{X}}_k(n\overline{\Gamma})^{-1/2}\right) \geq \frac{2}{\delta}\right)$$

$$\leq \frac{\delta}{2}\mathbb{E}\left[\lambda_{\max}\left((n\overline{\Gamma})^{-1/2}\overline{\mathbf{X}}_k^\top \overline{\mathbf{X}}_k(n\overline{\Gamma})^{-1/2}\right)\right]$$

$$\leq \frac{\delta}{2}\mathbb{E}\left[\text{tr}\left((n\overline{\Gamma})^{-1/2}\overline{\mathbf{X}}_k^\top \overline{\mathbf{X}}_k(n\overline{\Gamma})^{-1/2}\right)\right] \leq \delta,$$

where the last inequality is true since $\mathbb{E}\left[\overline{\mathbf{X}}_k^\top \overline{\mathbf{X}}_k\right] = \sum_{j=1}^n \Gamma_{k,j} \preceq n\overline{\Gamma}$ (for all $j \in [n]$, $\text{trace}(\overline{\Gamma} - \overline{\Gamma}_{k,j}) > 0$ and $\det(\overline{\Gamma} - \overline{\Gamma}_{k,j}) > 0$). Following Theorem 10, for $\delta \in (0,1)$, when the number of samples satisfy that

$$\frac{n}{j_\star} \geq \frac{10}{p^2}\left(\log(1/\delta) + 4\log(10/p) + \log\det(\overline{\Gamma}\Gamma_{\text{sb}}^{-1})\right),$$

we have that

$$\mathbb{P}\left(\|\widehat{A}_k - A_k\|_2 > \frac{90\sigma_{z,k}}{p}\sqrt{\frac{1 + 2\log(10/p) + \log\det\left(\overline{\Gamma}\Gamma_{\text{sb}}^{-1}\right) + \log(1/\delta)}{n\psi}}\right) \leq 3\delta.$$

$\square$

## D.2 Proof of Corollary 4 and Corollary 9

Similar to Appendix D.1, Corollary 4 is a special case of Corollary 9 when $m = 1$. Hence, we directly present the proof of Corollary 9 below.

*Proof of Corollary 9.* Fix $\delta \in (0,1)$. We have that with probability $1 - \delta$, $\epsilon(n, \delta, k) := \|\widehat{A}_k - A_k\|_2 \leq O(1/\sqrt{n})$. With probability at least $1 - \frac{\delta}{K}$, $\epsilon_{a_k} := |\widehat{a}_k - a_k| \leq \|\widehat{A}_k - A_k\|_2 = \epsilon(n, \delta/K, k) = O(1/\sqrt{n})$ and $\epsilon_{d_k} := |\widehat{d}_k - b_k| \leq \|\widehat{A}_k - A_k\|_2 = \epsilon(n, \delta/K, k) = O(1/\sqrt{n})$. When $m = 1$, then

$|\widehat{\gamma}_k - \gamma_k| = ||\widehat{a}_k| - a_k| \le |\widehat{a}_k - a_k| = \epsilon_{a_k} \le \epsilon(n, \delta/K, k)$. When $m \ge 2$, since $\gamma_k \ne 0$, we have that

$$|\widehat{\gamma}_k - \gamma_k| = \left| \frac{|\widehat{a}_k| - a_k}{|\widehat{a}_k|^{(m-1)/m} + |\widehat{a}_k|^{(m-2)/m}\gamma_k + \ldots + \gamma_k^{m-1}} \right| \le \frac{|\widehat{a}_k - a_k|}{\gamma_k^{m-1}}.$$

On the other hand, we obtain that

$$|\widehat{\lambda}_k - \lambda_k| = \left| \left| \frac{\widehat{d}_k}{\widehat{a}_k} \right| - \frac{d_k}{a_k} \right| \le \left| \frac{\widehat{d}_k}{\widehat{a}_k} - \frac{d_k}{\widehat{a}_k} + \frac{d_k}{\widehat{a}_k} - \frac{d_k}{a_k} \right| \le \frac{\epsilon_{d_k}}{\widehat{a}_k} + \frac{\lambda_k \epsilon_{a_k}}{\widehat{a}_k} \le O\left(\frac{1}{\sqrt{n}}\right).$$

The proof completes as follows:

$$\mathbb{P}\left( \forall k \in [K], |\widehat{\gamma}_k - \gamma_k| \le O(1/\sqrt{n}), |\widehat{\lambda}_k - \lambda_k| \le O(1/\sqrt{n}) \right) \ge \prod_{k=1}^{K} \left( 1 - \frac{\delta}{K} \right) \ge 1 - \delta,$$

where the last inequality follows from Bernoulli's inequality. $\qquad\square$

# E  Additional Proofs and Discussion of Section 6

## E.1  Proof of Theorem 5

**Lemma 12.** *Consider any episode $i + 1$ (from time $t_i + 1$ to $t_{i+1}$) where the initial state $x^i = (\mu_{1,t_i+1}(u_{1,0:t_i}), n_{1,t_i+1}, \dots, \mu_{K,t_i+1}(u_{K,0:t_i}), n_{K,t_i})$ and $\{u_{k,0:t_i}\}_{k=1}^K$ are the past pull sequences of the proposed policy $\pi_{1:t_i}$. For all $\tilde\pi_{t_i+1:t_{i+1}}$ such that $\tilde\pi_t = \tilde\pi_t(x_t) = \tilde\pi_t(x_t'), \tilde\pi_t \in [K], \forall t \in [t_i+1, t_{i+1}], x_t, x_t' \in \mathcal{X}$, we have that*

$$\sum_{t=t_i+1}^{t_{i+1}} \mathbb{E}_{x_{t_i+2},\dots,x_{t_i}}\left[r(x_t, \tilde\pi_t(x_t))|x_{t_i+1} = x^i\right] = \sum_{t=t_i+1}^{t_{i+1}} \mu_{k,t}(u_{k,0:t-1}),$$

*where $\{u_{k,t_i+1:t_{i+1}}\}_{k=1}^K$ is the arm pull sequence of $\tilde\pi_{t_i+1:t_{i+1}}$.*

*Proof.* Let $k$ denote $\tilde\pi_t$ where $t \in \{t_i + 1, \dots, t_{i+1}\}$. Recall that we use $u_{k,0:t-1}$ to denote the pull sequence of arm $k$ under policy $\tilde\pi_{1:t_{i+1}} = (\pi_{1:t_i}, \tilde\pi_{t_i+1:t_{i+1}})$. If $k$ has not been pulled before time $t$ by $\tilde\pi_{1:t_{i+1}}$, then $\mathbb{E}_{x_{t_i+2},\dots,x_{t_{i+1}}}\left[r(x_t, \tilde\pi_t)|x_{t_i+1} = x^i\right] = b_{\pi_t} = \mu_{\pi_t,t}(u_{\pi_t,0:t-1})$. If $k$ has been pulled before, then let $q_1, \dots, q_n$ denote the time steps that arm $k$ has been pulled before time $t$ by $\tilde\pi_{1:t_{i+1}}$, i.e., $u_{k,q_i} = 1$ for $i \in [n]$ and $u_{k,t'} = 0$ for $t' \notin \{q_1, \dots, q_n\}$. We have that for $t \in \{t_i+1, \dots, t_{i+1}\}$,

$$\mathbb{E}_{x_{t_i+2},\dots,x_{t_{i+1}}}\left[r(x_t, \tilde\pi_t)|x_{t_i+1} = x^i\right]$$

$$= b_k - \left(\mathbb{E}_{x_{t_i+2},\dots,x_{t_{i+1}-1}}\left[\mathbb{E}_{x_{t_i+1}}\left[\gamma_k^{n_{k,t_i+1}} x_{k,t_i+1} + \lambda_k \gamma_k^{n_{k,t_i+1}}\right]|x_{t_i+1} = x^i\right]\right)$$

$$= b_k - \left(\mathbb{E}_{x_{t_i+2},\dots,x_{q_n}}\left[\mathbb{E}_{x_{q_n+1}}\left[\gamma_k^{n_{k,t_i+1}} x_{k,q_n+1} + \lambda_k \gamma_k^{n_{k,t_i+1}}\right]|x_{t_i+1} = x^i\right]\right)$$

$$= b_k - \left(\mathbb{E}_{x_{t_i+2},\dots,x_{q_n}}\left[\gamma_k^{n_{k,t_i+1}}\left(\gamma_k^{n_{k,q_n}} x_{k,q_n} + \lambda_k \gamma_k^{n_{k,q_n}}\right) + \lambda_k \gamma_k^{n_{k,t_i+1}}|x_{t_i+1} = x^i\right]\right)$$

$$= \dots = b_k - \lambda_k\left(\gamma_k^{n_{k,t_i+1}} + \gamma_k^{n_{k,t_i+1}+n_{k,q_n}} + \dots + \gamma_k^{n_{k,t_i+1}+n_{k,q_n}+\dots n_{k,q_1}}\right)$$

$$= \mu_{k,t}(u_{k,0:t-1}),$$

where the second equality is true because when arm $k$ is not pulled for example at time $t_{i+1} - 1$, the state for arm $k$ at time $t_{i+1} - 1$ will satisfy that $x_{k,t_{i+1}} = x_{k,t_{i+1}-1}$ and $n_{k,t_{i+1}} = n_{k,t_{i+1}-1} + 1$ with probability 1. In this case, we have that

$$\mathbb{E}_{x_{t_{i+1}}}\left[\gamma_k^{n_{k,t_i+1}} x_{k,t_i+1} + \lambda_k \gamma_k^{n_{k,t_i+1}}|x_{t_{i+1}-1}\right] = \gamma_k^{n_{k,t_{i+1}}-1+1} x_{k,t_{i+1}-1} + \lambda_k \gamma_k^{n_{k,t_{i+1}}-1+1}$$

$$= \gamma_k^{n_{k,t_i+1}} x_{k,t_{i+1}-1} + \lambda_k \gamma_k^{n_{k,t_i+1}}.$$

The third equality is true since when arm $k$ is pulled for example at time $q_n$, then we have that

$$\mathbb{E}_{q_{n+1} \sim p_{\mathcal{M}}(\cdot|x_{q_n}, k, q_n)}\left[\gamma_k^{n_{k,t_i+1}} x_{k,q_n+1} + \lambda_k \gamma_k^{n_{k,t_i+1}}\right]$$

$$= \gamma_k^{n_{k,t_i+1}}\left(\gamma_k^{n_{k,q_n}} x_{k,q_n} + \lambda_k \gamma_k^{n_{k,q_n}}\right) + \lambda_k \gamma_k^{n_{k,t_i+1}},$$

where $p_{\mathcal{M}}$ is given in Appendix C.1. The second to last last equality holds because $x_{k,t_i+1} = \mu_{k,t_i+1}(u_{k,0:t_i})$ where $\mu_{k,t}(\cdot)$ is defined in (3). $\square$

**Lemma 13.** *For any episode $i + 1$ (from time $t_i + 1$ to $t_{i+1}$), given the past arm pull sequences $\{u_{k,0:t_i}\}_{k=1}^K$ of the proposed policy $\pi_{1:t_i}$, the optimal time-dependent competitor policy $\tilde\pi_{t_i+1:t_{i+1}}$, where $\tilde\pi_t = \tilde\pi_t(x_t) = \tilde\pi_t(x_t'), \tilde\pi_t \in [K], \forall t \in [t_i + 1, t_{i+1}], x_t, x_t' \in \mathcal{X}$, for this episode is given by* `Lookahead`$(\{\lambda_k, \gamma_k, b_k\}_{k=1}^K, \{u_{k,0:t_i}\}_{k=1}^K, t_i, t_{i+1})$ *where $\{\lambda_k, \gamma_k, b_k\}_{k=1}^K$ are the true reward parameters for the rebounding bandits instance.*

*Proof.* By Lemma 12, we have that the optimal time-dependent competitor policy $\tilde\pi_{t_i+1:t_{i+1}}$ maximizes $\sum_{t=t_i+1}^{t_{i+1}} \mu_{k,t}(u_{k,0:t-1})$, by choosing $u_{k,t_i+1:t_{i+1}}$. Thus, by the definition of `Lookahead` (5), given our proposed policy $\pi_{1:t_i}$, the optimal time-dependent competitor policy is given by `Lookahead`$(\{\lambda_k, \gamma_k, b_k\}_{k=1}^K, \{u_{k,0:t_i}\}_{k=1}^K, t_i, t_{i+1})$. $\square$

*Proof of Theorem 5.* Using Lemma 13, we have that given our policy $\pi_{1:T}$ and its corresponding pull sequence $u_{k,0:t-1}$ for $k \in [K], t \in [T]$, the optimal competitor policy for episode $i + 1$ where $i \in \{0, \ldots, \lfloor T/w \rfloor\}$ (episode $i + 1$ ranges from time $t_i + 1 = iw + 1$ to $t_{i+1} = \min\{iw + w, T\}$) is given by $\texttt{Lookahead}(\{\lambda_k, \gamma_k, b_k\}_{k=1}^{K}, \{u_{k,0:t_i}\}_{k=1}^{K}, t_i, t_{i+1})$. We use $\mathbf{M}(\{\lambda_k, \gamma_k, b_k\}_{k=1}^{K}, \{u_{k,0:t_i}\}_{k=1}^{K}, t_i, t_{i+1})$ to denote the (optimal) objective value of (5) given by $\texttt{Lookahead}(\{\lambda_k, \gamma_k, b_k\}_{k=1}^{K}, \{u_{k,0:t_i}\}_{k=1}^{K}, t_i, t_{i+1})$. Denote $\bar{b} = \max_k b_k$ and $\underline{b} = \min_k b_k$.

**Exploration Stage** Recall that in Algorithm 1, we have defined $\widetilde{T} = T^{2/3} + w - (T^{2/3} \bmod w)$ which is a multiple of $w$. For the first $\widetilde{T}$ time steps, as defined in Algorithm 1, our policy $\pi_{1:\widetilde{T}}$ is a time-dependent policy, i.e., it satisfies that $\pi_t = \pi_t(x_t) = \pi_t(x_t'), \pi_t \in [K], \forall t \in [1, \widetilde{T}], x_t, x_t' \in \mathcal{X}$. Using 12, we obtain that the regret for the first $\widetilde{T}/w$ episodes is given by

$$
\sum_{i=0}^{\tilde{T}/w-1} \max_{\tilde{\pi}_{1:w} \in \mathcal{C}^w} \mathbb{E}\left[\sum_{j=1}^{w} r(x_{iw+j}, \tilde{\pi}_j(x_{iw+j})) \Big| x_{iw+1} = x^i\right]
$$

$$
- \sum_{i=0}^{\tilde{T}/w-1} \mathbb{E}\left[r(x_{iw+j}, \pi_{iw+j}(x_{iw+j})) \Big| x_{iw+1} = x^i\right]
$$

$$
\leq \sum_{i=0}^{\tilde{T}/w-1} \mathbf{M}(\{\lambda_k, \gamma_k, b_k\}_{k=1}^{K}, \{u_{k,0:iw}\}_{k=1}^{K}, iw, iw + w) - \widetilde{T}\left(\underline{b} - \frac{\bar{\lambda}\bar{\gamma}}{1 - \bar{\gamma}}\right)
$$

$$
\leq \widetilde{T}\left(\bar{b} - \underline{b} + \frac{\bar{\lambda}\bar{\gamma}}{1 - \bar{\gamma}}\right) \lesssim \widetilde{T} \lesssim T^{2/3}.
$$

since $\widetilde{T} \leq T^{2/3} + w$ and by assumption, $w \leq T^{2/3}$.

**Estimation Stage** By Theorem 3 and Corollary 4, we have that for any $\delta \in (0, 1)$ and $n \geq n_0(\delta, k)$ where $n_0(\delta, k)$ depends on $\delta$ logarithmically, with probability $1 - \delta$, for all $k \in [K]$ $|\widehat{\gamma}_k - \gamma_k| \leq \frac{C_{\gamma_k} \log(1/\delta)}{\sqrt{n}}$ and $|\widehat{\lambda}_k - \lambda_k| \leq \frac{C_{\lambda_k} \log(1/\delta)}{\sqrt{n}}$ when $\widehat{\gamma}_k > 0$.

We define two numbers $T_0' := \min_T\{T : (\sum_{k=1}^{K} n_0(k, T^{-1/3}))^{3/2} = C_1 K(\log T)^{3/2} < T\}$ and $T_0'' := \min_T\left\{T : \max_k \gamma_k + \frac{C_{\gamma_k}}{\sqrt{T^{2/3}/K}} < 1\right\}$. These two numbers exist as $T$ can be chosen to be arbitrarily large. Take $T_0 = \max\{T_0', T_0''\}$. Then for all $T \geq T_0$, with probability $1 - \delta$ where $\delta = T^{-1/3}$, we have that $\forall k \in [K], |\widehat{\gamma}_k - \gamma_k| \leq \epsilon_\gamma = O(\sqrt{K}T^{-1/3} \log T), |\widehat{\lambda}_k - \lambda_k| \leq \epsilon_\lambda = O(\sqrt{K}T^{-1/3} \log T)$ and $\left(\epsilon_\lambda \left|\frac{\widehat{\gamma}_k}{1-\widehat{\gamma}_k}\right| + \epsilon_\gamma \left|\frac{\bar{\lambda}}{(1-\widehat{\gamma}_k)(1-\gamma_k)}\right|\right) \leq O(\sqrt{K}T^{-1/3} \log T)$ since $\widehat{\gamma}_k \leq \gamma_k + \frac{C_{\gamma_k}}{\sqrt{T_0^{2/3}/K}} < 1$ and $\gamma_k \leq \bar{\gamma} < 1$.

For any pull sequence $u_{k,0:t-1}$, using our obtained estimated parameters $\{\widehat{\gamma}_k, \widehat{\lambda}_k, \widehat{b}_k\}_{k=1}^{K}$, we define the estimated reward function: for $t \geq 2$, $\widehat{\mu}_{k,t}(u_{k,0:t-1}) = b_k - \widehat{\lambda}_k\left(\sum_{i=1}^{t-1} \widehat{\gamma}_k^{t-i} u_{k,i}\right)$, and for $t = 1$, $\widehat{\mu}_{k,1}(u_{k,0:1}) = b_k = \mu_{k,1}(u_{k,0:1})$, where we note that $\widehat{b}_k = b_k$ since it is the reward of the first pull of arm $k$. Given $t \geq 2$, we have that

$$
|\mu_{k,t}(u_{k,0:t-1}) - \widehat{\mu}_{k,t}(u_{k,0:t-1})|
$$

$$
= \left|\widehat{\lambda}_k\left(\sum_{i=1}^{t-1} \widehat{\gamma}_k^{t-i} u_{k,i}\right) - \lambda_k\left(\sum_{i=1}^{t-1} \gamma_k^{t-i} u_{k,i}\right)\right|
$$

$$
= \left|\widehat{\lambda}_k\left(\sum_{i=1}^{t-1} \widehat{\gamma}_k^{t-i} u_{k,i}\right) - \lambda_k\left(\sum_{i=1}^{t-1} \widehat{\gamma}_k^{t-i} u_{k,i}\right) + \lambda_k\left(\sum_{i=1}^{t-1} \widehat{\gamma}_k^{t-i} u_{k,i}\right) - \lambda_k\left(\sum_{i=1}^{t-1} \gamma_k^{t-i} u_{k,i}\right)\right|
$$

$$
\leq |\widehat{\lambda}_k - \lambda_k|\left|\frac{\widehat{\gamma}_k}{1-\widehat{\gamma}_k}\right| + \bar{\lambda}\left|\frac{\widehat{\gamma}_k}{1-\widehat{\gamma}_k} - \frac{\gamma_k}{1-\gamma_k}\right|
$$

$$
\leq \epsilon_\lambda\left|\frac{\widehat{\gamma}_k}{1-\widehat{\gamma}_k}\right| + \epsilon_\gamma\left|\frac{\bar{\lambda}}{(1-\widehat{\gamma}_k)(1-\gamma_k)}\right|. \tag{19}
$$

**Planning Stage** Given our policy $\pi_{1:T}$ (along with its pull sequence $\{u_{k,0:T}\}_{k=1}^K$), starting from time $\widetilde{T}+1$, for any episode $i+1 \geq \widetilde{T}/w$, we denote the optimal competitor policy to be $\pi^*_{t_i+1:t_{i+1}} =$ Lookahead$(\{\lambda_k, \gamma_k, b_k\}_{k=1}^K, \{u_{k,0:t_{i-1}}\}_{k=1}^K, t_i, t_{i+1})$ where $t_i = iw$ and $t_{i+1} = \min\{iw + w, T\}$. The cumulative expected reward collected by $\pi^*_{t_i+1:t_{i+1}}$ and $\pi_{t_i+1:t_{i+1}}$ has the difference

$$\mathbf{M}(\{\lambda_k, \gamma_k, b_k\}_{k=1}^K, \{u_{k,0:t_{i-1}}\}_{k=1}^K, t_i, t_{i+1}) - \mathbf{M}(\{\widehat{\lambda}_k, \widehat{\gamma}_k, b_k\}_{k=1}^K, \{u_{k,0:t_{i-1}}\}_{k=1}^K, t_i, t_{i+1})$$

$$= \sum_{t=t_i+1}^{t_{i+1}} \mu_{\pi_t^*, t}(u^*_{\pi_t^*, 0:t-1}) - \sum_{t=t_i+1}^{t_{i+1}} \mu_{\pi_t, t}(u_{\pi_t, 0:t-1})$$

$$= \sum_{t=t_i+1}^{t_{i+1}} \mu_{\pi_t^*, t}(u^*_{\pi_t^*, 0:t-1}) - \sum_{t=t_i+1}^{t_{i+1}} \widehat{\mu}_{\pi_t^*, t}(u^*_{\pi_t^*, 0:t-1})$$

$$+ \sum_{t=t_i+1}^{t_{i+1}} \widehat{\mu}_{\pi_t^*, t}(u^*_{\pi_t^*, 0:t-1}) - \sum_{t=t_i+1}^{t_{i+1}} \widehat{\mu}_{\pi_t, t}(u_{\pi_t, 0:t-1})$$

$$+ \sum_{t=t_i+1}^{t_{i+1}} \widehat{\mu}_{\pi_t, t}(u_{\pi_t, 0:t-1}) - \sum_{t=t_i+1}^{t_{i+1}} \mu_{\pi_t, t}(u_{\pi_t, 0:t-1})$$

$$\leq \sum_{t=t_i+1}^{t_{i+1}} \mu_{\pi_t^*, t}(u^*_{\pi_t^*, 0:t-1}) - \sum_{t=t_i+1}^{t_{i+1}} \widehat{\mu}_{\pi_t^*, t}(u^*_{\pi_t^*, 0:t-1})$$

$$+ \sum_{t=t_i+1}^{t_{i+1}} \widehat{\mu}_{\pi_t, t}(u_{\pi_t, 0:t-1}) - \sum_{t=t_i+1}^{t_{i+1}} \mu_{\pi_t, t}(u_{\pi_t, 0:t-1}).$$

where $u^*_{\pi_t^*, 0:t-1}$ is the corresponding pull sequence of arm $\pi_t^*$ under policy $\pi_{1:t}^* = (\pi_{1:t_i}, \pi^*_{t_i+1:t})$, and the last inequality holds because $\pi_{t_i+1:t_{i+1}} = $ Lookahead$(\{\widehat{\lambda}_k, \widehat{\gamma}_k, \widehat{b}_k\}_{k=1}^K, \{u_{k,0:t_i}\}_{k=1}^K, t_i, t_{i+1})$ is the optimal solution under the estimated parameters $\{\widehat{\lambda}_k, \widehat{\gamma}_k, \widehat{b}_k\}_{k=1}^K$ and $\pi$'s previous past pull sequence $\{u_{k,0:t_i}\}_{k=1}^K$. Further, using (19) and the fact that $t_i - t_{i-1} \leq w$, we obtain that

$$\sum_{t=t_i+1}^{t_{i+1}} \mu_{\pi_t^*, t}(u^*_{\pi_t^*, 0:t-1}) - \sum_{t=t_i+1}^{t_{i+1}} \widehat{\mu}_{\pi_t^*, t}(u^*_{\pi_t^*, 0:t-1})$$

$$+ \sum_{t=t_i+1}^{t_{i+1}} \widehat{\mu}_{\pi_t, t}(u_{\pi_t, 0:t-1}) - \sum_{t=t_i+1}^{t_{i+1}} \mu_{\pi_t, t}(u_{\pi_t, 0:t-1})$$

$$\leq 2w \max_k \left( \epsilon_\lambda \left| \frac{\widehat{\gamma}_k}{1 - \widehat{\gamma}_k} \right| + \epsilon_\gamma \left| \frac{\widehat{\lambda}}{(1 - \widehat{\gamma}_k)(1 - \gamma_k)} \right| \right).$$

Finally, putting it altogether, we have obtained that for all $T \geq T_0$,

$$\text{Reg}^w(T) = \sum_{i=0}^{\lceil T/w \rceil - 1} \max_{\widetilde{\pi}_{1:w} \in \mathcal{C}^w} \mathbb{E}\left[ \sum_{j=1}^{\min\{w, T-iw\}} r(x_{iw+j}, \widetilde{\pi}_j(x_{iw+j})) \Big| x_{iw+1} = x^i \right]$$

$$- \mathbb{E}\left[ \sum_{j=1}^{\min\{w, T-iw\}} r(x_{iw+j}, \pi_{iw+j}(x_{iw+j})) \Big| x_{iw+1} = x^i \right]$$

$$\leq O(T^{2/3}) + (1 - T^{-1/3}) \left( \sum_{i=T/w}^{\lceil T/w \rceil - 1} 2wO(\sqrt{K}T^{-1/3}\log T) \right) + T^{-1/3}\left( T\left( \overline{b} - \underline{b} + \frac{\overline{\lambda}\overline{\gamma}}{1 - \overline{\gamma}} \right) \right)$$

$$\leq O(T^{2/3}) + (T - T^{2/3})O(\sqrt{K}T^{-1/3}\log T) + O(T^{2/3})$$

$$\leq O(\sqrt{K}T^{2/3}\log T),$$

which we notice that with probability $\delta = T^{-1/3}$, the cumulative expected reward from time $\widetilde{T}$ to $T$ between the optimal competitor policy and our policy $\pi$ is at most $T\left( \overline{b} - \underline{b} + \frac{\overline{\lambda}\overline{\gamma}}{1 - \overline{\gamma}} \right)$. This completes the proof. $\qquad \square$

### E.2  Exploration Strategies

In the exploration phase of Algorithm 1 (from time 1 to $\widetilde{T}$), in addition to playing each arm repeatedly for $\widetilde{T}/K$ times, in general, we could explore by playing each arm at a fixed interval, i.e., the time interval between two consecutive pulls of arm $k$ should be a constant $m_k$. For example, this includes playing the arms cyclically with the cylce being $1, 2, \ldots, K$ or playing the first two arms in an alternating fashion from time 1 to $2\widetilde{T}/K$, then the next two arms, etc. As shown in Theorem 8 and Corollary 9, using the datasets (of size $n$) collected by these exploration strategies, we can obtain estimators $\widehat{\gamma}_k$ and $\widehat{\lambda}_k$ with the estimation error being on the order of $O(1/\sqrt{n})$. Using these results (in replacement of Theorem 3 and Corollary 4 in the estimation stage of the proof of Theorem 5), we can obtain that there exists $T_0$ such that for all $T \geq T_0$, the regret upper bound of EEP under these exploration strategies are of order $O(\sqrt{K}T^{2/3}\log T)$.

# F  Additional Proofs of Appendix C

## F.1  Proof of Corollary 7

*Proof.* Fix $\delta \in (0,1)$. By Theorem 6, for all $k \in [K]$, with probability $1 - \frac{\delta}{2K}$, we have the following:
When $m = 1$, then $|\widehat{\gamma}_k - \gamma_k| = ||\widehat{a}_k| - a_k| \le |\widehat{a}_k - a_k| \le \epsilon_a(n, \frac{\delta}{2K}, k)$. When $m \ge 2$, we have that

$$|\widehat{\gamma}_k - \gamma_k| = \left| \frac{|\widehat{a}_k| - a_k}{|\widehat{a}_k|^{(m-1)/m} + |\widehat{a}_k|^{(m-2)/m}\gamma_k + \ldots + \gamma_k^{m-1}} \right| \le \frac{|\widehat{a}_k - a_k|}{\gamma_k^{m-1}}.$$

On the other hand, given that $|\widehat{a}_k - a_k| \le \epsilon_a(n, \frac{\delta}{2K}, k)$, we have that with probability $1 - \frac{\delta}{2K}$,

$$|\widehat{\lambda}_k - \lambda_k| = \left| \left| \frac{\widehat{d}_k}{\widehat{a}_k} \right| - \frac{d_k}{a_k} \right| \le \left| \frac{\widehat{d}_k}{\widehat{a}_k} - \frac{d_k}{\widehat{a}_k} \right| + \left| \frac{d_k}{\widehat{a}_k} - \frac{d_k}{a_k} \right| \le \frac{\epsilon_d(n, \frac{\delta}{2K}, k)}{\widehat{a}_k} + \frac{\lambda_k \epsilon_a(n, \frac{\delta}{2K}, k)}{\widehat{a}_k} \le O\left( \frac{1}{\sqrt{n}} \right).$$

The proof completes as follows:

$$\mathbb{P}\left( \forall k, |\widehat{\gamma}_k - \gamma_k| \le \frac{|\widehat{a}_k - a_k|}{\gamma_k^{m-1}}, |\widehat{\lambda}_k - \lambda_k| \le \frac{\epsilon_d(n, \frac{\delta}{2K}, k)}{\widehat{a}_k} + \frac{\lambda_k \epsilon_a(n, \frac{\delta}{2K}, k)}{\widehat{a}_k} \right) \ge \prod_{k=1}^{K} \left( 1 - \frac{\delta}{2K} \right)^2 \ge 1 - \delta,$$

where the last inequality follows from Bernoulli's inequality. $\square$

## F.2  Proof of Lemma 4

*Proof.* Let $\pi_{1:T}$ denote the sequence that policy $\pi$ will take from time $1$ to $T$. By the definition of the value function, we have that

$$V_{1,\mathcal{M}}^{\pi}(x_{\text{init}}) = b_{\pi_1} + \sum_{t=2}^{T} \mathbb{E}_{x_2, \ldots, x_t} [r(x_t, \pi_t)],$$

where $x_t \sim p_{\mathcal{M}}(\cdot | x_{t-1}, \pi_{t-1}, t-1)$ is a state vector drawn from the transition distribution defined in Section C.1. Let $k$ denote $\pi_t$ and $u_{k,0:t-1}$ denote the past pull sequence for arm $k$ under policy $\pi$. If $k$ has not been pulled before time $t$, then $\mathbb{E}_{x_2, \ldots, x_t}[r(x_t, \pi_t)] = b_{\pi_t} = \mu_{\pi_t, t}(u_{\pi_t, 0:t-1})$. If $k$ has been pulled before, then let $t_1, \ldots, t_n$ denote the time steps that arm $k$ has been pulled before time $t$. We have that

$$
\begin{aligned}
\mathbb{E}_{x_2, \ldots, x_t}[r(x_t, k)] &= b_k - \left( \mathbb{E}_{x_2, \ldots, x_{t-1}} \left[ \mathbb{E}_{x_t \sim p_{\mathcal{M}}(\cdot | x_{t-1}, k, t-1)} \left[ \gamma_k^{n_{k,t}} x_{k,t} + \lambda_k \gamma_k^{n_{k,t}} \right] \right] \right) \\
&= b_k - \left( \mathbb{E}_{x_2, \ldots, x_{t_n}} \left[ \mathbb{E}_{x_{t_n+1} \sim p_{\mathcal{M}}(\cdot | x_{t_n}, k, t_n)} \left[ \gamma_k^{n_{k,t}} x_{k, t_n+1} + \lambda_k \gamma_k^{n_{k,t}} \right] \right] \right) \\
&= b_k - \left( \mathbb{E}_{x_2, \ldots, x_{t_n}} \left[ \gamma_k^{n_{k,t}} \left( \gamma_k^{n_{k,t_n}} x_{k, t_n} + \lambda_k \gamma_k^{n_{k,t_n}} \right) + \lambda_k \gamma_k^{n_{k,t}} \right] \right) \\
&= \ldots = b_k - \lambda_k \left( \gamma_k^{n_{k,t}} + \gamma_k^{n_{k,t} + n_{k,t_n}} + \ldots + \gamma_k^{n_{k,t} + n_{k,t_n} + \ldots n_{k,t_1}} \right) \\
&= \mu_{k,t}(u_{k, 0:t-1}),
\end{aligned}
$$

where we note that the second equality is true because when arm $k$ is not pulled for example at time $t - 1$, the state for arm $k$ at time $t - 1$ will satisfy that $x_{k,t} = x_{k,t-1}$ and $n_{k,t} = n_{k,t-1} + 1$ with probability 1. In this case, we have that $\mathbb{E}_{x_t \sim p_{\mathcal{M}}(\cdot | x_{t-1}, k, t-1)} \left[ \gamma_k^{n_{k,t}} x_{k,t} + \lambda_k \gamma_k^{n_{k,t}} \right] = \gamma_k^{n_{k,t-1}+1} x_{k,t-1} + \lambda_k \gamma_k^{n_{k,t-1}+1} = \gamma_k^{n_{k,t}} x_{k,t-1} + \lambda_k \gamma_k^{n_{k,t}}$. The third equality is true since when arm $k$ is pulled for example at time $t-1$, then we have that $\mathbb{E}_{x_t \sim p_{\mathcal{M}}(\cdot | x_{t-1}, k, t-1)} \left[ \gamma_k^{n_{k,t}} x_{k,t} + \lambda_k \gamma_k^{n_{k,t}} \right] = \gamma_k^{n_{k,t}} \left( \gamma_k^{n_{k,t-1}} x_{k,t-1} + \lambda_k \gamma_k^{n_{k,t-1}} \right) + \lambda_k \gamma_k^{n_{k,t}}$. The proof completes by summing over $\mathbb{E}_{x_2, \ldots, x_t}[r(x_t, \pi_t)]$ for all $t \ge 2$. $\square$

## F.3  Proof of Proposition 5

*Proof.* Fix $\delta \in (0,1)$. Let $E_1$ be the event that

$$\forall k \in [K], \ |\widehat{\gamma}_k - \gamma_k| = \epsilon_{\gamma_k} \le O\left( \frac{1}{\sqrt{n}} \right), \ |\widehat{\lambda}_k - \lambda_k| = \epsilon_{\lambda_k} \le O\left( 1/\sqrt{n} \right).$$

From Corollary 7, we have that $\mathbb{P}(E_1) \ge 1 - \delta$. Let $\pi_{1:T}$ denote the sequence that policy $\pi$ will take from time $1$ to $T$. From Lemma 4, we have that

$$|V_{1,\mathcal{M}}^{\pi}(x_{\text{init}}) - V_{1,\widehat{\mathcal{M}}}^{\pi}(x_{\text{init}})| = \left| \sum_{t=1}^{T} \mu_{\pi_t, t}(u_{\pi_t, 0:t-1}) - \widehat{\mu}_{\pi_t, t}(u_{\pi_t, 0:t-1}) \right|,$$

where $u_{\pi_t,0:t-1}$ is the past pull sequence for arm $\pi_t$ under policy $\pi$ before time $t$ and $\widehat{\mu}_{k,t}(u_{k,0:t-1}) = b_k - \widehat{\lambda}_k \left( \sum_{i=1}^{t-1} \widehat{\gamma}_k^{t-i} u_{k,i} \right)$ for $t \geq 2$ and $\widehat{\mu}_{k,1}(u_{k,0:1}) = b_k = \mu_{k,1}(u_{k,0:1})$. Given $t \geq 2$, let $k$ denote $\pi_t$, we have that

$$|\mu_{k,t}(u_{k,0:t-1}) - \widehat{\mu}_{k,t}(u_{k,0:t-1})|$$

$$= \left| \widehat{\lambda}_k \left( \sum_{i=1}^{t-1} \widehat{\gamma}_k^{t-i} u_{k,i} \right) - \lambda_k \left( \sum_{i=1}^{t-1} \gamma_k^{t-i} u_{k,i} \right) \right|$$

$$= \left| \widehat{\lambda}_k \left( \sum_{i=1}^{t-1} \widehat{\gamma}_k^{t-i} u_{k,i} \right) - \lambda_k \left( \sum_{i=1}^{t-1} \widehat{\gamma}_k^{t-i} u_{k,i} \right) + \lambda_k \left( \sum_{i=1}^{t-1} \widehat{\gamma}_k^{t-i} u_{k,i} \right) - \lambda_k \left( \sum_{i=1}^{t-1} \gamma_k^{t-i} u_{k,i} \right) \right|$$

$$\leq |\widehat{\lambda}_k - \lambda_k| \left| \frac{\widehat{\gamma}_k}{1 - \widehat{\gamma}_k} \right| + \overline{\lambda} \left| \frac{\widehat{\gamma}_k}{1 - \widehat{\gamma}_k} - \frac{\gamma_k}{1 - \gamma_k} \right|$$

$$\leq \frac{\widehat{\gamma}_k \epsilon_{\lambda_k}}{1 - \widehat{\gamma}_k} + \frac{\overline{\lambda} \epsilon_{\gamma_k}}{(1 - \widehat{\gamma}_k)(1 - \gamma_k)}$$

Since $\widehat{\gamma}_k < 1$ ($\widehat{a}_k \in (\underline{a}, \overline{a})$) almost surely and with probability $1 - \delta$, for all $k \in [K]$, $\epsilon_{\gamma_k} \leq O\left(1/\sqrt{n}\right)$ and $\epsilon_{\lambda_k} \leq O\left(1/\sqrt{n}\right)$. We have that with probability $1 - \delta$,

$$\left| \sum_{t=1}^{T} \mu_{\pi_t,t}(u_{\pi_t,0:t-1}) - \widehat{\mu}_{\pi_t,t}(u_{\pi_t,0:t-1}) \right| \leq \sum_{t=1}^{T} |\mu_{\pi_t,t}(u_{\pi_t,0:t-1}) - \widehat{\mu}_{\pi_t,t}(u_{\pi_t,0:t-1})| \leq \left( \frac{T}{\sqrt{n}} \right).$$

$\square$

### F.4 Proof of Proposition 6

*Proof.* Fix $\delta \in (0,1)$. Let $E_1$ be the event that

$$\forall k \in [K], \ |\widehat{\gamma}_k - \gamma_k| = \epsilon_{\gamma_k} \leq O\left( \frac{1}{\sqrt{n}} \right), \ |\widehat{\lambda}_k - \lambda_k| = \epsilon_{\lambda_k} \leq O\left(1/\sqrt{n}\right).$$

From Corollary 9, we have that $\mathbb{P}(E_1) \geq 1 - \delta/2$. Let $\epsilon_\lambda := \max_k \epsilon_{\lambda_k}$. Let $E_2$ denote the event that $\forall t \in [T], k \in [K], |x_{k,t}| \leq B(\delta/2)$ (10). We know that $\mathbb{P}(E_2) \geq 1 - \delta/2$. When $E_1$ and $E_2$ happen, we first observe that for all positive integer $n$ and $k \in [K]$,

$$|\widehat{\gamma}_k^n - \gamma_k^n| \leq |\widehat{\gamma}_k - \gamma_k| \left( n \max(\gamma_k^{n-1}, \widehat{\gamma}_k^{n-1}) \right) \leq \frac{|\widehat{\gamma}_k - \gamma_k|}{\max(\gamma_k, \widehat{\gamma}_k) \ln(1/\max(\gamma_k, \widehat{\gamma}_k))} = O(1/\sqrt{n}),$$

whereand the second inequality uses the assumption that $\widehat{a}_k, \gamma_k$ are bounded away from 0 and 1.

To continue, we first bound the distance between the transition function in $\widehat{\mathcal{M}}$ and $\mathcal{M}$. At any any time $t$ and state $x_t = (x_{1,t}, n_{1,t}, \ldots, x_{K,t}, n_{K,t})$, when we pull arm $\pi_t = k$, the next state $x_{t+1}$ is updated by: (i) for arm $k$, $n_{k,t+1} = 1$ and (ii) for all other arms $k' \neq k$, $n_{k,t+1} = n_{k,t} + 1$ if $n_k \neq 0$, $n_{k,t+1} = 0$ if $n_{k,t} = 0$, and $x_{k',t+1} = x_{k',t}$. Then, by [8, Theorem 1.3], we have that when $n_{\pi_t,t} \neq 0$,

$$\|p_{\widehat{\mathcal{M}}}(x_{t+1}|x_t, \pi_t, t) - p_{\mathcal{M}}(x_{t+1}|x_t, \pi_t, t)\|_1$$

$$\overset{(*)}{\leq} \frac{3|\widehat{\lambda}_k^2 \sum_{i=0}^{n_{k,t}-1} \widehat{\gamma}_k^{2i} - \lambda_k^2 \sum_{i=0}^{n_{k,t}-1} \gamma_k^{2i}|}{\lambda_k^2 \sum_{i=0}^{n_k-1} \gamma_k^{2i}} + \frac{|\gamma_k^{n_{k,t}} x_{k,t} + \lambda_k \gamma_k^{n_{k,t}} - \widehat{\gamma}_k^{n_{k,t}} x_{k,t} - \widehat{\lambda}_k \widehat{\gamma}_k^{n_{k,t}}|}{\lambda_k \sqrt{\sum_{i=0}^{n_{k,t}-1} \gamma_k^{2i}}}$$

$$= \frac{3|\widehat{\lambda}_k^2 \left( \sum_{i=0}^{n_{k,t}-1} \widehat{\gamma}_k^{2i} - \sum_{i=0}^{n_{k,t}-1} \gamma_k^{2i} \right) + (\widehat{\lambda}_k^2 - \lambda_k^2) \sum_{i=0}^{n_{k,t}-1} \gamma_k^{2i}|}{\lambda_k^2 \sum_{i=0}^{n_k-1} \gamma_k^{2i}}$$

$$+ \frac{|\widehat{\gamma}_k^{n_{k,t}} - \gamma_k^{n_{k,t}}| B(\delta/2) + |\lambda_k \gamma_k^{n_{k,t}} - \widehat{\lambda}_k \widehat{\gamma}_k^{n_{k,t}}|}{\lambda_k \sqrt{\sum_{i=0}^{n_{k,t}-1} \gamma_k^{2i}}}$$

$$\overset{(**)}{\leq} 3 \left| \widehat{\lambda}_k^2 \left( \sum_{i=0}^{n_{k,t}-1} \widehat{\gamma}_k^{2i} - \sum_{i=0}^{n_{k,t}-1} \gamma_k^{2i} \right) + (\widehat{\lambda}_k^2 - \lambda_k^2) \sum_{i=0}^{n_{k,t}-1} \gamma_k^{2i} \right| + |\widehat{\gamma}_k^{n_{k,t}} - \gamma_k^{n_{k,t}}| (B(\delta/2) + \lambda_k)$$

$$+ |\lambda_k \widehat{\gamma}_k^{n_{k,t}} - \widehat{\lambda}_k \widehat{\gamma}_k^{n_{k,t}}|$$

$$\leq 3 \left( (\lambda_k + \epsilon_\lambda)^2 \left| \frac{1}{1 - \widehat{\gamma}_k^2} - \frac{1}{1 - \gamma_k^2} \right| + \frac{|\widehat{\lambda}_k - \lambda_k|(2\lambda_k + \epsilon_\lambda)}{1 - \gamma_k^2} \right) + |\widehat{\gamma}_k^{n_{k,t}} - \gamma_k^{n_{k,t}}| \left( B(\delta/2) + \lambda_k \right) + |\lambda_k - \widehat{\lambda}_k|$$

$$= 3 \left( \frac{(\lambda_k + \epsilon_\lambda)^2 |\widehat{\gamma}_k^2 - \gamma_k^2|}{(1 - \widehat{\gamma}_k^2)(1 - \gamma_k^2)} + \frac{|\widehat{\lambda}_k - \lambda_k|(2\lambda_k + \epsilon_\lambda)}{1 - \gamma_k^2} \right) + |\widehat{\gamma}_k^{n_{k,t}} - \gamma_k^{n_{k,t}}| \left( B(\delta/2) + \lambda_k \right) + |\lambda_k - \widehat{\lambda}_k|$$

$$=: \epsilon_P = O\left( \frac{1}{\sqrt{n}} \right),$$

where $(*)$ holds since $p_{\mathcal{M}}(x_{t+1}|x_t, \pi_t, t)$ is a Gaussian density with mean $\gamma_k^{n_{k,t}} x_{k,t} + \lambda_k \gamma_k^{n_{k,t}}$ and variance $\lambda_k^2 \sum_{i=0}^{n_k - 1} \gamma_k^{2i}$ and $(**)$ uses the fact that $\lambda_k^2 \sum_{i=0}^{n_k - 1} \gamma_k^{2i} \geq \lambda_k^2 \geq 1$. When $n_{\pi_t, t} = 0$ and condition (i) and (ii) are fulfilled, we have that $\| p_{\widehat{\mathcal{M}}}(x_{t+1}|x_t, \pi_t, t) - p_{\mathcal{M}}(x_{t+1}|x_t, \pi_t, t) \|_1 = 0$. Otherwise, that is, if condition (i) or (ii) is not satisfied, we also have that $\| p_{\widehat{\mathcal{M}}}(x_{t+1}|x_t, \pi_t, t) - p_{\mathcal{M}}(x_{t+1}|x_t, \pi_t, t) \|_1 = 0$ since $p_{\widehat{\mathcal{M}}}(x_{t+1}|x_t, \pi_t, t) = p_{\mathcal{M}}(x_{t+1}|x_t, \pi_t, t) = 0$. Next, we examine the difference of the expected reward obtained by pulling arm $k$ at state $x_t$ at time $t$ in MDP $\mathcal{M}$ and $\widehat{\mathcal{M}}$; when $n_{k,t} \neq 0$, this is given by

$$|\widehat{r}(x_t, k)] - r(x_t, k)]| = |\gamma_k^{n_{k,t}} x_{k,t} + \lambda_k \gamma_k^{n_{k,t}} - \widehat{\gamma}_k^{n_{k,t}} x_{k,t} - \widehat{\lambda}_k \widehat{\gamma}_k^{n_{k,t}}|$$
$$\leq |x_{k,t}| \cdot |\gamma^{n_{k,t}} - \widehat{\gamma}^{n_{k,t}}| + |\lambda_k \gamma_k^{n_{k,t}} - \lambda_k \widehat{\gamma}_k^{n_{k,t}} + \lambda_k \widehat{\gamma}_k^{n_{k,t}} - \widehat{\lambda}_k \widehat{\gamma}_k^{n_{k,t}}|$$
$$\leq (B(\delta/2) + \lambda_k) |\widehat{\gamma}_k^{n_{k,t}} - \gamma_k^{n_{k,t}}| + |\widehat{\lambda}_k - \lambda_k| =: \epsilon_R = O\left( \frac{1}{\sqrt{n}} \right),$$

where $\widehat{r}(x_t, k)$ is the expected reward of pulling arm $k$ at state $x_t$ in MDP $\widehat{\mathcal{M}}$. Putting it altogether, we have that for any deterministic policy $\pi$,

$$V_{1,\mathcal{M}}^\pi(x_{\text{init}}) - V_{1,\widehat{\mathcal{M}}}^\pi(x_{\text{init}}) = r(x_{\text{init}}, \pi_1(x_{\text{init}})) - \widehat{r}(x_{\text{init}}, \pi_1(x_{\text{init}})) + \mathbb{E}_{x_2 \sim p_{\mathcal{M}}(\cdot|x_1, \pi, 1)}[V_{2,\mathcal{M}}^\pi(x_2)]$$
$$- \mathbb{E}_{x_2 \sim p_{\widehat{\mathcal{M}}}(\cdot|x_1, \pi, 1)}[V_{2,\widehat{\mathcal{M}}}^\pi(x_2)]$$
$$\leq \epsilon_R + \mathbb{E}_{x_2 \sim p_{\mathcal{M}}(\cdot|x_1, \pi, 1)}[V_{2,\mathcal{M}}^\pi(x_2)] - \mathbb{E}_{x_2 \sim p_{\widehat{\mathcal{M}}}(\cdot|x_1, \pi, 1)}[V_{2,\mathcal{M}}^\pi(x_2)]$$
$$+ \mathbb{E}_{x_2 \sim p_{\widehat{\mathcal{M}}}(\cdot|x_1, \pi, 1)}[V_{2,\mathcal{M}}^\pi(x_2)] - \mathbb{E}_{x_2 \sim p_{\widehat{\mathcal{M}}}(\cdot|x_1, \pi, 1)}[V_{2,\widehat{\mathcal{M}}}^\pi(x_2)]$$
$$\leq T\epsilon_R + \sum_{t=1}^{T} \mathbb{E}_{\widehat{\mathcal{M}}, \pi} \Big\{ \mathbb{E}_{x_{t+1} \sim p_{\mathcal{M}}(\cdot|x_t, \pi, t)}[V_{t+1,\mathcal{M}}^\pi(x_{t+1})]$$
$$- \mathbb{E}_{x_{t+1} \sim p_{\widehat{\mathcal{M}}}(\cdot|x_t, \pi, t)}[V_{t+1,\mathcal{M}}^\pi(x_{t+1})] \Big\}$$
$$\leq T\epsilon_R + T^2 \epsilon_P \max_k b_k,$$

where $p_{\mathcal{M}}(\cdot|x_t, \pi, t)$ denotes $p(\cdot|x_t, \pi_t(x_t), t)$ in MDP $\mathcal{M}$ and the last inequality uses the fact that $\langle p_{\mathcal{M}}(\cdot|x_t, \pi, t) - p_{\widehat{\mathcal{M}}}(\cdot|x_t, \pi, t), V_{t+1,\mathcal{M}}^\pi \rangle \leq \| p_{\mathcal{M}}(\cdot|x_t, \pi, t) - p_{\widehat{\mathcal{M}}}(\cdot|x_t, \pi, t) \|_1 \| V_{t+1,\mathcal{M}}^\pi \|_\infty \leq \epsilon_P T \max_k b_k$. Finally, we have that

$$V_{1,\mathcal{M}}^*(x_{\text{init}}) - V_{1,\mathcal{M}}^{\pi_{\widehat{\mathcal{M}}}^*}(x_{\text{init}}) = V_{1,\mathcal{M}}^{\pi_{\mathcal{M}}^*}(x_{\text{init}}) - V_{1,\widehat{\mathcal{M}}}^{\pi_{\mathcal{M}}^*}(x_{\text{init}}) + V_{1,\widehat{\mathcal{M}}}^{\pi_{\mathcal{M}}^*}(x_{\text{init}}) - V_{1,\widehat{\mathcal{M}}}^{\pi_{\widehat{\mathcal{M}}}^*}(x_{\text{init}})$$
$$+ V_{1,\widehat{\mathcal{M}}}^{\pi_{\widehat{\mathcal{M}}}^*}(x_{\text{init}}) - V_{1,\mathcal{M}}^{\pi_{\widehat{\mathcal{M}}}^*}(x_{\text{init}}) \leq 2T\epsilon_R + 2T^2 \epsilon_P \max_k b_k,$$

where the equation follows from the fact that $V_{1,\mathcal{M}}^*(x_{\text{init}}) = V_{1,\mathcal{M}}^{\pi_{\mathcal{M}}^*}(x_{\text{init}})$ and rearranging the terms, and the inequality follows from applying the bound of $V_{1,\mathcal{M}}^\pi(x_{\text{init}}) - V_{1,\widehat{\mathcal{M}}}^\pi(x_{\text{init}}) \leq T\epsilon_R + T^2 \epsilon_P \max_k b_k$ that was derived above for $\pi = \pi_{\mathcal{M}}^*$ and $\pi = \pi_{\widehat{\mathcal{M}}}^*$ and using the fact that the policy $\pi_{\widehat{\mathcal{M}}}^*$ is optimal for MDP $\widehat{\mathcal{M}}$. Let $E_3$ denote the event that $V_{1,\mathcal{M}}^*(x_{\text{init}}) - V_{1,\mathcal{M}}^{\pi_{\widehat{\mathcal{M}}}^*}(x_{\text{init}}) \leq O(T^2/\sqrt{n})$. Putting it altogether, we have that $\mathbb{P}(E_3) \geq \mathbb{P}(E_2, E_1) = 1 - \mathbb{P}(E_2^c \cup E_1^c) \geq 1 - \delta$. $\qquad \square$

# G  Additional Experimental Details and Results

We present additional experimental details and results.

$w$**-lookahead Performance**    When evaluating the performance of $w$-lookahead policies, in addition to the case where $T = 30$ (Figure 3a), we have also run the experiments with $T = 100$ (Figure 4a). When solving for the 100-lookahead policy, we have increased the number of threads to $50$ to solve for (4) and stopped the program at a time limit of $24$ hours. In such settings, we obtain an upper bound on the absolute optimality gap of $64.0$ (percentage optimality gap of $13.0\%$). When solved for $w$-lookahead policies with $w$ in between 1 and 15 using 10 threads, Gurobi ends up solving (5) within $40$s for all different $w$ values. Thus, despite using significantly lower computational time, $w$-lookahead policies achieve a similar expected cumulative reward to the $T$-lookahead policies (see Figures 3a and 4a).

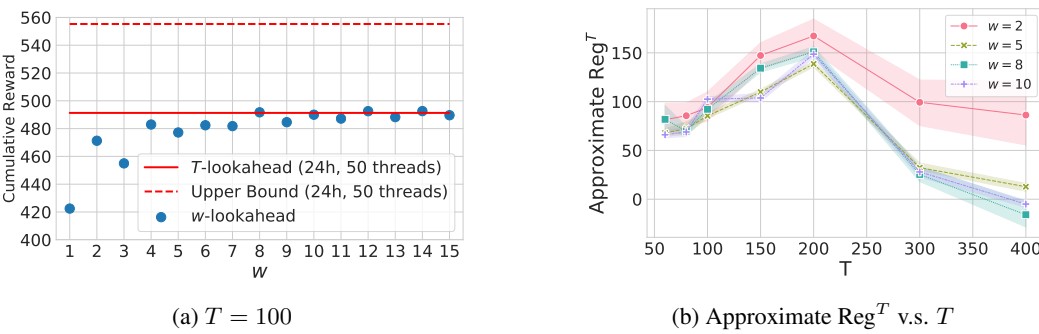

(a) $T = 100$                       (b) Approximate Reg$^T$ v.s. $T$

Figure 4: Figure 4a shows the expected cumulative reward collected by $w$-lookahead policies (blue dots) when $T = 100$. When solving for the $T$-lookahead policy (solving (4) with $T = 100$), after $24$ hours, Gurobi 9.1 obtains an objective value of $491.3$ (red solid line) with an upper bound $555.3$ (red dotted line) and an absolute optimality gap $64.0$ ($13.0\%$). The true expected cumulative reward for $T$-lookahead policy for this problem lies in between the solid and dotted red lines. Figure 4b shows the approximate $T$-step lookahead regret of $w$-lookahead EEP. The reason why it is an approximate $T$-step lookahead regret is that the $T$-lookahead policy used to obtain the regret is set to be the one attained by Gurobi using 25 threads for 24 hours. The percentage optimality gaps for these attained policies are $6.8\%, 10.1\%, 12.4\%, 14.2\%, 15.1\%, 33.5\%, 39.8\%$ when $T = 60, 80, 100, 150, 200, 300, 400$, respectively. The results are averaged over 20 random runs.

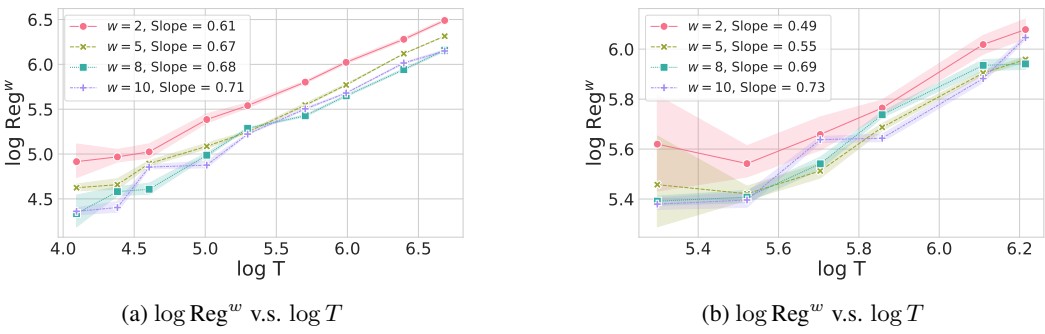

(a) $\log \mathrm{Reg}^w$ v.s. $\log T$                   (b) $\log \mathrm{Reg}^w$ v.s. $\log T$

Figure 5: Figure 5a shows the log-log plot of the $w$-step lookahead regret of $w$-lookahead EEP (averaged over 20 random runs) under different $T$ when there are 5 arms. Figure 5b shows the log-log plot of the $w$-step lookahead regret of $w$-lookahead EEP (averaged over 20 random runs) under different $T$ when there are 10 arms.

**EEP Performance**    Figure 3b is the log-log plot of the $w$-step lookahead regret of $w$-lookahead EEP against the horizon $T$ when $T = 60, 80, 100, 150, 200, 300, 400$ (averaged over 20 random runs) and Figure 5a is the log-log plot when $T = 60, 80, 100, 150, 200, 300, 400, 600, 800$ (averaged over 20 random runs), under the experimental setup provided in § 7.

To compare the $w$-lookahead EEP under the same regret definition, we present the (approximate) $T$-step lookahead regret for these policies (Figure 4b). We note that in order to obtain the $T$-step lookahead regret (6), we need to find the $T$-lookahead policy which requires us to solve (4) when $T = 60, 80, 100, 150, 200, 300, 400$. As we have noted earlier and demonstrated empirically (Figure 4a), solving (4) for large $T$ can be computationally intractable. In contrast, $w$-lookahead EEP only requires us to solve much smaller programs (5). In Figure 4b, we use the policy attained by Gurobi using 25 threads for 24 hours as the competitor policy against $w$-lookahead EEP to obtain the approximate $T$-step lookahead regret. The percentage optimality gaps for these attained approximate $T$-lookahead policies are $6.8\%, 10.1\%, 12.4\%, 14.2\%, 15.1\%, 33.5\%, 39.8\%$ when $T = 60, 80, 100, 150, 200, 300, 400$, respectively. Notably, there are cases when the $w$-lookahead EEP outperforms the attained approximate $T$-lookahead policies, resulting in negative approximate $T$-step lookahead regret.

Finally, we present the result when we include 5 additional arms to the existing problem. The 5 new arms have parameters $\gamma_6 = .4, \gamma_7 = .5, \gamma_8 = .6, \gamma_9 = .8, \gamma_{10} = .7, \lambda_6 = 2, \lambda_7 = 3, \lambda_8 = 2, \lambda_9 = 3, \lambda_{10} = 1$, and $b_6 = 10, b_7 = 5, b_8 = 6, b_9 = 7, b_{10} = 8$. Figure 5b is the log-log plot of the $w$-step lookahead regret of $w$-lookahead EEP against the horizon $T$ when $T = 200, 250, 300, 350, 450, 500$ (averaged over 20 random runs).