# OpenReview forum: "Rebounding Bandits for Modeling Satiation Effects"
_NeurIPS.cc/2021/Conference — NeurIPS 2021 Poster_

### Official Review · Reviewer_rYye · 2021-07-14

**Rating:** 7
**Confidence:** 4

**Summary:**

This paper introduces a problem of modeling satiation effects in multi-armed bandits. For example, if we assume the arms are restaurants that an agent is going to over time. Said agent may have a preference for pizza hut, but eating there every night would lead to a lower reward than it would if you haven't been to pizza hut in a while. The authors prove that the greedy algorithm is optimal when satiation retention, base rewards, and satiation influence is known and is equivalent over all arms. The authors then provide an upper bound on regret when using their EEP (Explore, Estimate, Plan) policy. The authors also provide empirical results on a toy version of the problem.

**Limitations And Societal Impact:**

The authors thoroughly provided limitations throughout the paper and included remarks to address potential questions about limitations. As this is a highly theoretical paper the authors provided sufficiently discussed potential negative societal impacts by stating that only certain aspects of human behavior can be captured in this setting and that more work is needed before deploying this in practice.

**Main Review:**

I have read through the paper in detail but did not read carefully through the proofs provided in the appendices. Based on that information, I believe the theoretical results to be sound. The problem of rebounding bandits for this kind of satiation effects is a novel contribution to this area. I also appreciate the literature review of the psychology behind satiation effects and how the authors formulation of the problem fits into the current understanding of satiation. One question I have is did the authors observe any cyclic-like behavior in the setting of unknown dynamics if given a sufficient enough T?

**Time Spent Reviewing:**

3

---

> ### Author Response · Authors · 2021-08-10
> **Reply to Reviewer rYye**
>
> Thanks for the thoughtful review. We are glad that you appreciated the rebounding bandits problem setup and the literature review.
>
> *Re: “Did the authors observe any cyclic-like behavior in the setting of unknown dynamics if given a sufficient enough T?”*
>
> We expect that with a sufficiently long horizon T, the optimal strategy may be periodic. However, when the arms are not identical, the cycle may not be playing all arms once (e.g., 1, 2, …, K). It would be an interesting future work to characterize the optimal policy under an infinite horizon.

---

### Official Review · Reviewer_q6Cg · 2021-07-16

**Rating:** 7
**Confidence:** 3

**Summary:**

The paper proposes a Multi-armed Bandits model with non-stationary reward affected by users' satiation level. The authors consider two cases of satiation dynamic: known deterministic dynamics and unknown stochastic dynamics. Then the authors show that under specific conditions a greedy algorithm is optimal. In the general setting (unknown stochastic satiation dynamics), they propose a two-step algorithm EEP with error estimation and analysis.

**Limitations And Societal Impact:**

The authors have adequately addressed the limitations and potential negative societal impact of their work.

**Main Review:**

The paper proposes an interesting framework modelling users non-stationary interests on the same item, which is practical in many real-world scenes, and novel in bandit learning. The proposed model is also general enough to cover both deterministic and stochastic cases of satiation level. The experiment results look promising. Also, the paper is well written and easy to follow.

**Time Spent Reviewing:**

3h

---

> ### Author Response · Authors · 2021-08-10
> **Reply to Reviewer q6Cg**
>
> Thanks for your thoughtful review. We are glad that you appreciated the novelty and applicability of our framework.

---

### Official Review · Reviewer_kguD · 2021-07-16

**Rating:** 6
**Confidence:** 4

**Summary:**

This paper introduced rebounding bandits where satiation dynamics are modeled as time-invariant linear dynamical systems. With consecutive exposures to the same item, the expected rewards for each arm decline monotonically, and whenever that arm is not pulled, the reward rebounds. It first studied the offline problem and then characterized the regret bound for the proposed online algorithm. The paper propose Explore-Estimate-Plan and proves that the greedy policy is optimal when the arms exhibit identical deterministic dynamics. It also claims that this is the first bandits paper to model evolving rewards through continuous-state linear stochastic dynamical systems with unknown parameters.


**Main Review:**

In general, the paper is well written and I enjoy reading the paper. I have the following main comments:

1. Can you specify the key difference between your model and 1) recharging bandits 2) non-stationary bandits? I understand that you model evolving rewards through continuous-state linear stochastic dynamical systems, but 1) from the modeling perspective, what additional features does it capture? Is it a more general model?; 2) from the technical aspect, what is the technical contribution compared to the existing literature?

2. Can you characterize the computational cost for solving the w-lookahead policy (formula (5))? The algorithm can be inefficient if it is the \sqrt{T}-lookahead policy.

3. In the definition of regret in Section 5.2, the regret is defined for the w-step lookahead. It seems doable that in each step, the algorithm solves the w-step lookahead policy, but it only plays the arm for the current state. When the response is received, it moves to the next time step and resolves the optimization problem. It may perform better if the policy is planned ahead of time and update at each time step.

4. For the online learning part, can UCB-type algorithm be designed to improve the regret from O(T^{2/3}) to O(T^{1/2})? The algorithm is suboptimal since in the K-armed bandits case, the regret of Algorithm 1 is still O(T^{2/3}). UCB-type algorithm seems possible since larger value of gamma and smaller value of d is more optimistic.

5. It would help to show the empirical computational cost for different values of w in the numerical section.


**Time Spent Reviewing:**

4

---

> ### Author Response · Authors · 2021-08-10
> **Reply to Reviewer kguD**
>
> Thanks for your thoughtful review. Please find responses to each of your main comments below.
>
> *Re: key differences vs 1)  non-stationary bandits 2) recharging bandits*
>
> First, non-stationary bandits refers to a broad family of problems, of which rebounding bandits and recharging bandits are both instances. These problems are differentiated by the specific structure of nonstationarity.
>
> In recharging bandits, the expected rewards for all arms increase as concave functions of the time since they were last pulled. By contrast, in rebounding bandits, the expected rewards depend on the entire pull sequence. Thus, if an arm were pulled repeatedly for many steps and then rested for 5 steps, the satiation level would be higher than if it were pulled only once 5 steps ago. From a technical perspective, to the best of our knowledge, rebounding bandits is the first bandits paper to model evolving rewards through continuous-state linear stochastic dynamical systems with unknown parameters. By virtue of casting our problem this way, we are able to leverage recent results in system identification of linear dynamical systems.
>
>
> *Re: “the computational cost for solving the w-lookahead policy”*
>
> Computing the w-lookahead policy involves solving a binary integer program with $Kw$ binary variables. The worst case computational complexity of this is $2^{Kw}$. As such, $\sqrt{T}$-lookahead policies may have a large computational cost when $T$ is large. On the other hand, our numerical study illustrates that this computation time is minor (in seconds) for moderate values of $w$.
>
> *Re: “[the lookahead policy] may perform better if the policy is planned ahead of time and update at each time step.”*
>
> While we originally considered this policy where we update at every timestep, we adopted the current policy to lighten the computational load. For the future manuscript of the paper, we will add this policy into our experimental section.
>
> *Re: “Can UCB-type algorithm be designed to improve the regret from O(T^{2/3}) to O(T^{1/2})?”*
>
> Coming up with an UCB-type algorithm for general rebounding bandits instance is a very interesting future direction. As commented in L305-308, the main technical obstacle would lie in the identification of the arm parameters: “When the rewards of each arm are not observed periodically, the obtained satiation influences can no longer be viewed as samples from the same time-invariant affine dynamical system, since the parameters of the system depend on the duration between pulls.”
>
> *Re: “It would help to show the empirical computational cost for different values of w in the numerical section.”*
>
> In the current experimental setup (Figure 3a), all the w-lookahead policies are solved within 2s (L333-335). The T-lookahead policy was solved in 1610s. We plan to include more numerical results on the computational cost of w-lookahead policy in future manuscripts of the paper.

---

### Official Review · Reviewer_GJkC · 2021-07-20

**Rating:** 6
**Confidence:** 4

**Summary:**

This paper considers a class of Multi-Armed Bandits in which the rewards on arms decline monotonically with consecutive arm pulls and rebound to the base value when arms are not pulled. The paper distinguishes itself from previous similar approaches, which assume the reward to collapse to some “zero” after an arm pull and recharge with time — in this paper, the degree of satiation effect depends on the number of recent pulls (not just the last pull).

The paper first provides analysis for this problem setting with all arms sharing the same deterministic and known satiation dynamics and shows that the optimal policy in this case pulls arms cyclically. Further, the paper analyses the stochastic setting with unknown satiation dynamics, providing an algorithm to explore and plan in this situation and provides regret bounds for the same. Finally, the paper concludes with numerical evaluation to (1) evaluate the proposed algorithm and (2) run validation checks for the theoretical regret bounds.


**Limitations And Societal Impact:**

The authors acknowledge some technical limitations of their work (but not negative societal impact in particular). I don't foresee any negative social impact of this work either.

**Main Review:**

Strengths/Positives:

1. The problem is well motivated. Previous relevant work studying similar bandit classes seems to assume that the bandit reward only depends on the time since the last pull. However, the added complexity of this paper fits the motivation well.

2. Paper is very engaging to read and is written well. The paper does a good job of introducing the problem setup (through illustrative examples, etc.), distinguishing it from previous related work and the overall narrative/flow of the paper is great.

3. Theoretical analysis in the paper seems to be solid -- there are regret bounds presented for the proposed algorithm and other theoretical guarantees for the estimation errors for the other satiation parameters.

Weaknesses/Questions/Negatives:

1. I’m not sure the linear satiation dynamics make perfect sense for the following reason: In the linear satiation dynamics defined in Equation 1, it seems with enough number of arms pulls, the (degree of satiation) > (base reward). In that case the reward accrued from an arm pull would be negative. In other words, inserting extra arm pulls (more resources) can be detrimental? This seems a little counterintuitive.
2. One of the contributions of the paper is the observation that when all arms have the same rewards and satiation dynamics, the optimal policy is cyclic. However, this result seems a little trivial to me -- because intuitively, that seems like the natural thing to do (maximizing the gap between consecutive pulls of the same arm).
3. Line 49: Why are states partially observable when the satiation dynamics are known but stochastic? If the decay factor \gamma is known, can’t we observe the satiation effect of a previous arm pull, when the arm is pulled next?
4. The w-lookahead policy doesn’t seem like a great solution to the problem (more in the following point on experiments): It seems to be still very expensive for large w and inaccurate for small w. The theoretical bound in Line 190, seems pretty lossy for w= sqrt(T). Similarly for the expression in Theorem2, the bound can blow up when T is large and w<T with \gamma close to 1. So while the theoretical analysis and the guarantee is good to have, the bound doesn’t seem very promising.
5. Experiments section seems a little weak:

a) Experiments are run with just 5 arms in the simulation. It would be good to see what happens to the performance and runtime (scalability of the algorithm) for larger problem sizes.

b) In Fig.3, the results are averaged over just 5 runs. It would be good to have atleast 30 or 50 independent runs and/or have error bars in Fig 3(a).

c) While the figures establish the validity of the theoretical regret bounds presented earlier, it’s hard to judge the actual performance of the algorithms because there are no baselines. For instance, in Fig 3(b), there seems to be a sharp drop in regret when w increased from just w=2 to w=10. That indicates this regret could be much higher in comparison to the w=T policy (not shown in the figure), meaning w=2 or 10 is probably a bad policy?

d) Part of Fig 2 is hard to read because of the legends. Having legends outside of the chart would be great.


Other comments:

1. Line 58: Point (iv) is hard to understand, without reading the rest of the paper.
2. Typos: Line 143: “to to”


**Time Spent Reviewing:**

12

---

> ### Author Response · Authors · 2021-08-10
> **Reply to Reviewer GJkC**
>
> Thanks for your thoughtful review. We briefly touch on each of your primary concerns.
>
> *Re: Sensibility of linear satiation dynamics, possibility of negative rewards*
>
> First, we would like to clarify that it is not necessarily true that with enough consecutive arm pulls, the expected reward will eventually be negative. In our setup,  $\gamma_k < 1$, and thus with endless pulls the expected satiation level converges to $\gamma_k/(1-\gamma_k)$ for each arm $k$. For any arm $k$, whether or not it is possible to hit a negative expected reward depends on the values of $\lambda_k$, $\gamma_k$ and the base reward $b_k$.
>
> In our setup, there is no intrinsic significance to “positive” vs “negative” reward. All that matters are the relative values of the reward received for taking one action versus another. However, we might imagine a version of our setup equipped with a “rest” action that always gives a reward of 0 and allows the other arms to rebound. In this case, the value 0 would become significant, and indeed it would never make sense to pull an arm when it gives a negative expected reward.
>
>
> *Re: Triviality of the cyclic policy result (under identical rewards and satiation dynamics)*
>
> It might seem intuitive that the optimal policy under identical rewards and satiation dynamics would pick arms cyclically. However, while this turns out to hold for our satiation dynamics, (i) proving it is not trivial; and (ii) it does not hold under some alternative satiation dynamics (e.g., if the expected reward was calculated by dividing the base reward by the satiation level $b_k/(1 + \lambda_k s_{k,t})$, rather than applying a subtractive penalty).
>
> *Re: “Line 49: Why are states partially observable when the satiation dynamics are known but stochastic? If the decay factor \gamma is known, can’t we observe the satiation effect of a previous arm pull, when the arm is pulled next?”*
>
> While we can observe the satiation level of a given arm at the moment when we pull it, we cannot observe the current satiation levels of all the other arms that we haven’t pulled. Because the satiation levels evolve with (unobserved) stochastic noises between pulls, the problem is partially observable.
>
> *Re: “The w-lookahead policy … very expensive for large w and inaccurate for small w … bound can blow up when T is large and w<T with \gamma close to 1”*
>
> While in theory, our bound can be pessimistic in the worst case for large T and $\gamma$ close to 1, its empirical performance paints a more favorable picture. In our current experimental setup, it performs similarly to the T-lookahead policy when w is relatively small (i.e., 5) and T=30. We plan to include more empirical studies on w-lookahead policy in future manuscripts of the paper.
>
> Similarly, although the computation of w-lookahead policy involves solving a binary integer program with $Kw$ binary variables, in our numerical studies (T=30), we find that the computation requirements are comparatively modest (seconds for 15 steps), compared to solving for the T-lookahead policy (~26 minutes).
>
> *Re: Experiments section seems a little weak*
> - *“Experiments are run with just 5 arms in the simulation.”*
>   - We will include more empirical results with larger problem instances (more arms) in future manuscripts of the paper.
> - *“In Fig.3, the results are averaged over just 5 runs. It would be good to have at least 30 or 50 independent runs and/or have error bars in Fig 3(a).”*
>   - For all reported results, we will average the results across more runs in future manuscripts of the paper.
> - *“hard to judge the actual performance of the algorithms because there are no baselines.”*
>   - We will add a baseline for w=T to contextualize all of our experimental results in future manuscripts of the paper.
> - *“Part of Fig 2 is hard to read because of the legends. Having legends outside of the chart would be great.”*
>   - We will move the legend outside the figure in the future manuscripts of the paper.

---

> > ### Comment · Reviewer_GJkC · 2021-08-14
> > **Response to Rebuttal**
> >
> > Thanks for the detailed comments.
> >
> > I'm satisfied with the responses to comments #2, #3 and somewhat satisfied with the response to comment #4.
> >
> > I am confused about the response to #1 and want to check if I'm missing something:
> >
> > > I agree that "it is not necessarily true... that rewards will be negative with endless pulls", but I'm worried that in *some* situations (for suitable choice of parameters), the rewards would become negative with continuous pulls. I believe that is true?
> > If that is true, secondly, in the current setting, rewards are always accrued only from arms that are pulled (from equation 2), so the default reward from arms not pulled is zero? Isn't that correct?
> > If the above is also true, wouldn't it mean "inserting additional arm pulls in the optimal schedule could be detrimental"? Or in other words, wouldn't it sometimes be optimal to not pull an arm, even though it's the best arm to pull (potentially breaking the algorithm proposed)?

---

> > > ### Author Response · Authors · 2021-08-18
> > > **Reply to Reviewer GJkC**
> > >
> > > Thanks for your quick reply. We are glad that you are satisfied with our answers to comments #2, #3, and (somewhat) #4, and believe based on your reply that we are close to reaching consensus on comment  #1.
> > >
> > > Re parameter choices: yes it is true that there exist choices of parameters for which the rewards can go negative. We gently point out that our contribution is the framework itself and tools for estimating the parameters and planning, all of which apply regardless of whether one restricts attention to specific ranges of the parameters. In short, the discussion here centers not on whether our framework is reasonable or contributions substantial, but on whether certain parameter choices are reasonable.
> > >
> > > That said, we are also happy to continue the discussion on why/when we should think that there is anything special about the number “0”. In the traditional bandits setup, one must pull an arm at every time step, and there is no possibility of not pulling any arm. Thus, all that matters are the *relative rewards* and the problem is mathematically identical, regardless of whether the reward ranges from -10 to 0, or -5 to 5, or 0 to 10. In short, the traditional bandit setup does not have such an allowable action as “inserting additional arm pulls” or “not pulling any arms”.
> > >
> > > However, we could augment our setup by adding a special “rest arm” that always returns reward 0 (zero base reward and no satiation influence). This seems natural in many settings where, e.g.,  arms correspond to eating foods, or listening to songs, since individuals do have an option to “not consume”. Here, by virtue of adding the always-available “default” option to collect zero reward, it truly matters whether the other expected rewards are non-negative.
> > >
> > > Finally, we add that even in this setup, one might construct examples where negative rewards are reasonable. Say the recommender is recommending songs (and pulling an arm means a song is recommended). Additionally, say that the recommender has a “rest arm” that encodes the choice to recommend no song at all and collect 0 reward. If the recommender repeatedly recommends some song that the user quickly tires of (e.g., MMMBop by Hansen), then perhaps the user enjoys listening to it the first time, and is only a little annoyed on the second listen, but soon comes to experience “negative” reward. Here the interpretation of negative reward would be that this item is less preferred relative to not being recommended.

---

> > > > ### Author Response · Authors · 2021-09-02
> > > > **Following up**
> > > >
> > > > Dear Reviewer GJkC,
> > > >
> > > > Thanks again for taking the time to engage with us. To recap, we are glad to know that you are satisfied with our answers to comments #2, #3, and (somewhat) #4. We wanted to reach out to see if our subsequent response to your question on #1 addressed your concerns. If so, would you consider updating your score to reflect the resolution of these issues? And if not, would you please help us to understand any residual issues that we might help to resolve? Thanks again for your time and your detailed reply!

---

### Author Response · Authors · 2021-08-10
**General Reply**

We thank the reviewers for their positive and detailed reviews. In particular, we were glad to see that all reviewers recommend acceptance and that they generally appreciated the clarity and thoroughness our exposition (GJkC, kguD, q6Cg, rYye), the clear motivation (GJkC, q6Cg), the novelty of the framework (GJkC, q6Cg, rYye), and the solidity of our theoretical results (GJkC, rYye). We also appreciated the many clarifying questions and constructive suggestions and have responded to each reviewer’s comments in the respective threads.

---

### Decision · Program_Chairs · 2021-09-27

**Decision:**

Accept (Poster)

**Comment:**

The rebounding bandits model proposed and analyzed in the paper was appreciated by all the reviewers as an interesting and relevant problem, and by and large the algorithm and the analysis and the presentation was considered to be a decent contribution. The reviewers raised a number of questions in their reviews, and the authors did a reasonable job of addressing most of them. It will be critical that the authors revise the paper to ensure that the issues raised are addressed and to prevent some of the misunderstandings from persisting with the general reader. Two specific issues that lingered with the reviewers: (1) the issue of negative rewards generated some discussion. I do think the authors’ elaboration in the discussion with one of the reviewers is legitimate, but this point should be included/clarified in the paper. (2) The empirical results are quite weak (specifically, number of arms and number of runs in results, baseline (e.g., w=T)) . The author response to all questions about these matters is that they will be addressed “in future manuscripts of the paper.” I’m not sure what this means: in fact, I would almost consider it to be a condition of acceptance that these be included in the revised paper